# ELUCIDATING THE SOLUTION SPACE OF EXTENDED REVERSE-TIME SDE FOR DIFFUSION MODELS

## ABSTRACT

Diffusion models (DMs) demonstrate potent image generation capabilities in various generative modeling tasks. Nevertheless, their primary limitation lies in slow sampling speed, requiring hundreds or thousands of sequential function evaluations through large neural networks to generate high-quality images. Sampling from DMs can be seen alternatively as solving corresponding stochastic differential equations (SDEs) or ordinary differential equations (ODEs). In this work, we formulate the sampling process as an extended reverse-time SDE (ER SDE), unifying prior explorations into ODEs and SDEs. Leveraging the semi-linear structure of ER SDE solutions, we offer exact solutions and approximate solutions for VP SDE and VE SDE, respectively. Based on the solution space of the ER SDE, we yield mathematical insights elucidating the superior performance of ODE solvers over SDE solvers in terms of fast sampling. Additionally, we unveil that VP SDE solvers stand on par with their VE SDE counterparts. Finally, we devise fast and training-free samplers, ER-SDE-Solvers, achieving state-of-the-art performance across all stochastic samplers. Experimental results demonstrate achieving 3.45 FID in 20 function evaluations and 2.24 FID in 50 function evaluations on the ImageNet $64 \times 64$ dataset.

## 1 INTRODUCTION

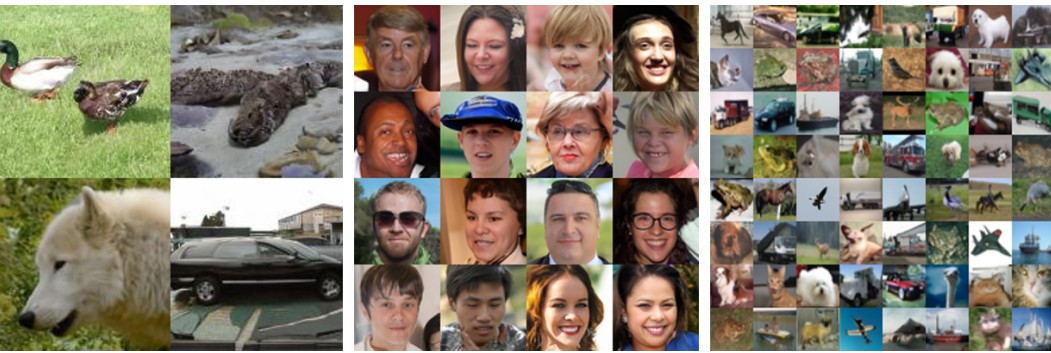

Figure 1: Samples generated using Ours (ER-SDE-Solver-3) on ImageNet $128 \times 128$ (Left, FID=8.33), FFHQ $64 \times 64$ (Middle, FID=4.67) and CIFAR-10 (Right, FID=3.15) when NFE=20.

Diffusion Models (DMs) demonstrate an aptitude for producing high-quality samples and exhibit a stable training process that is resilient to disruptions. They have found extensive utilization in diverse generative modeling tasks, encompassing image synthesis (Dhariwal & Nichol, 2021; Ho et al., 2020; Song et al., 2021b), image super-resolution (Saharia et al., 2022b; Gao et al., 2023), image restoration (Chung et al., 2022; Luo et al., 2023), image editing (Avrahami et al., 2022; Meng et al., 2021), image-to-image translation (Zhao et al., 2022; Su et al., 2022), and similar domains. However, in comparison to alternative generative models like Generative Adversarial Networks (GANs) (Goodfellow et al., 2014), DMs frequently necessitate multiple function evaluations, constraining their applicability in real-time scenarios.

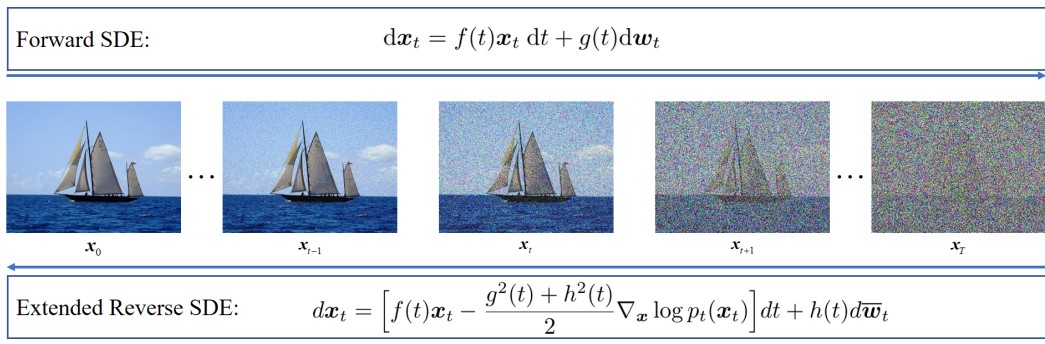

| Forward SDE: | $\mathrm{d}\boldsymbol{x}_t = f(t)\boldsymbol{x}_t\,\mathrm{d}t + g(t)\mathrm{d}\boldsymbol{w}_t$ |
|---|---|

| Extended Reverse SDE: | $d\boldsymbol{x}_t = \left[ f(t)\boldsymbol{x}_t - \dfrac{g^2(t) + h^2(t)}{2}\nabla_{\boldsymbol{x}}\log p_t(\boldsymbol{x}_t)\right]dt + h(t)d\overline{\boldsymbol{w}}_t$ |

Figure 2: A unified framework for DMs: The forward process described by an SDE transforms real data into noise, while the reverse process characterized by an ER SDE generates real data from noise. Once the score function $\nabla_{\mathbf{x}}\log p_t(\mathbf{x}_t)$ is estimated by a neural network, solving the ER SDE enables the generation of high-quality samples.

Hence, a substantial body of research revolves around designing fast diffusion model samplers. Fast samplers of DMs can be categorized into training-based and training-free methods. While training-based methods can generate high quality samples at $2\sim 4$ sampling steps (Salimans & Ho, 2021; Meng et al., 2023; Song et al., 2023), the prerequisite for retraining renders them cost-intensive. Conversely, training-free methods directly utilize raw information without retraining, offering broad applicability and high flexibility. Song et al. (2021b) have indicated that the image generation process is equivalent to solving stochastic differential equations (SDEs) or ordinary differential equations (ODEs) in reverse time. The essence of training-free methods lies in designing efficient solvers for SDEs (Bao et al., 2022; Zhang et al., 2023) or ODEs (Lu et al., 2022a; Zhang & Chen, 2023). Presently, accomplishments based on ODE solvers have been particularly notable. For instance, UniPC (Zhao et al., 2023) achieves rapid sampling up to 10 steps, yet exploration into works based on SDE solvers remains limited. Compared with ODE-based deterministic sampling, SDE-based stochastic samplers are more challenging due to the injection of additional noise into the data state at each sampling step. However, observations indicate that SDE-based stochastic samplers show promise in producing data of superior quality when increasing sampling steps (Song et al., 2021b; Xue et al., 2023), motivating us to explore the efficient SDE-based stochastic samplers further.

In this work, we present a unified framework for DMs, wherein Extended SDE formulation is proposed (see Fig.2). Within this framework, we define a solution space and design some highly effective SDE solvers. Specifically, we first model the sampling process as an Extended Reverse-Time (ER) SDE, which is an extension of Song et al. (2021b) and Zhang & Chen (2023). Inspired by Lu et al. (2022a), we unveil the semi-linear structure inherent in the solutions of the ER SDE—comprising linear functions of data variables, nonlinear functions parameterized by neural networks and noise terms. Building on it, we deduce exact solutions for both Variance Exploding (VE) SDE and Variance Preserving (VP) SDE (Song et al., 2021b). This is achieved by the analytical computation of the linear portions and noise terms, thereby circumventing associated discretization errors. We also offer practical approximations for both VP SDE and VE SDE.

By analyzing the errors of the approximate solutions of the ER SDE, we discover that varying levels of discretization errors emerge due to the incorporation of different noise scales during the reverse process. This phenomenon gives rise to the solution space inherent in the ER SDE. We ascertain that the optimal solution within this space aligns with ODE, which mathematically demonstrates that the performance of SDE-based methods falls short when the number of steps is limited. We also theoretically establish that the VP SDE solvers yield image quality equivalent to VE SDE solvers, given the consistency of the employed pretrained models. Moreover, through selecting the noise scale functions carefully, we devise some specialized Extended Reverse-Time SDE Solvers (ER-SDE-Solvers), which rival ODE solvers in terms of rapid sampling.

In summary, we have made several theoretical and practical contributions: 1) We formulate an ER SDE and provide an exact solution as well as approximations for VP SDE and VE SDE, respectively. 2) Through a rigorous analysis of errors in the approximate solutions, we establish mathematically that the performance of SDE solvers for rapid sampling is inferior to that of ODE solvers. Moreover, VP SDE solvers achieve the same level of image quality compared with VE SDE solvers. 3) We present specialized ER-SDE-Solvers for rapid sampling. Extensive experimentation reveals that ER-

SDE-Solvers achieve state-of-the-art performance across all stochastic samplers. Specifically, they obtain 3.45 FID in 20 NFE and 2.24 FID in 50 NFE on the ImageNet $64 \times 64$ dataset, while 3.09 FID in 20 NFE and 1.97 FID in 50 NFE on the CIFAR-10 dataset.

## 2 DIFFUSION MODELS

Diffusion models (DMs) represent a category of probabilistic generative models encompassing both forward and backward processes. During the forward process, DMs gradually incorporate noise at different scales, while noise is gradually eliminated to yield real samples in the backward process. In the context of continuous time, the forward and backward processes can be described by SDEs or ODEs. In this section, we primarily review the stochastic differential equations (SDEs) and ordinary differential equations (ODEs) pertinent to DMs.

### 2.1 FORWARD DIFFUSION SDEs

The forward process can be expressed as a linear SDE (Kingma et al., 2021):

$$\mathrm{d}\boldsymbol{x}_t = f(t)\boldsymbol{x}_t\,\mathrm{d}t + g(t)\mathrm{d}\boldsymbol{w}_t, \quad \boldsymbol{x}_0 \sim p_0\left(\boldsymbol{x}_0\right), \tag{1}$$

where $\boldsymbol{x}_0 \in \mathbb{R}^D$ is a D-dimensional random variable following an unknown probability distribution $p_0(\boldsymbol{x}_0)$. $\{\boldsymbol{x}_t\}_{t\in[0,T]}$ denotes each state in the forward process, and $\boldsymbol{w}_t$ stands for a standard Wiener process. When the coefficients $f(t)$ and $g(t)$ are piecewise continuous, a unique solution exists (Oksendal, 2013). By judiciously selecting these coefficients, Eq.(1) can map the original data distribution to a priory known tractable distribution $p_T(x_T)$, such as the Gaussian distribution.

The selection of $f(t)$ and $g(t)$ in Eq.(1) is diverse. Based on the distinct noise employed in SMLD (Song & Ermon, 2019) and DDPM (Ho et al., 2020; Sohl-Dickstein et al., 2015), two distinct SDE formulations (Song et al., 2021b) are presented.

**Variance Exploding (VE) SDE:** The noise perturbations used in SMLD can be regarded as the discretization of the following SDE:

$$\mathrm{d}\boldsymbol{x}_t = \sqrt{\frac{\mathrm{d}\sigma_t^2}{\mathrm{d}t}}\,\mathrm{d}\boldsymbol{w}_t, \tag{2}$$

where $\sigma_t$ is the positive noise scale. As $t \to \infty$, the variance of this stochastic process also tends to infinity, thus earning the appellation of Variance Exploding (VE) SDE.

**Variance Preserving (VP) SDE:** The noise perturbations used in DDPM can be considered as the discretization of the following SDE:

$$\mathrm{d}\boldsymbol{x}_t = \frac{d\log\alpha_t}{dt}\boldsymbol{x}_t\mathrm{d}t + \sqrt{\frac{d\sigma_t^2}{dt} - 2\frac{d\log\alpha_t}{dt}\sigma_t^2}\mathrm{d}\boldsymbol{w}_t, \tag{3}$$

where $\alpha_t$ is also the positive noise scale. Unlike the VE SDE, the variance of this stochastic process remains bounded as $t \to \infty$. Therefore, it is referred to as Variance Preserving (VP) SDE.

### 2.2 REVERSE DIFFUSION SDEs

The backward process can similarly be described by a reverse-time SDE (Song et al., 2021b):

$$\mathrm{d}\boldsymbol{x}_t = \left[f(t)\boldsymbol{x}_t - g^2(t)\nabla_{\boldsymbol{x}}\log p_t\left(\boldsymbol{x}_t\right)\right]\mathrm{d}t + g(t)\mathrm{d}\overline{\boldsymbol{w}}_t, \quad \boldsymbol{x}_T \sim p_T\left(\boldsymbol{x}_T\right), \tag{4}$$

where $\overline{\boldsymbol{w}}_t$ is the standard Wiener process in the reverse time. $p_t(\boldsymbol{x}_t)$ represents the probability distribution of the state $\boldsymbol{x}_t$, and its logarithmic gradient $\nabla_{\boldsymbol{x}}\log p_t\left(\boldsymbol{x}_t\right)$ is referred to as the score function, which is often estimated by a neural network $\boldsymbol{s}_\theta(\boldsymbol{x}_t, t)$.

There are also studies (Zhang & Chen, 2023; 2021) that consider the following reverse-time SDE:

$$\mathrm{d}\boldsymbol{x}_t = \left[f(t)\boldsymbol{x}_t - \frac{1+\lambda^2}{2}g^2(t)\nabla_{\boldsymbol{x}}\log p_t\left(\boldsymbol{x}_t\right)\right]\mathrm{d}t + \lambda g(t)\mathrm{d}\overline{\boldsymbol{w}}_t, \quad \boldsymbol{x}_T \sim p_T\left(\boldsymbol{x}_T\right), \tag{5}$$

where the parameter $\lambda \geq 0$. Eq.(5) similarly shares the same marginal distribution as Eq.(1).

Once the score-based network $s_\theta(x_t, t)$ is trained, generating images only requires solving the reverse-time SDE in Eq.(4) or Eq.(5). The conventional ancestral sampling method (Ho et al., 2020) can be viewed as a first-order SDE solver (Song et al., 2021b), yet it needs thousands of function evaluations to generate high-quality images. Numerous endeavors (Jolicoeur-Martineau et al., 2021; Bao et al., 2022) have sought to enhance sampling speed by devising highly accurate SDE solvers, but they still require hundreds of function evaluations, presenting a gap compared to ODE solvers.

## 2.3 REVERSE DIFFUSION ODEs

In the backward process, in addition to directly solving the reverse-time SDE in Eq.(4), a category of methods (Zhao et al., 2023; Lu et al., 2022a; Zhang & Chen, 2023) focuses on solving the probability flow ODE corresponding to Eq.(4), expressed specifically as

$$\mathrm{d}x_t = \left[f(t)x_t - \frac{1}{2}g^2(t)\nabla_x \log p_t(x_t)\right]\mathrm{d}t, \quad x_T \sim p_T(x_T). \tag{6}$$

Eq.(6) shares the same marginal distribution at each time $t$ with the SDE in Eq.(4), and the score function $\nabla_x \log p_t(x_t)$ can also be estimated by a neural network. Unlike SDEs, which introduce stochastic noise at each step, ODEs correspond to a deterministic sampling process. Despite several experiments (Song et al., 2021b) suggesting that ODE solvers outperform SDE solvers in terms of efficient sampling, SDE solvers can generate higher-quality images with a minimal increase in NFE.

In summary, SDE-based methods can generate higher-quality samples, but exhibit slower convergence in high dimensions (Kloeden & Platen, 1992b; Lu et al., 2022a). Conversely, ODE-based methods demonstrate the opposite behavior. To strike a balance between high-quality and efficiency in the generation process, in Sec.3, we model the backward process as an extended SDE, and provide analytical solutions as well as approximations for both VP SDE and VE SDE. Furthermore, we devise some SDE solvers in Sec.4, whose efficiency is comparable to that of ODE solvers.

## 3 EXTENDED REVERSE-TIME SDE SOLVER FOR DMs

There are three types of methods for recovering samples from noise in DMs. The first predicts the noise added in the forward process, achieved by a noise prediction model $\epsilon_\theta(x_t, t)$ (Ho et al., 2020). The second utilizes a score prediction model $s_\theta(x_t, t)$ to match the score function $\nabla_x \log p_t(x_t)$ (Song et al., 2021b; Hyvärinen & Dayan, 2005). The last directly restores the original data from the noisy samples, achieved by a data prediction model $x_\theta(x_t, t)$. These models can be mutually derived (Kingma et al., 2021). Previously, most SDE solvers (Song et al., 2021b; Jolicoeur-Martineau et al., 2021; Bao et al., 2022) have relied on the score-based model. Based on modeling the backward process as an extended SDE in Sec.3.1, we proceed to solve the VE SDE and VP SDE for the data prediction model in Sec.3.2 and Sec.3.3, respectively. Compared with the other two types of models, the data prediction model can be synergistically combined with thresholding methods (Ho et al., 2020; Saharia et al., 2022a) to mitigate the adverse impact of large guiding scales, thereby finding broad application in guided image generation (Lu et al., 2022b).

## 3.1 EXTENDED REVERSE-TIME SDE

Besides Eq.(4) and Eq.(5), an infinite variety of diffusion processes can be employed. In this paper, we consider the following family of SDEs (referred to as Extended Reverse-Time SDE (ER SDE)):

$$dx_t = \left[f(t)x_t - \frac{g^2(t) + h^2(t)}{2}\nabla_x \log p_t(x_t)\right]dt + h(t)d\overline{w}_t. \tag{7}$$

The score function $\nabla_x \log p_t(x_t)$ can be estimated using the pretrained neural network. Hence, generating samples only requires solving Eq.(7), which is guaranteed by Proposition 1.

**Proposition 1** (The validity of the ER SDE, proof in Appendix A.1). *When* $s_\theta(x_t, t) = \nabla_x \log p_t(x_t)$ *for all* $x_t$, $\overline{p}_T(x_T) = p_T(x_T)$, *the marginal distribution* $\overline{p}_t(x_t)$ *of Eq.(7) matches* $p_t(x_t)$ *of the forward diffusion Eq.(1) for all* $0 \le t \le T$.

Eq.(7) extends the reverse-time SDE proposed in Song et al. (2021b); Zhang & Chen (2023); Xue et al. (2023); Karras et al. (2022). Specifically, in Song et al. (2021b), the noise scale $g(t)$ added at

each time step $t$ of the reverse process is the same as that of the corresponding moment in the forward process. Zhang & Chen (2023); Xue et al. (2023); Karras et al. (2022) introduce a non-negative parameter to control the extent of noise added during the reverse process. However, the form of the noise scale is relevant to $g(t)$. In contrast, our ER SDE introduces a completely new noise scale $h(t)$ for the reverse process. This implies that the noise scale $h(t)$ added during the reverse process may not necessarily be correlated with the scale $g(t)$ of the forward process. Particularly, the ER SDE reduces to the reverse-time SDE in Eq.(4), Eq.(5) and the ODE depicted in Eq.(6) respectively when the specific values of $h(t)$ are chosen.

By expanding the reverse-time SDE, we not only unify ODEs and SDEs under a single framework, facilitating the comparative analysis of these two methods, but also lay the groundwork for designing more efficient SDE-based samplers. Further details are discussed in Sec.4.

## 3.2 VE ER-SDE-SOLVERS

For the VE SDE, $f(t) = 0$ and $g(t) = \sqrt{\frac{d\sigma_t^2}{dt}}$ (Kingma et al., 2021). The relationship between score prediction model $s_\theta(x_t, t)$ and data prediction model $x_\theta(x_t, t)$ is $-[x_t - x_\theta(x_t, t)]/\sigma_t^2 = s_\theta(x_t, t)$. By replacing the score function with the data prediction model, Eq.(7) can be expressed as

$$d\boldsymbol{x}_t = \frac{1}{2\sigma_t^2}\Big[\frac{d\sigma_t^2}{dt} + h^2(t)\Big][\boldsymbol{x}_t - \boldsymbol{x}_\theta(\boldsymbol{x}_t, t)]dt + h(t)d\overline{\boldsymbol{w}}_t. \tag{8}$$

Denote $d\boldsymbol{w}_\sigma := \sqrt{\frac{d\sigma_t}{dt}}d\overline{\boldsymbol{w}}_t$, $h^2(t) = \xi(t)\frac{d\sigma_t}{dt}$, we can rewrite Eq.(8) w.r.t $\sigma$ as

$$d\boldsymbol{x}_\sigma = \Big[\frac{1}{\sigma} + \frac{\xi(\sigma)}{2\sigma^2}\Big][\boldsymbol{x}_\sigma - \boldsymbol{x}_\theta(\boldsymbol{x}_\sigma, \sigma)]d\sigma + \sqrt{\xi(\sigma)}d\boldsymbol{w}_\sigma. \tag{9}$$

We propose the exact solution for Eq.(9) using *variation-of-constants* formula (Lu et al., 2022b).

**Proposition 2** (Exact solution of the VE SDE, proof in Appendix A.2). *Given an initial value $\boldsymbol{x}_s$ at time $s > 0$, the solution $\boldsymbol{x}_t$ at time $t \in [0, s]$ of VE SDE in Eq.(9) is:*

$$\boldsymbol{x}_t = \underbrace{\frac{\phi(\sigma_t)}{\phi(\sigma_s)}\boldsymbol{x}_s}_{(a)\ Linear\ term} + \underbrace{\phi(\sigma_t)\int_{\sigma_t}^{\sigma_s}\frac{\phi^{(1)}(\sigma)}{\phi^2(\sigma)}\boldsymbol{x}_\theta(\boldsymbol{x}_\sigma, \sigma)d\sigma}_{(b)\ Nonlinear\ term} + \underbrace{\sqrt{\sigma_t^2 - \sigma_s^2\Big[\frac{\phi(\sigma_t)}{\phi(\sigma_s)}\Big]^2}\boldsymbol{z}_s}_{(c)\ Noise\ term}, \tag{10}$$

*where $\boldsymbol{z}_s \sim \mathcal{N}(\boldsymbol{0}, \boldsymbol{I})$. $\phi(x)$ is derivable and $\int \frac{1}{\sigma} + \frac{\xi(\sigma)}{2\sigma^2}d\sigma = \ln\phi(\sigma)$.*

Notably, the nonlinear term in Eq(10) involves the integration of a non-analytical neural network $\boldsymbol{x}_\theta(\boldsymbol{x}_\sigma, \sigma)$, which can be challenging to compute. For practical applicability, Proposition 3 furnishes high-stage solvers (followed by (Gonzalez et al., 2023)) for Eq.(10).

**Proposition 3** (High-stage approximations of the VE SDE, proof in Appendix A.3). *Given an initial value $\boldsymbol{x}_T$ and $M+1$ time steps $\{t_i\}_{i=0}^M$ decreasing from $t_0 = T$ to $t_M = 0$. Starting with $\tilde{\boldsymbol{x}}_{t_0} = \boldsymbol{x}_T$, the sequence $\{\tilde{\boldsymbol{x}}_{t_i}\}_{i=1}^M$ is computed iteratively as follows:*

$$\tilde{\boldsymbol{x}}_{t_i} = \frac{\phi(\sigma_{t_i})}{\phi(\sigma_{t_{i-1}})}\tilde{\boldsymbol{x}}_{t_{i-1}} + \Big[1 - \frac{\phi(\sigma_{t_i})}{\phi(\sigma_{t_{i-1}})}\Big]\boldsymbol{x}_\theta(\tilde{\boldsymbol{x}}_{\sigma_{t_{i-1}}}, \sigma_{t_{i-1}}) + \sqrt{\sigma_{t_i}^2 - \sigma_{t_{i-1}}^2\Big[\frac{\phi(\sigma_{t_i})}{\phi(\sigma_{t_{i-1}})}\Big]^2}\,\boldsymbol{z}_{t_{i-1}}$$

$$+ \sum_{n=1}^{k-1}\boldsymbol{x}_\theta^{(n)}(\tilde{\boldsymbol{x}}_{\sigma_{t_{i-1}}}, \sigma_{t_{i-1}})\Big[\frac{(\sigma_{t_i} - \sigma_{t_{i-1}})^n}{n!} + \phi(\sigma_{t_i})\int_{\sigma_{t_i}}^{\sigma_{t_{i-1}}}\frac{(\sigma - \sigma_{t_{i-1}})^{n-1}}{(n-1)!\phi(\sigma)}d\sigma\Big], \tag{11}$$

*where $k \geq 1$. $\boldsymbol{x}_\theta^{(n)}(\boldsymbol{x}_\sigma, \sigma) := \frac{d^n\boldsymbol{x}_\theta(\boldsymbol{x}_\sigma, \sigma)}{d\sigma^n}$ is the $n$-th order total derivative of $\boldsymbol{x}_\theta(\boldsymbol{x}_\sigma, \sigma)$ w.r.t $\sigma$.*

The term $\int_{\sigma_{t_i}}^{\sigma_{t_{i-1}}}\frac{(\sigma - \sigma_{t_{i-1}})^{n-1}}{(n-1)!\phi(\sigma)}d\sigma$ in Eq.(11) lacks an analytical expression, and we resort to $N$-point numerical integration for estimation. The detailed algorithms refer to Appendix B.

## 3.3 VP ER-SDE-SOLVERS

For the VP SDE, $f(t) = \frac{d\log\alpha_t}{dt}$ and $g(t) = \sqrt{\frac{d\sigma_t^2}{dt} - 2\frac{d\log\alpha_t}{dt}\sigma_t^2}$ (Kingma et al., 2021). The relationship between the score prediction model $s_\theta(x_t, t)$ and data prediction model $x_\theta(x_t, t)$ is

$-[\boldsymbol{x}_t - \alpha_t \boldsymbol{x}_\theta(\boldsymbol{x}_t, t)]/\sigma_t^2 = s_\theta(\boldsymbol{x}_t, t)$. By replacing the score function with the data prediction model, Eq.(7) can be written as:

$$d\boldsymbol{x}_t = \left\{ \left[ \frac{1}{\sigma_t}\frac{d\sigma_t}{dt} + \frac{h^2(t)}{2\sigma_t^2} \right]\boldsymbol{x}_t - \left[ \frac{1}{\sigma_t}\frac{d\sigma_t}{dt} - \frac{1}{\alpha_t}\frac{d\alpha_t}{dt} + \frac{h^2(t)}{2\sigma_t^2} \right]\alpha_t\boldsymbol{x}_\theta(\boldsymbol{x}_t, t) \right\}dt + h(t)d\overline{\boldsymbol{w}}_t. \quad (12)$$

Let $h(t) = \eta(t)\alpha_t$, $\boldsymbol{y}_t = \frac{\boldsymbol{x}_t}{\alpha_t}$ and $\lambda_t = \frac{\sigma_t}{\alpha_t}$. Denote $d\boldsymbol{w}_\lambda := \sqrt{\frac{d\lambda_t}{dt}}d\overline{\boldsymbol{w}}_t$, $\eta^2(t) = \xi(t)\frac{d\lambda_t}{dt}$, we can rewrite Eq.(12) w.r.t $\lambda$ as

$$d\boldsymbol{y}_\lambda = \left[ \frac{1}{\lambda} + \frac{\xi(\lambda)}{2\lambda^2} \right][\boldsymbol{y}_\lambda - \boldsymbol{x}_\theta(\boldsymbol{x}_\lambda, \lambda)]d\lambda + \sqrt{\xi(\lambda)}d\boldsymbol{w}_\lambda. \quad (13)$$

Following Lu et al. (2022b), we propose the exact solution for Eq.(13) using the *variation-of-constants* formula.

**Proposition 4** (Exact solution of the VP SDE, proof in Appendix A.4). *Given an initial value $\boldsymbol{x}_s$ at time $s > 0$, the solution $\boldsymbol{x}_t$ at time $t \in [0, s]$ of VP SDE in Eq.(13) is:*

$$\boldsymbol{x}_t = \underbrace{\frac{\alpha_t}{\alpha_s}\frac{\phi(\lambda_t)}{\phi(\lambda_s)}\boldsymbol{x}_s}_{(a)\ Linear\ term} + \underbrace{\alpha_t\phi(\lambda_t)\int_{\lambda_t}^{\lambda_s}\frac{\phi^{(1)}(\lambda)}{\phi^2(\lambda)}\boldsymbol{x}_\theta(\boldsymbol{x}_\lambda, \lambda)d\lambda}_{(b)\ Nonlinear\ term} + \underbrace{\alpha_t\sqrt{\lambda_t^2 - \lambda_s^2\left[\frac{\phi(\lambda_t)}{\phi(\lambda_s)}\right]^2}\boldsymbol{z}_s}_{(c)\ Noise\ term}, \quad (14)$$

*where $\boldsymbol{z}_s \sim \mathcal{N}(\boldsymbol{0}, \boldsymbol{I})$. $\phi(x)$ is derivable and $\int \frac{1}{\lambda} + \frac{\xi(\lambda)}{2\sigma^2}d\lambda = \ln\phi(\lambda)$.*

The solution of the VP SDE also involves integrating a non-analytical and nonlinear neural network. Proposition 5 furnishes high-stage solvers (followed by (Gonzalez et al., 2023)) for Eq.(14).

**Proposition 5** (High-stage approximations of the VP SDE, proof in Appendix A.5). *Given an initial value $\boldsymbol{x}_T$ and $M + 1$ time steps $\{t_i\}_{i=0}^M$ decreasing from $t_0 = T$ to $t_M = 0$. Starting with $\tilde{\boldsymbol{x}}_{t_0} = \boldsymbol{x}_T$ , the sequence $\{\tilde{\boldsymbol{x}}_{t_i}\}_{i=1}^M$ is computed iteratively as follows:*

$$\tilde{\boldsymbol{x}}_{t_i} = \frac{\alpha_{t_i}}{\alpha_{t_{i-1}}}\frac{\phi(\lambda_{t_i})}{\phi(\lambda_{t_{i-1}})}\tilde{\boldsymbol{x}}_{t_{i-1}} + \alpha_{t_i}\left[1 - \frac{\phi(\lambda_{t_i})}{\phi(\lambda_{t_{i-1}})}\right]\boldsymbol{x}_\theta(\tilde{\boldsymbol{x}}_{\lambda_{t_{i-1}}}, \lambda_{t_{i-1}}) + \alpha_{t_i}\sqrt{\lambda_{t_i}^2 - \lambda_{t_{i-1}}^2\left[\frac{\phi(\lambda_{t_i})}{\phi(\lambda_{t_{i-1}})}\right]^2}\boldsymbol{z}_{t_{i-1}}$$

$$+ \alpha_{t_i}\sum_{n=1}^{k-1}\boldsymbol{x}_\theta^{(n)}(\tilde{\boldsymbol{x}}_{\lambda_{t_{i-1}}}, \lambda_{t_{i-1}})\left[\frac{(\lambda_{t_i} - \lambda_{t_{i-1}})^n}{n!} + \phi(\lambda_{t_i})\int_{\lambda_{t_i}}^{\lambda_{t_{i-1}}}\frac{(\lambda - \lambda_{t_{i-1}})^{n-1}}{(n-1)!\phi(\lambda)}d\lambda\right],$$
$$(15)$$

*where $k \geq 1$. $\boldsymbol{x}_\theta^{(n)}(\boldsymbol{x}_\lambda, \lambda) := \frac{d^n\boldsymbol{x}_\theta(\boldsymbol{x}_\lambda, \lambda)}{d\lambda^n}$ is the $n$-th order total derivative of $\boldsymbol{x}_\theta(\boldsymbol{x}_\lambda, \lambda)$ w.r.t $\lambda$.*

Similarly, we employ $N$-point numerical integration to estimate $\int_{\lambda_{t_i}}^{\lambda_{t_{i-1}}}\frac{(\lambda - \lambda_{t_{i-1}})^{n-1}}{(n-1)!\phi(\lambda)}d\lambda$ in Eq.(15). The detailed algorithms are proposed in Appendix B.

## 4 ELUCIDATING THE SOLUTION SPACE OF ER SDE

This section primarily focuses on the solution space of the ER SDE. Specifically, in Sec.4.1, we provide a mathematical explanation for experimental observations made in previous research. Furthermore, we introduce various specialized Extended Reverse-Time SDE Solvers (ER-SDE-Solvers) in Sec.4.2, which achieve competitive rapid sampling performance compared to ODE solvers.

### 4.1 INSIGHTS ABOUT THE SOLUTION SPACE OF ER SDE

Sec.3 demonstrates that the exact solution of the ER SDE comprises three components: a linear function of the data variables, a non-linear function parameterized by neural networks and a noise term. The linear and noise terms can be precisely computed, while discretization errors are present in the non-linear term. Due to the decreasing error as the stage increases (see Table1), the first-order error predominantly influences the overall error. Therefore, we exemplify the case with order $k = 1$ for error analysis. Specifically, the first-order approximation for VE SDE is given by

$$\tilde{\boldsymbol{x}}_{t_i} = \frac{\phi(\sigma_{t_i})}{\phi(\sigma_{t_{i-1}})}\tilde{\boldsymbol{x}}_{t_{i-1}} + \left[1 - \frac{\phi(\sigma_{t_i})}{\phi(\sigma_{t_{i-1}})}\right]\boldsymbol{x}_\theta(\tilde{\boldsymbol{x}}_{\sigma_{t_{i-1}}}, \sigma_{t_{i-1}}) + \sqrt{\sigma_{t_i}^2 - \sigma_{t_{i-1}}^2\left[\frac{\phi(\sigma_{t_i})}{\phi(\sigma_{t_{i-1}})}\right]^2}\boldsymbol{z}_{t_{i-1}}, \quad (16)$$

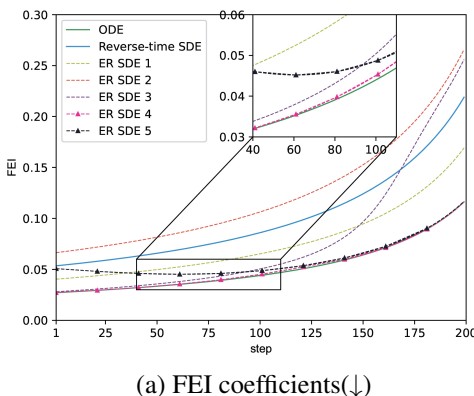 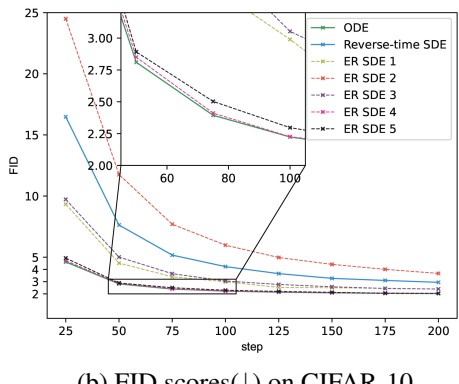

(a) FEI coefficients($\downarrow$)   (b) FID scores($\downarrow$) on CIFAR-10

Figure 3: FEI coefficients (a) and FID scores (b) for distinct noise scale functions. 1st-order solver is used here with the pretrained EDM. In the solution space of ER SDE, ODE solver shows minimal discretization error. ER SDE 4 exhibits discretization error that closely adheres to the behavior of the ODE. ER SDE 5 demonstrates elevated error in the initial 100 steps and gradually converges to the ODE's error profile. Both ER SDE 4 and ER SDE 5 exhibit comparable efficiency to the optimal ODE solver. Image quality deteriorates for ill-suited noise scale functions (like ER SDE 2).

and the first-order approximation for VP SDE is

$$\frac{\tilde{\boldsymbol{x}}_{t_i}}{\alpha_{t_i}} = \frac{\phi(\lambda_{t_i})}{\phi(\lambda_{t_{i-1}})}\frac{\tilde{\boldsymbol{x}}_{t_{i-1}}}{\alpha_{t_{i-1}}} + \left[1 - \frac{\phi(\lambda_{t_i})}{\phi(\lambda_{t_{i-1}})}\right]\boldsymbol{x}_\theta(\tilde{\boldsymbol{x}}_{\lambda_{t_{i-1}}}, \lambda_{t_{i-1}}) + \sqrt{\lambda_{t_i}^2 - \lambda_{t_{i-1}}^2\left[\frac{\phi(\lambda_{t_i})}{\phi(\lambda_{t_{i-1}})}\right]^2}\,\boldsymbol{z}_{t_{i-1}}. \quad (17)$$

We observe that the discretization errors of both VE SDE and VP SDE are influenced by the First-order Euler Integral (FEI) coefficient $1 - \frac{\phi(x_t)}{\phi(x_s)}$, which is only determined by the noise scale function $\phi(x)$ introduced in the reverse process. As $\phi(x)$ is arbitrary, different noise scale functions correspond to different solutions, collectively forming the solution space of the ER SDE (here we borrow the concept of solution space from linear algebra (Leon et al., 2006)).

**ODE Solvers outperform SDE Solvers:** Taking the first-order approximation of VE SDE as an example, an intuitive strategy for reducing discretization errors is to decrease the FEI coefficient. Due to $\frac{\phi(\sigma_t)}{\phi(\sigma_s)} \leq \frac{\sigma_t}{\sigma_s}$ (see A.7), the minimum value for the FEI coefficient is $1 - \frac{\sigma_t}{\sigma_s}$ rather than 0. Interestingly, when the FEI coefficient reaches its minimum value, the ER SDE precisely reduces to ODE (in this case, $\phi(\sigma) = \sigma$). This implies that the optimal solution of the ER SDE is exactly the ODE. Further analysis reveals that when $\phi(\sigma) = \sigma^2$, the ER SDE reduces to the reverse-time SDE. The detailed derivation can be found in Appendix A.6. As shown in Fig.3(a), the discretization error of the ODE solver is smaller than that of the SDE solver when the number of steps is the same. Fig.3(b) further illustrates that the ODE solver outperforms the SDE solver in achieving efficient sampling, consistent with Zhang & Chen (2023).

**VP SDE Solvers achieve parity with VE SDE Solvers:** The only difference between Eq.(16) and Eq.(17) lies in the latter being scaled by $1/\alpha_t$, but their relative errors remain the same. In other words, the performance of the VP SDE and the VE SDE solver is equivalent under the same number of steps and pretrained model. Directly comparing VP SDE and VE SDE solvers by experiments has been challenging in prior research due to the absence of a generative model simultaneously supporting both types of SDEs. This has led to divergent conclusions, with some studies (Song et al., 2021b) finding that VE SDE provides better sample quality than VP SDE, while others (Jolicoeur-Martineau et al., 2021) reaching the opposite conclusion. Fortunately, EDM (Karras et al., 2022) allows us for a fair comparison between VP SDE and VE SDE, as elaborated in Appendix C.3.

### 4.2 CUSTOMIZED FAST SDE SOLERS

To further demonstrate how the noise scale function $\phi(x)$ directly impacts the efficiency of the sampling process, we initially provide three different forms of $\phi(x)$:

ER SDE 1:$\phi(x) = x^{1.5}$,   ER SDE 2:$\phi(x) = x^{2.5}$,   ER SDE 3:$\phi(x) = x^{0.9}\log_{10}\left(1 + 100x^{1.5}\right)$.

Fig.3 illustrates that unfavorable choices of $\phi(x)$ (such as ER SDE 2) lead to significant discretization errors and inefficient sampling. Thus, it is crucial to carefully select the noise scale function $\phi(x)$ to achieve high-quality sampling with fewer steps. Since the optimal solution of ER SDE is ODE, our initial idea is to make the FEI coefficient as close as possible to the ODE case, i.e.,

$$\text{ER SDE 4:} \quad \phi(x) = x\left(e^{-\frac{1}{x}} + 10\right). \tag{18}$$

Since Eq.(7) involves implicit Langevin diffusion, which can effectively correct any errors in the early sampling steps (Karras et al., 2022), it is reasonable to allow for a controlled amplification of errors when the number of steps is relatively small. Consequently, we propose an alternative ER SDE solver where

$$\text{ER SDE 5:} \quad \phi(x) = x(e^{x^{0.3}} + 10). \tag{19}$$

Although ER SDE 5 exhibits more significant errors in the early stages ($\sim$100 steps) in Fig.3(a), its later-stage errors closely approach the minimum error (i.e., the error of the ODE). As a result, the image quality produced by ER SDE 5 becomes comparable to that of ER SDE 4 on the CIFAR-10 dataset (Krizhevsky et al., 2009), as shown in Fig.3(b). In fact, ER SDE 5 demonstrates superior performance on the ImageNet $64 \times 64$ dataset (Deng et al., 2009)(see Table 4). Therefore, we select ER SDE 5 as the noise scale function by default in the subsequent experiments. This strategy not only facilitates rapid sampling but also contributes to preserving the stochastic noise introduced by the reverse process, thereby enhancing higher-quality in the generated images. In fact, there are countless possible choices for $\phi(x)$, and we have only provided a few examples here. Researchers should select the suitable one based on specific application scenarios, as detailed in Appendix A.8.

## 5 EXPERIMENTS

In this section, we demonstrate that ER-SDE-Solvers can significantly accelerate the sampling process of existing pretrained DMs. We vary the number of function evaluations (NFE), i.e., the invocation number of the data prediction model, and compare the sample quality between ER-SDE-Solvers of different stages and other training-free samplers. For each experiment, we draw 50K samples and employ the widely-used FID score (Heusel et al., 2017) to evaluate sample quality, where a lower FID typically signifies better sample quality. For detailed implementation and experimental settings, please refer to Appendix C.

### 5.1 DIFFERENT STAGES OF VE ER-SDE-SOLVERS AND VP ER-SDE-SOLVERS

To ensure a fair comparison between VP ER-SDE-Solvers and VE ER-SDE-Solvers, we opt for EDM (Karras et al., 2022) as the pretrained model, as detailed in Appendix C.3. It can be observed from Table 1 that the image generation quality produced by both of them is similar, consistent with the findings in Sec.4.1. Additionally, the high-stage ER-SDE-Solver-3 converges faster than ER-SDE-Solver-2, particularly in the few-step regime under 20 NFE. This is because higher stages result in more minor discretization errors. We also arrive at the same conclusions on the CIFAR-10 dataset (Krizhevsky et al., 2009) as can be found in Table 6.

Table 1: Sample quality measured by FID↓ on ImageNet $64 \times 64$ for different stages of VE(P) ER-SDE-Solvers with EDM, varying the NFE. VE(P)-x denotes the x-th stage VE(P) ER-SDE-Solver.

| Method\NFE | 10 | 20 | 30 | 50 | Method\NFE | 10 | 20 | 30 | 50 |
|---|---|---|---|---|---|---|---|---|---|
| VE-2 | 11.81 | 3.67 | 2.67 | 2.31 | VE-3 | 11.46 | 3.45 | 2.58 | 2.24 |
| VP-2 | 11.94 | 3.73 | 2.67 | 2.27 | VP-3 | 11.32 | 3.48 | 2.58 | 2.28 |

### 5.2 COMPARISON WITH OTHER TRAINING-FREE METHODS

We compare ER-SDE-Solvers with other training-free sampling methods, including stochastic samplers such as SDE-DPM-Solver++ (Lu et al., 2022b), as well as deterministic samplers like DDIM (Song et al., 2021a), DPM-Solver (Lu et al., 2022a) and DPM-Solver++ (Lu et al., 2022b). Table 2 presents experimental results on the ImageNet $64 \times 64$ dataset (Deng et al., 2009) using the same pretrained model EDM (Karras et al., 2022). It is evident that ER-SDE-Solvers emerge

Table 2: Sample quality measured by FID↓ on ImageNet $64 \times 64$ with the pretrained model EDM, varying the NFE. The upper right $-$ indicates a reduction of NFE by one.

|  | Sampling method\NFE | 10 | 20 | 30 | 50 |
|---|---|---|---|---|---|
| Stochastic Sampling | DDIM($\eta = 1$) (Song et al., 2021a) | 49.28 | 23.32 | 13.73 | 7.35 |
|  | SDE-DPM-Solver++(2M) (Lu et al., 2022b) | 21.30 | 6.36 | 3.61 | 2.29 |
|  | EDM-Stochastic (Karras et al., 2022) | $57.47^-$ | $6.17^-$ | $3.46^-$ | $2.49^-$ |
|  | Ours(ER-SDE-Solver-3) | **11.32** | **3.45** | **2.58** | **2.24** |
| Deterministic Sampling | DDIM (Song et al., 2021a) | 17.33 | 6.45 | 4.33 | 3.15 |
|  | EDM-Deterministic (Karras et al., 2022) | $35.59^-$ | $3.56^-$ | $2.60^-$ | $\mathbf{2.34}^-$ |
|  | DPM-Solver-3 (Lu et al., 2022a) | 6.88 | 3.02 | **2.56** | 2.35 |
|  | DPM-Solver++(2M) (Lu et al., 2022b) | **6.48** | **2.98** | **2.56** | 2.35 |

Table 3: Sample quality measured by FID↓ on class-conditional ImageNet $256 \times 256$ with the pretrained model Guided-diffusion (classifier guidance scale = 2.0, linear noise schedule), varying the NFE.

|  | Sampling method\NFE | 10 | 20 | 30 | 50 |
|---|---|---|---|---|---|
| Stochastic Sampling | DDIM($\eta = 1$) (Song et al., 2021a) | 17.97 | 10.23 | 8.19 | 6.85 |
|  | SDE-DPM-Solver++(2M) (Lu et al., 2022b) | 9.21 | 6.01 | 5.47 | 5.19 |
|  | Ours(ER-SDE-Solver-3) | **6.24** | **4.76** | **4.62** | **4.57** |
| Deterministic Sampling | DDIM (Song et al., 2021a) | 8.63 | 5.60 | 5.00 | **4.59** |
|  | DPM-Solver-3 (Lu et al., 2022a) | **6.45** | **5.03** | **4.94** | 4.92 |
|  | DPM-Solver++(2M) (Lu et al., 2022b) | 7.19 | 5.54 | 5.32 | 5.16 |

as the most efficient stochastic samplers, achieving a remarkable $2 \sim 8\times$ speedup compared with previously state-of-the-art stochastic sampling methods, particularly when the NFE is limited. It is not surprising that image quality generated by ER-SDE-Solvers within the initial 20 NFE is relatively inferior to some deterministic samplers such as DPM-Solver, consistent with the theoretical justification in Sec.4.1. However, our stochastic sampler leverages the noise introduced during the reverse process to rectify errors present in earlier steps. Notably, the FID drops to 2.24 when NFE is 50, surpassing even the deterministic samplers.

Particularly, we combine ER-SDE-Solvers with *classifier guidance* to generate high-resolution images. Table 3 provides comparative results on ImageNet $256 \times 256$ (Deng et al., 2009) using Guided-diffusion (Dhariwal & Nichol, 2021) as the pretrained model. We surprisingly find that *classifier guidance* significantly improves the image generation quality of ER-SDE-Solvers even with very low NFE compared to ODE Solvers. This may be attributed to the customized noise injected into the sampling process, which mitigates the inaccuracies in data estimation introduced by classifier gradient guidance. Further investigation into the specific reasons is left for future work.

Moreover, the images produced by ER-SDE-Solvers exhibit greater variability compared to deterministic samplers, as illustrated in Appendix D. We also provide comparisons on various datasets using different pretrained models in Appendix D.

## 6 CONCLUSION

We address the challenges of fast and training-free sampling in DMs. Initially, we formulate the sampling problem as an ER SDE, which unifies ODEs and SDEs in previous studies. Leveraging the semi-linear structure of ER SDE solutions, we provide exact solutions and high-stage approximations for both VP SDE and VE SDE. Building upon it, we establish two crucial findings from a mathematical standpoint: the superior performance of ODE solvers for rapid sampling over SDE solvers, and the comparable performance of VP SDE solvers with VE SDE solvers. Finally, we introduce state-of-the-art stochastic fast samplers, ER-SDE-Solvers, by adeptly selecting noise scale functions for the sampling process.

REPRODUCIBILITY STATEMENT

For the theoretical findings of the proposed diffusion equation ER SDE and its solutions, comprehensive explanations and complete proofs are provided in Appendix A. For the algorithms of ER Solvers, pseudocodes are presented in Appendix B, while detailed experimental setups can be found in Appendix C. Our code utilizes pretrained checkpoints from https://github.com/NVlabs/edm and https://github.com/openai/guided-diffusion. Detailed code is made available in the supplementary material. As for the datasets employed in our experiments, CIFAR-10 (Krizhevsky et al., 2009), FFHQ (Karras et al., 2019), ImageNet (Deng et al., 2009) and LSUN (Yu et al., 2015) are publicly accessible datasets, ensuring transparency and reproducibility in our research.

ETHICS STATEMENT

In line with other advanced deep generative models like GANs, DMs can be harnessed to produce deceptive or misleading content, particularly in manipulated images. The efficient solvers we propose herein offer the capability to expedite the sampling process of DMs, thereby enabling faster image generation and manipulation, potentially leading to the creation of convincing but fabricated visuals. As with any technology, this acceleration could accentuate the potential ethical concerns associated with DMs, particularly their susceptibility to misuse or malicious applications. For instance, more frequent image generation might elevate the likelihood of unauthorized exposure of personal information, facilitate content forgery and dissemination of false information, and potentially infringe upon intellectual property rights.

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

## A ADDITIONAL PROOFS

### A.1 PROOF OF PROPOSITION 1

In this section, we provide the derivation process of Eq.(7), with the key insight being that the forward and backward SDEs share the same marginal distribution.

We begin by considering the forward process. As outlined in Sec.2.1, the forward process can be expressed as the SDE shown in Eq.(1). In accordance with the Fokker-Plank Equation (also known as the Forward Kolmogorov Equation) (Risken & Risken, 1996), we obtain:

$$
\begin{aligned}
\frac{\partial p_t(\boldsymbol{x}_t)}{\partial t} &= -\nabla_{\boldsymbol{x}}[f(t)\boldsymbol{x}_t p_t(\boldsymbol{x}_t)] + \frac{\partial}{\partial x_i \partial x_j}\Big[\frac{1}{2}g^2(t)p_t(\boldsymbol{x}_t)\Big] \\
&= -\nabla_{\boldsymbol{x}}[f(t)\boldsymbol{x}_t p_t(\boldsymbol{x}_t)] + \nabla_{\boldsymbol{x}}\Big[\frac{1}{2}g^2(t)\nabla_{\boldsymbol{x}}p_t(\boldsymbol{x}_t)\Big] \\
&= -\nabla_{\boldsymbol{x}}[f(t)\boldsymbol{x}_t p_t(\boldsymbol{x}_t)] + \nabla_{\boldsymbol{x}}\Big[\frac{1}{2}g^2(t)p_t(\boldsymbol{x}_t)\nabla_{\boldsymbol{x}}\log p_t(\boldsymbol{x}_t)\Big] \\
&= -\nabla_{\boldsymbol{x}}\Big\{\Big[f(t)\boldsymbol{x}_t - \frac{1}{2}g^2(t)\nabla_{\boldsymbol{x}}\log p_t(\boldsymbol{x}_t)\Big]p_t(\boldsymbol{x}_t)\Big\},
\end{aligned}
\tag{20}
$$

where $p(\boldsymbol{x}_t)$ denotes the probability density function of state $\boldsymbol{x}_t$.

Most processes defined by a forward-time or conventional diffusion equation model possess a corresponding reverse-time model (Song et al., 2021b; Anderson, 1982), which can be formulated as:

$$
d\boldsymbol{x}_t = \mu(t, \boldsymbol{x}_t)dt + \sigma(t, \boldsymbol{x}_t)d\overline{\boldsymbol{w}}_t.
\tag{21}
$$

According to the Backward Kolmogorov Equation (Risken & Risken, 1996), we have:

$$
\begin{aligned}
\frac{\partial p_t(\boldsymbol{x}_t)}{\partial t} &= -\nabla_{\boldsymbol{x}}[\mu(t, \boldsymbol{x}_t)p_t(\boldsymbol{x}_t)] - \frac{\partial}{\partial x_i \partial x_j}\Big[\frac{1}{2}\sigma^2(t, \boldsymbol{x}_t)p_t(\boldsymbol{x}_t)\Big] \\
&= -\nabla_{\boldsymbol{x}}[\mu(t, \boldsymbol{x}_t)p_t(\boldsymbol{x}_t)] - \nabla_{\boldsymbol{x}}\Big[\frac{1}{2}\sigma^2(t, \boldsymbol{x}_t)\nabla_{\boldsymbol{x}}p_t(\boldsymbol{x}_t)\Big] \\
&= -\nabla_{\boldsymbol{x}}[\mu(t, \boldsymbol{x}_t)p(\boldsymbol{x}_t)] - \nabla_{\boldsymbol{x}}\Big[\frac{1}{2}\sigma^2(t, \boldsymbol{x}_t)p(\boldsymbol{x}_t)\nabla\log p_t(\boldsymbol{x}_t)\Big] \\
&= -\nabla_{\boldsymbol{x}}\Big\{\Big[\mu(t, \boldsymbol{x}_t) + \frac{1}{2}\sigma^2(t, x_t)\nabla_{\boldsymbol{x}}\log p_t(\boldsymbol{x}_t)\Big]p_t(\boldsymbol{x}_t)\Big\}.
\end{aligned}
\tag{22}
$$

We aim for the forward process and the backward process to share the same distribution, namely:

$$
\begin{aligned}
\mu(t, \boldsymbol{x}_t) + \frac{1}{2}\sigma^2(t, \boldsymbol{x}_t)\nabla_{\boldsymbol{x}}\log p_t(\boldsymbol{x}_t) &= f(t)\boldsymbol{x}_t - \frac{1}{2}g^2(t)\nabla_{\boldsymbol{x}}\log p_t(\boldsymbol{x}_t) \\
\mu(t, \boldsymbol{x}_t) &= f(t)\boldsymbol{x}_t - \frac{g^2(t) + \sigma^2(t, \boldsymbol{x}_t)}{2}\nabla_{\boldsymbol{x}}\log p(\boldsymbol{x}_t).
\end{aligned}
\tag{23}
$$

Let $\sigma(t, \boldsymbol{x}_t) = h(t)$, yielding the Extended Reverse-Time SDE (ER-SDE):

$$
d\boldsymbol{x}_t = \Big[f(t)\boldsymbol{x}_t - \frac{g^2(t) + h^2(t)}{2}\nabla_{\boldsymbol{x}}\log p_t(\boldsymbol{x}_t)\Big]dt + h(t)d\overline{\boldsymbol{w}}_t.
\tag{24}
$$

## A.2 PROOF OF PROPOSITION 2

Eq.(9) has the following analytical solution (Kloeden & Platen, 1992a):

$$\boldsymbol{x}_t = e^{\int_{\sigma_s}^{\sigma_t} \frac{1}{\sigma} + \frac{\xi(\sigma)}{2\sigma^2} d\sigma} \boldsymbol{x}_s - \int_{\sigma_s}^{\sigma_t} e^{\int_{\sigma}^{\sigma_t} \frac{1}{\tau} + \frac{\xi(\tau)}{2\tau^2} d\tau} \Big[\frac{1}{\sigma} + \frac{\xi(\sigma)}{2\sigma^2}\Big] \boldsymbol{x}_\theta(\boldsymbol{x}_\sigma, \sigma) d\sigma + \int_{\sigma_s}^{\sigma_t} e^{\int_{\sigma}^{\sigma_t} \frac{1}{\tau} + \frac{\xi(\tau)}{2\tau^2} d\tau} \sqrt{\xi(\sigma)} d\boldsymbol{w}_\sigma.$$

(25)

Let $\int \frac{1}{\sigma} + \frac{\xi(\sigma)}{2\sigma^2} d\sigma = \ln \phi(\sigma)$ and suppose $\phi(x)$ is derivable, then

$$\frac{1}{\sigma} + \frac{\xi(\sigma)}{2\sigma^2} = \frac{\phi^{(1)}(\sigma)}{\phi(\sigma)},$$

(26)

where $\phi^{(1)}(x)$ is the first derivative of $\phi(x)$.

Substituting Eq.(26) into Eq.(25), we obtain

$$\begin{aligned}
\boldsymbol{x}_t &= \frac{\phi(\sigma_t)}{\phi(\sigma_s)}\boldsymbol{x}_s - \int_{\sigma_s}^{\sigma_t} \frac{\phi(\sigma_t)}{\phi(\sigma)} \frac{\phi^{(1)}(\sigma)}{\phi(\sigma)} \boldsymbol{x}_\theta(\boldsymbol{x}_\sigma, \sigma) d\sigma + \int_{\sigma_s}^{\sigma_t} \frac{\phi(\sigma_t)}{\phi(\sigma)} \sqrt{\xi(\sigma)} d\boldsymbol{w}_\sigma \\
&= \underbrace{\frac{\phi(\sigma_t)}{\phi(\sigma_s)}\boldsymbol{x}_s}_{\text{(a) Linear term}} + \underbrace{\phi(\sigma_t) \int_{\sigma_t}^{\sigma_s} \frac{\phi^{(1)}(\sigma)}{\phi^2(\sigma)} \boldsymbol{x}_\theta(\boldsymbol{x}_\sigma, \sigma) d\sigma}_{\text{(b) Nonlinear term}} + \underbrace{\int_{\sigma_s}^{\sigma_t} \frac{\phi(\sigma_t)}{\phi(\sigma)} \sqrt{\xi(\sigma)} d\boldsymbol{w}_\sigma}_{\text{(c) Noise term}}.
\end{aligned}$$

(27)

Inspired by Lu et al. (2022b), the noise term $(c)$ can be computed as follows:

$$(c) = -\phi(\sigma_t) \int_{\sigma_t}^{\sigma_s} \frac{\sqrt{\xi(\sigma)}}{\phi(\sigma)} d\boldsymbol{w}_\sigma = -\phi(\sigma_t) \sqrt{\int_{\sigma_t}^{\sigma_s} \frac{\xi(\sigma)}{\phi^2(\sigma)} d\sigma} \boldsymbol{z}_s \,,\, \boldsymbol{z}_s \sim \mathcal{N}(\boldsymbol{0}, \boldsymbol{I}).$$

(28)

Substituting Eq.(26) into Eq.(28), we have

$$\begin{aligned}
(c) &= -\phi(\sigma_t) \sqrt{\int_{\sigma_t}^{\sigma_s} \Big[2\sigma^2 \frac{\phi^{(1)}(\sigma)}{\phi^3(\sigma)} - 2\sigma \frac{1}{\phi^2(\sigma)}\Big] d\sigma} \boldsymbol{z}_s \\
&= -\phi(\sigma_t) \sqrt{-\frac{\sigma^2}{\phi^2(\sigma)}\Big|_{\sigma_t}^{\sigma_s}} \boldsymbol{z}_s \\
&= -\sqrt{\sigma_t^2 - \sigma_s^2 \Big[\frac{\phi(\sigma_t)}{\phi(\sigma_s)}\Big]^2} \boldsymbol{z}_s.
\end{aligned}$$

(29)

Considering that adding Gaussian noise is equivalent to subtracting Gaussian noise, we can rewrite Eq.(27) as

$$\boldsymbol{x}_t = \frac{\phi(\sigma_t)}{\phi(\sigma_s)}\boldsymbol{x}_s + \phi(\sigma_t) \int_{\sigma_t}^{\sigma_s} \frac{\phi^{(1)}(\sigma)}{\phi^2(\sigma)} \boldsymbol{x}_\theta(\boldsymbol{x}_\sigma, \sigma) d\sigma + \sqrt{\sigma_t^2 - \sigma_s^2 \Big[\frac{\phi(\sigma_t)}{\phi(\sigma_s)}\Big]^2} \boldsymbol{z}_s.$$

(30)

## A.3 PROOF OF PROPOSITION 3

In order to minimize the approximation error between $\tilde{\boldsymbol{x}}_M$ and the true solution at time $0$, it is essential to progressively reduce the approximation error for each $\tilde{\boldsymbol{x}}_{t_i}$ during each step. Beginning from the preceding value $\tilde{\boldsymbol{x}}_{t_{i-1}}$ at the time $t_{i-1}$, as outlined in Eq.(30), the exact solution $\boldsymbol{x}_{t_i}$ at time $t_i$ can be determined as follows:

$$\boldsymbol{x}_{t_i} = \frac{\phi(\sigma_{t_i})}{\phi(\sigma_{t_{i-1}})}\tilde{\boldsymbol{x}}_{t_{i-1}} + \phi(\sigma_{t_i}) \int_{\sigma_{t_i}}^{\sigma_{t_{i-1}}} \frac{\phi^{(1)}(\sigma)}{\phi^2(\sigma)} \boldsymbol{x}_\theta(\boldsymbol{x}_\sigma, \sigma) d\sigma + \sqrt{\sigma_{t_i}^2 - \sigma_{t_{i-1}}^2 \Big[\frac{\phi(\sigma_{t_i})}{\phi(\sigma_{t_{i-1}})}\Big]^2} \boldsymbol{z}_{t_{i-1}}.$$

(31)

Inspired by Lu et al. (2022a), we approximate the integral of $\boldsymbol{x}_\theta$ from $\sigma_{t_{i-1}}$ to $\sigma_{t_i}$ in order to compute $\tilde{\boldsymbol{x}}_{t_i}$ for approximating $\boldsymbol{x}_{t_i}$. Denote $\boldsymbol{x}_\theta^{(n)}(\boldsymbol{x}_\sigma, \sigma) := \frac{d^n \boldsymbol{x}_\theta(\boldsymbol{x}_\sigma, \sigma)}{d\sigma^n}$ as the $n$-th order total derivative of

$\boldsymbol{x}_\theta(\boldsymbol{x}_\sigma, \sigma)$ w.r.t $\sigma$. For $k \geq 1$, the $k-1$-th order Itô-Taylor expansion of $\boldsymbol{x}_\theta(\boldsymbol{x}_\sigma, \sigma)$ w.r.t $\sigma$ at $\sigma_{t_{i-1}}$ is (Gonzalez et al., 2023)

$$\boldsymbol{x}_\theta(\boldsymbol{x}_\sigma, \sigma) = \sum_{n=0}^{k-1} \frac{(\sigma - \sigma_{t_{i-1}})^n}{n!} \boldsymbol{x}^{(n)}(\boldsymbol{x}_{\sigma_{t_{i-1}}}, \sigma_{t_{i-1}}) + \mathcal{R}_k, \tag{32}$$

where the residual $\mathcal{R}_k$ comprises of deterministic iterated integrals of length greater than $k$ and all iterated integrals with at least one stochastic component.

Substituting the above Itô-Taylor expansion into Eq.(31), we obtain

$$\begin{aligned}
\boldsymbol{x}_{t_i} &= \frac{\phi(\sigma_{t_i})}{\phi(\sigma_{t_{i-1}})} \boldsymbol{x}_{t_{i-1}} + \phi(\sigma_{t_i}) \sum_{n=0}^{k-1} \boldsymbol{x}^{(n)}(\boldsymbol{x}_{\sigma_{t_{i-1}}}, \sigma_{t_{i-1}}) \int_{\sigma_{t_i}}^{\sigma_{t_{i-1}}} \frac{\phi^{(1)}(\sigma)}{\phi^2(\sigma)} \frac{(\sigma - \sigma_{t_{i-1}})^n}{n!} d\sigma \\
&+ \sqrt{\sigma_{t_i}^2 - \sigma_{t_{i-1}}^2 \left[\frac{\phi(\sigma_{t_i})}{\phi(\sigma_{t_{i-1}})}\right]^2} \boldsymbol{z}_{t_{i-1}} + \tilde{\mathcal{R}}_k,
\end{aligned} \tag{33}$$

where $\tilde{\mathcal{R}}_k$ can be easily obtained from $\mathcal{R}_k$ and $\phi(\sigma_{t_i}) \int_{\sigma_{t_i}}^{\sigma_{t_{i-1}}} \frac{\phi^{(1)}(\sigma)}{\phi^2(\sigma)} d\sigma$.

When $n \geq 1$,

$$\begin{aligned}
\int_{\sigma_{t_i}}^{\sigma_{t_{i-1}}} \frac{\phi^{(1)}(\sigma)}{\phi^2(\sigma)} \frac{(\sigma - \sigma_{t_{i-1}})^n}{n!} d\sigma &= \int_{\sigma_{t_i}}^{\sigma_{t_{i-1}}} \frac{(\sigma - \sigma_{t_{i-1}})^n}{n!} d\left[-\frac{1}{\phi(\sigma)}\right] \\
&= -\frac{(\sigma - \sigma_{t_{i-1}})^n}{n!} \frac{1}{\phi(\sigma)}\Big|_{\sigma_{t_i}}^{\sigma_{t_{i-1}}} - \int_{\sigma_{t_i}}^{\sigma_{t_{i-1}}} -\frac{1}{\phi(\sigma)} d\left[\frac{(\sigma - \sigma_{t_{i-1}})^n}{n!}\right] \\
&= \frac{(\sigma_{t_i} - \sigma_{t_{i-1}})^n}{n!\phi(\sigma_{t_i})} + \int_{\sigma_{t_i}}^{\sigma_{t_{i-1}}} \frac{(\sigma - \sigma_{t_{i-1}})^{n-1}}{(n-1)!\phi(\sigma)} d\sigma.
\end{aligned} \tag{34}$$

Thus, by dropping the $\tilde{\mathcal{R}}_k$ contribution as in (Gonzalez et al., 2023), we have

$$\begin{aligned}
\tilde{\boldsymbol{x}}_{t_i} &= \frac{\phi(\sigma_{t_i})}{\phi(\sigma_{t_{i-1}})} \tilde{\boldsymbol{x}}_{t_{i-1}} + \left[1 - \frac{\phi(\sigma_{t_i})}{\phi(\sigma_{t_{i-1}})}\right] \boldsymbol{x}_\theta(\tilde{\boldsymbol{x}}_{\sigma_{t_{i-1}}}, \sigma_{t_{i-1}}) + \sqrt{\sigma_{t_i}^2 - \sigma_{t_{i-1}}^2 \left[\frac{\phi(\sigma_{t_i})}{\phi(\sigma_{t_{i-1}})}\right]^2} \boldsymbol{z}_{t_{i-1}} \\
&+ \sum_{n=1}^{k-1} \boldsymbol{x}_\theta^{(n)}(\tilde{\boldsymbol{x}}_{\sigma_{t_{i-1}}}, \sigma_{t_{i-1}}) \left[\frac{(\sigma_{t_i} - \sigma_{t_{i-1}})^n}{n!} + \phi(\sigma_{t_i}) \int_{\sigma_{t_i}}^{\sigma_{t_{i-1}}} \frac{(\sigma - \sigma_{t_{i-1}})^{n-1}}{(n-1)!\phi(\sigma)} d\sigma\right],
\end{aligned} \tag{35}$$

where $k \geq 2$. Notably, when $k = 1$, the summation term in Eq.(35) with the upper index smaller than the lower index can be defined as an empty sum, and its value is 0 (Treeby & Cox, 2014). In conclusion, Eq.(35) holds when $k \geq 1$.

**Note:** The proposed algorithm has a global error of at least $\mathcal{O}((\sigma_{t_i} - \sigma_{t_{i-1}}))$ (Gonzalez et al., 2023). Therefore, when $k = 1$, we refer to it as a first-order solver with a strong convergence guarantee. When $k \geq 2$, we designate it as a $k$th-stage solver in accordance with the statement in Gonzalez et al. (2023).

## A.4  PROOF OF PROPOSITION 4

Eq.(13) has the following analytical solution (Kloeden & Platen, 1992a):

$$\boldsymbol{y}_t = e^{\int_{\lambda_s}^{\lambda_t} \frac{1}{\lambda} + \frac{\xi(\lambda)}{2\lambda^2} d\lambda} \boldsymbol{y}_s - \int_{\lambda_s}^{\lambda_t} e^{\int_\lambda^{\lambda_t} \frac{1}{\tau} + \frac{\xi(\tau)}{2\tau^2} d\tau} \left[\frac{1}{\lambda} + \frac{\xi(\lambda)}{2\lambda^2}\right] \boldsymbol{x}_\theta(\boldsymbol{x}_\lambda, \lambda) d\lambda + \int_{\lambda_s}^{\lambda_t} e^{\int_\lambda^{\lambda_t} \frac{1}{\tau} + \frac{\xi(\tau)}{2\tau^2} d\tau} \sqrt{\xi(\lambda)} d\boldsymbol{w}_\lambda. \tag{36}$$

Similar to VE SDE, let $\int \frac{1}{\lambda} + \frac{\xi(\lambda)}{2\lambda^2} d\lambda = \ln\phi(\lambda)$. Substituting $\boldsymbol{y}_t = \frac{\boldsymbol{x}_t}{\alpha_t}$, we have

$$\boldsymbol{x}_t = \frac{\alpha_t}{\alpha_s} \frac{\phi(\lambda_t)}{\phi(\lambda_s)} \boldsymbol{x}_s + \alpha_t \phi(\lambda_t) \int_{\lambda_t}^{\lambda_s} \frac{\phi^{(1)}(\lambda)}{\phi^2(\lambda)} \boldsymbol{x}_\theta(\boldsymbol{x}_\lambda, \lambda) d\lambda + \alpha_t \int_{\lambda_s}^{\lambda_t} \frac{\phi(\lambda_t)}{\phi(\lambda)} \sqrt{\xi(\lambda)} d\boldsymbol{w}_\lambda. \tag{37}$$

Similarly, the noise term can be computed as follows:

$$
\begin{aligned}
\alpha_t \int_{\lambda_s}^{\lambda_t} \frac{\phi(\lambda_t)}{\phi(\lambda)} \sqrt{\xi(\lambda)} d\boldsymbol{w}_\lambda &= -\alpha_t \phi(\lambda_t) \int_{\lambda_t}^{\lambda_s} \frac{\sqrt{\xi(\lambda)}}{\phi(\lambda)} d\boldsymbol{w}_\lambda \\
&= -\alpha_t \phi(\lambda_t) \sqrt{\int_{\lambda_t}^{\lambda_s} \frac{\xi(\lambda)}{\phi^2(\lambda)} d\lambda} \boldsymbol{z}_s \\
&= -\alpha_t \phi(\lambda_t) \sqrt{\int_{\lambda_t}^{\lambda_s} \left[ 2\lambda^2 \frac{\phi^{(1)}(\lambda)}{\phi^3(\lambda)} - 2\lambda \frac{1}{\phi^2(\lambda)} \right] d\lambda} \boldsymbol{z}_s \\
&= -\alpha_t \phi(\lambda_t) \sqrt{-\frac{\lambda^2}{\phi^2(\lambda)} \Big|_{\lambda_t}^{\lambda_s}} \boldsymbol{z}_s \\
&= -\alpha_t \sqrt{\lambda_t^2 - \lambda_s^2 \left[ \frac{\phi(\lambda_t)}{\phi(\lambda_s)} \right]^2} \boldsymbol{z}_s.
\end{aligned}
\tag{38}
$$

Considering that adding Gaussian noise is equivalent to subtracting Gaussian noise. Above all, we have exact solution

$$
\boldsymbol{x}_t = \frac{\alpha_t}{\alpha_s} \frac{\phi(\lambda_t)}{\phi(\lambda_s)} \boldsymbol{x}_s + \alpha_t \phi(\lambda_t) \int_{\lambda_t}^{\lambda_s} \frac{\phi^{(1)}(\lambda)}{\phi^2(\lambda)} \boldsymbol{x}_\theta(\boldsymbol{x}_\lambda, \lambda) d\lambda + \alpha_t \sqrt{\lambda_t^2 - \lambda_s^2 \left[ \frac{\phi(\lambda_t)}{\phi(\lambda_s)} \right]^2} \boldsymbol{z}_s.
\tag{39}
$$

## A.5 PROOF OF PROPOSITION 5

Similarly, to diminish the approximation error between $\tilde{\boldsymbol{x}}_M$ and the true solution at time $0$, it is necessary to iteratively decrease the approximation error for each $\tilde{\boldsymbol{x}}_{t_i}$ at each step. Starting from the preceding value $\tilde{\boldsymbol{x}}_{t_{i-1}}$ at the time $t_{i-1}$, following Eq.(39), the precise solution $\boldsymbol{x}_{t_i}$ at time $t_i$ is derived as:

$$
\begin{aligned}
\boldsymbol{x}_{t_i} = {} & \frac{\alpha_{t_i}}{\alpha_{t_{i-1}}} \frac{\phi(\lambda_{t_i})}{\phi(\lambda_{t_{i-1}})} \boldsymbol{x}_{t_{i-1}} + \alpha_{t_i} \phi(\lambda_{t_i}) \int_{\lambda_{t_i}}^{\lambda_{t_{i-1}}} \frac{\phi^{(1)}(\lambda)}{\phi^2(\lambda)} \boldsymbol{x}_\theta(\boldsymbol{x}_\lambda, \lambda) d\lambda \\
& + \alpha_{t_i} \sqrt{\lambda_{t_i}^2 - \lambda_{t_{i-1}}^2 \left[ \frac{\phi(\lambda_{t_i})}{\phi(\lambda_s)} \right]^2} \boldsymbol{z}_{t_{i-1}}.
\end{aligned}
\tag{40}
$$

We also approximate the integral of $\boldsymbol{x}_\theta$ from $\sigma_{t_{i-1}}$ to $\sigma_{t_i}$ to compute $\tilde{\boldsymbol{x}}_{t_i}$ for approximating $\boldsymbol{x}_{t_i}$ (Lu et al., 2022a). Denote $\boldsymbol{x}_\theta^{(n)}(\boldsymbol{x}_\lambda, \lambda) := \frac{d^n \boldsymbol{x}_\theta(\boldsymbol{x}_\lambda, \lambda)}{d\lambda^n}$ as the $n$-th order total derivative of $\boldsymbol{x}_\theta(\boldsymbol{x}_\lambda, \lambda)$ w.r.t $\lambda$. For $k \geq 1$, the $k-1$-th order Itô-Taylor expansion of $\boldsymbol{x}_\theta(\boldsymbol{x}_\lambda, \lambda)$ w.r.t $\lambda$ at $\lambda_{t_{i-1}}$ is (Gonzalez et al., 2023)

$$
\boldsymbol{x}_\theta(\boldsymbol{x}_\lambda, \lambda) = \sum_{n=0}^{k-1} \frac{(\lambda - \lambda_{t_{i-1}})^n}{n!} \boldsymbol{x}^{(n)}(\boldsymbol{x}_{\lambda_{t_{i-1}}}, \lambda_{t_{i-1}}) + \mathcal{R}_k,
\tag{41}
$$

where the residual $\mathcal{R}_k$ comprises of deterministic iterated integrals of length greater than $k$ and all iterated integrals with at least one stochastic component.

Substituting the above Itô-Taylor expansion into Eq.(40), we obtain

$$
\begin{aligned}
\boldsymbol{x}_{t_i} = {} & \frac{\alpha_{t_i}}{\alpha_{t_{i-1}}} \frac{\phi(\lambda_{t_i})}{\phi(\lambda_{t_{i-1}})} \boldsymbol{x}_{t_{i-1}} + \alpha_{t_i} \phi(\lambda_{t_i}) \sum_{n=0}^{k-1} \boldsymbol{x}^{(n)}(\boldsymbol{x}_{\lambda_{t_{i-1}}}, \lambda_{t_{i-1}}) \int_{\lambda_{t_i}}^{\lambda_{t_{i-1}}} \frac{\phi^{(1)}(\lambda)}{\phi^2(\lambda)} \frac{(\lambda - \lambda_{t_{i-1}})^n}{n!} d\lambda \\
& + \alpha_{t_i} \sqrt{\lambda_{t_i}^2 - \lambda_{t_{i-1}}^2 \left[ \frac{\phi(\lambda_{t_i})}{\phi(\lambda_{t_{i-1}})} \right]^2} \boldsymbol{z}_{t_{i-1}} + \tilde{\mathcal{R}}_k,
\end{aligned}
\tag{42}
$$

where $\tilde{\mathcal{R}}_k$ can be easily obtained from $\mathcal{R}_k$ and $\alpha_{t_i} \phi(\lambda_{t_i}) \int_{\lambda_{t_i}}^{\lambda_{t_{i-1}}} \frac{\phi^{(1)}(\lambda)}{\phi^2(\lambda)} d\lambda$.

When $n \geq 1$,

$$
\begin{aligned}
\int_{\lambda_{t_i}}^{\lambda_{t_{i-1}}} \frac{\phi^{(1)}(\lambda)}{\phi^2(\lambda)} \frac{(\lambda - \lambda_{t_{i-1}})^n}{n!} d\lambda &= \int_{\lambda_{t_i}}^{\lambda_{t_{i-1}}} \frac{(\lambda - \lambda_{t_{i-1}})^n}{n!} d\Big[-\frac{1}{\phi(\lambda)}\Big] \\
&= -\frac{(\lambda - \lambda_{t_{i-1}})^n}{n!} \frac{1}{\phi(\lambda)}\Big|_{\lambda_{t_i}}^{\lambda_{t_{i-1}}} - \int_{\lambda_{t_i}}^{\lambda_{t_{i-1}}} -\frac{1}{\phi(\lambda)} d\Big[\frac{(\lambda - \lambda_{t_{i-1}})^n}{n!}\Big] \\
&= \frac{(\lambda_{t_i} - \lambda_{t_{i-1}})^n}{n!\phi(\lambda_{t_i})} + \int_{\lambda_{t_i}}^{\lambda_{t_{i-1}}} \frac{(\lambda - \lambda_{t_{i-1}})^{n-1}}{(n-1)!\phi(\lambda)} d\lambda.
\end{aligned}
\tag{43}
$$

Thus, by dropping the $\tilde{\mathcal{R}}_k$ contribution as in (Gonzalez et al., 2023), we have

$$
\begin{aligned}
\tilde{\boldsymbol{x}}_{t_i} = &\frac{\alpha_{t_i}}{\alpha_{t_{i-1}}} \frac{\phi(\lambda_{t_i})}{\phi(\lambda_{t_{i-1}})} \tilde{\boldsymbol{x}}_{t_{i-1}} + \alpha_{t_i}\Big[1 - \frac{\phi(\lambda_{t_i})}{\phi(\lambda_{t_{i-1}})}\Big] \boldsymbol{x}_\theta(\tilde{\boldsymbol{x}}_{\lambda_{t_{i-1}}}, \lambda_{t_{i-1}}) + \alpha_{t_i}\sqrt{\lambda_{t_i}^2 - \lambda_{t_{i-1}}^2\Big[\frac{\phi(\lambda_{t_i})}{\phi(\lambda_{t_{i-1}})}\Big]^2} \boldsymbol{z}_{t_{i-1}} \\
&+ \alpha_{t_i}\sum_{n=1}^{k-1} \boldsymbol{x}_\theta^{(n)}(\tilde{\boldsymbol{x}}_{\lambda_{t_{i-1}}}, \lambda_{t_{i-1}})\Big[\frac{(\lambda_{t_i} - \lambda_{t_{i-1}})^n}{n!} + \phi(\lambda_{t_i})\int_{\lambda_{t_i}}^{\lambda_{t_{i-1}}} \frac{(\lambda - \lambda_{t_{i-1}})^{n-1}}{(n-1)!\phi(\lambda)} d\lambda\Big],
\end{aligned}
\tag{44}
$$

where $k \geq 2$. Notably, when $k = 1$, the summation term in Eq.(44) with the upper index smaller than the lower index can be defined as an empty sum, and its value is $0$ (Treeby & Cox, 2014). In conclusion, Eq.(44) holds when $k \geq 1$.

**Note:** The proposed algorithm has a global error of at least $\mathcal{O}((\lambda_{t_i} - \lambda_{t_{i-1}}))$ (Gonzalez et al., 2023). Therefore, when $k = 1$, we refer to it as a first-order solver with a strong convergence guarantee. When $k \geq 2$, we designate it as a $k$th-stage solver in accordance with the statement in Gonzalez et al. (2023).

### A.6 RELATIONSHIP WITH SDE AND ODE

We derive conditions under which ER SDE reduces to SDE (Song et al., 2021b) and ODE (Song et al., 2021b) from the perspective of noise scale function selection.

**Related to SDE:** When $h(t) = g(t)$, ER SDE reduces to SDE.

For the VE SDE, we have

$$
h^2(t) = g^2(t) = \frac{d\sigma_t^2}{dt} = 2\sigma_t \frac{d\sigma_t}{dt}.
\tag{45}
$$

Since $h^2(t) = \xi(t)\frac{d\sigma_t}{dt}$, we further obtain $\xi(t) = 2\sigma_t$ and $\xi(\sigma) = 2\sigma$. Substituting it into $\int \frac{1}{\sigma} + \frac{\xi(\sigma)}{2\sigma^2} d\sigma = \ln\phi(\sigma)$, we have

$$
\begin{aligned}
\int \frac{1}{\sigma} + \frac{2\sigma}{2\sigma^2} d\sigma &= \ln\phi(\sigma) \\
\phi(\sigma) &= \sigma^2.
\end{aligned}
\tag{46}
$$

For a VP SDE, we also have

$$
h(t) = g(t) = \sqrt{\frac{d\sigma_t^2}{dt} - 2\frac{d\log\alpha_t}{dt}\sigma_t^2}.
\tag{47}
$$

Thus,

$$
\begin{aligned}
\eta(t) &= \frac{h(t)}{\alpha_t} = \frac{1}{\alpha_t}\sqrt{\frac{d\sigma_t^2}{dt} - 2\frac{d\log\alpha_t}{dt}\sigma_t^2} \\
\eta^2(t) &= \frac{1}{\alpha_t^2}\Big(\frac{d\sigma_t^2}{dt} - 2\frac{d\log\alpha_t}{dt}\sigma_t^2\Big) = 2\frac{\sigma_t}{\alpha_t}\Big(\frac{1}{\alpha_t}\frac{d\sigma_t}{dt} - \frac{\sigma_t}{\alpha_t^2}\frac{d\alpha_t}{dt}\Big).
\end{aligned}
\tag{48}
$$

By $\lambda_t = \frac{\sigma_t}{\alpha_t}$, it holds that

$$
d\lambda_t = \frac{1}{\alpha_t}\frac{d\sigma_t}{dt} - \frac{\sigma_t}{\alpha_t^2}\frac{d\alpha_t}{dt}.
\tag{49}
$$

Substituting Eq.(49) into Eq.(48), we have

$$\eta^2(t) = 2\frac{\sigma_t}{\alpha_t}\frac{d\lambda_t}{dt} = 2\lambda_t\frac{d\lambda_t}{dt}. \tag{50}$$

Since $\eta^2(t) = \xi(t)\frac{d\lambda_t}{dt}$, we further obtain $\xi(t) = 2\lambda_t$ and $\xi(\lambda) = 2\lambda$. Substituting it into $\int \frac{1}{\lambda} + \frac{\xi(\lambda)}{2\sigma^2}d\lambda = \ln\phi(\lambda)$, we have

$$\int \frac{1}{\lambda} + \frac{2\lambda}{2\lambda^2}d\lambda = \ln\phi(\lambda) \tag{51}$$
$$\phi(\lambda) = \lambda^2.$$

Therefore, when the noise scale function $\phi(x) = x^2$, ER SDE reduces to SDE.

**Related to ODE:** When $h(t) = 0$, ER SDE reduces to ODE.

For the VE SDE, we have $\xi(t) = 0$ and $\xi(\sigma) = 0$. Substituting it into $\int \frac{1}{\sigma} + \frac{\xi(\sigma)}{2\sigma^2}d\sigma = \ln\phi(\sigma)$, we have

$$\int \frac{1}{\sigma}d\sigma = \ln\phi(\sigma) \tag{52}$$
$$\phi(\sigma) = \sigma.$$

For a VP SDE, we also have $\eta(t) = 0, \xi(t) = 0$ and $\xi(\sigma) = 0$. Substituting it into $\int \frac{1}{\lambda} + \frac{\xi(\lambda)}{2\sigma^2}d\lambda = \ln\phi(\lambda)$, we have

$$\int \frac{1}{\lambda}d\lambda = \ln\phi(\lambda) \tag{53}$$
$$\phi(\lambda) = \lambda.$$

Therefore, when the noise scale function $\phi(x) = x$, ER SDE reduces to ODE.

## A.7 RESTRICTION OF $\phi(x)$

Due to the fact that the constraint equation $\int \frac{1}{\sigma} + \frac{\xi(\sigma)}{2\sigma^2}d\sigma = \ln\phi(\sigma)$ for VE SDE and VP SDE have the same expression form (see Eq.10 and Eq.14), VE SDE is used as an example for derivation here.

Since $\phi(x)$ satisfies $\frac{1}{\sigma} + \frac{\xi(\sigma)}{2\sigma^2} = \frac{\phi^{(1)}(\sigma)}{\phi(\sigma)}$ and $\xi(\sigma) \geq 0$, we have

$$\frac{\phi^{(1)}(\sigma)}{\phi(\sigma)} \geq \frac{1}{\sigma}. \tag{54}$$

Suppose $t < s$ and $\sigma(t)$ is monotonically increasing, we have

$$\phi^{(1)}(\sigma_s) = \lim_{t \to s}\frac{\phi(\sigma_s) - \phi(\sigma_t)}{\sigma_s - \sigma_t}. \tag{55}$$

Combining with Eq.54, we obtain

$$\lim_{t \to s}\frac{\phi(\sigma_s) - \phi(\sigma_t)}{\sigma_s - \sigma_t} \geq \frac{\phi(\sigma_s)}{\sigma_s}, \tag{56}$$

which means that when $t$ is in the left neighboring domain of $s$, it satisfies

$$\frac{\phi(\sigma_t)}{\phi(\sigma_s)} \leq \frac{\sigma_t}{\sigma_s}. \tag{57}$$

In fact, Eq.(57) is consistent with the limitation of noise term in Eq.10, which is

$$\sqrt{\sigma_t^2 - \sigma_s^2\left[\frac{\phi(\sigma_t)}{\phi(\sigma_s)}\right]^2} \geq 0$$
$$\left[\frac{\phi(\sigma_t)}{\phi(\sigma_s)}\right]^2 \leq \frac{\sigma_t^2}{\sigma_s^2} \tag{58}$$
$$\frac{\phi(\sigma_t)}{\phi(\sigma_s)} \leq \frac{\sigma_t}{\sigma_s}.$$

## A.8 HOW TO CUSTOMIZE THE NOISE SCALE FUNCTION $\phi(x)$

The noise scale function $\phi(x)$ constitutes the solution space of ER SDE. In this section, we elaborate on two ways to customize $\phi(x)$.

**Indirect determination method via the intermediate variable $\xi(x)$:** For the ER SDE in Eq.(7), $h(t)$ can take any function as long as $h(t) \geq 0$. For the VE SDE, $h^2(t) = \xi(t)\frac{d\sigma_t}{dt}$ and $\frac{d\sigma_t}{dt} \geq 0$. Thus, $\xi(t)$ can take on any arbitrary function, provided that $\xi(t) \geq 0$ and $\xi(\sigma)$ also meets this condition. For the VP SDE, $h(t) = \eta(t)\alpha_t$, $\eta^2(t) = \xi(t)\frac{d\lambda_t}{dt}$ and $\frac{d\lambda_t}{dt} \geq 0$. Similarly, $\xi(\lambda)$ is also can take any function as long as $\xi(\lambda) \geq 0$. In general, as long as $\xi(x)$ takes a function which satisfies $\xi(x) \geq 0$, and then based on the constraint equation $\int \frac{1}{\sigma} + \frac{\xi(\sigma)}{2\sigma^2} d\sigma = \ln \phi(\sigma)$, we can obtain the corresponding $\phi(x)$. Then, by plotting the FEI-step curve (just like Fig.3(a)), we can observe the discretization errors caused by the determined $\phi(x)$.

**Direct determination method:** The first method involves determining an initial function $\xi(x)$ and then obtaining the corresponding $\phi(x)$ by computing an indefinite integral $\int \frac{1}{\sigma} + \frac{\xi(\sigma)}{2\sigma^2} d\sigma$. However, this integral is not easy to calculate in many cases. Appendix A.7 mentions that this indefinite integral is equivalent to Eq.(57) and the right-hand side of the inequality corresponds to the ODE case (see Appendix A.6). Therefore, $\phi(x)$ can be directly written without considering the specific expression of $\xi(x)$. The FEI-step curve can be plotted to check whether $\phi(x)$ is valid. More specifically, only if the FEI-step curve drawn based on $\phi(x)$ lies above the ODE curve, $\phi(x)$ is valid.

## B ALGORITHMS

The first to third-stage solvers for VE SDE are provided in Algorithm 1, 2 and 3. The detailed VP ER-SDE-Solver-1, 2, 3 are listed in Algorithms 4, 5, 6 respectively.

---

**Algorithm 1** VE ER-SDE-Solver-1(Euler).

---

**Require:** initial value $\boldsymbol{x}_T$, time steps $\{t_i\}_{i=0}^M$, customized noise scale function $\phi$, data prediction model $\boldsymbol{x}_\theta$.

1: $\boldsymbol{x}_{t_0} \leftarrow \boldsymbol{x}_T$
2: **for** $i \leftarrow 1$ to $M$ **do**
3:    $r_i \leftarrow \frac{\phi(\sigma_{t_i})}{\phi(\sigma_{t_{i-1}})}, \boldsymbol{z} \sim \mathcal{N}(\boldsymbol{0}, \mathbf{I})$
4:    $\boldsymbol{n}_{t_i} \leftarrow \sqrt{\sigma_{t_i}^2 - r_i^2 \sigma_{t_{i-1}}^2} \boldsymbol{z}, \tilde{\boldsymbol{x}}_0 \leftarrow \boldsymbol{x}_\theta(\boldsymbol{x}_{t_{i-1}}, t_{i-1})$
5:    $\boldsymbol{x}_{t_i} \leftarrow r_i \boldsymbol{x}_{t_{i-1}} + (1 - r_i)\tilde{\boldsymbol{x}}_0 + \boldsymbol{n}_{t_i}$
6: **end for**
7: **return** $\boldsymbol{x}_{t_M}$

---

---

**Algorithm 2** VE ER-SDE-Solver-2.

---

**Require:** initial value $\boldsymbol{x}_T$, time steps $\{t_i\}_{i=0}^M$, customized noise scale function $\phi$, data prediction model $\boldsymbol{x}_\theta$, number of numerical integration points $N$.

1: $\boldsymbol{x}_{t_0} \leftarrow \boldsymbol{x}_T, Q \leftarrow None$
2: **for** $i \leftarrow 1$ to $M$ **do**
3:    $r_i \leftarrow \frac{\phi(\sigma_{t_i})}{\phi(\sigma_{t_{i-1}})}, \boldsymbol{z} \sim \mathcal{N}(\boldsymbol{0}, \mathbf{I})$
4:    $\boldsymbol{n}_{t_i} \leftarrow \sqrt{\sigma_{t_i}^2 - r_i^2 \sigma_{t_{i-1}}^2} \boldsymbol{z}$
5:    **if** $Q = None$ **then** $\boldsymbol{x}_{t_i} \leftarrow r_i \boldsymbol{x}_{t_{i-1}} + (1 - r_i)\boldsymbol{x}_\theta(\boldsymbol{x}_{t_{i-1}}, t_{i-1}) + \boldsymbol{n}_{t_i}$
6:    **else**
7:       $\Delta_\sigma \leftarrow \frac{\sigma_{t_{i-1}} - \sigma_{t_i}}{N}, S_i \leftarrow \sum_{k=0}^{N-1} \frac{\Delta_\sigma}{\phi(\sigma_{t_i} + k\Delta_\sigma)}$     $\triangleright$ Numerical integration
8:       $\mathbf{D}_i \leftarrow \frac{\boldsymbol{x}_\theta(\boldsymbol{x}_{t_{i-1}}, t_{i-1}) - \boldsymbol{x}_\theta(\boldsymbol{x}_{t_{i-2}}, t_{i-2})}{\sigma_{t_{i-1}} - \sigma_{t_{i-2}}}, \boldsymbol{\delta}_{t_i} \leftarrow [\sigma_{t_i} - \sigma_{t_{i-1}} + S_i \phi(\sigma_{t_i})]\mathbf{D}_i$
9:       $\boldsymbol{x}_{t_i} \leftarrow r_i \boldsymbol{x}_{t_{i-1}} + (1 - r_i)\boldsymbol{x}_\theta(\boldsymbol{x}_{t_{i-1}}, t_{i-1}) + \boldsymbol{\delta}_{t_i} + \boldsymbol{n}_{t_i}$
10:    **end if**
11:    $Q \xleftarrow{\text{buffer}} \boldsymbol{x}_\theta(\boldsymbol{x}_{t_{i-1}}, t_{i-1})$
12: **end for**
13: **return** $\boldsymbol{x}_{t_M}$

---

---

**Algorithm 3** VE ER-SDE-Solver-3.

---

**Require:** initial value $\boldsymbol{x}_T$, time steps $\{t_i\}_{i=0}^M$, customized noise scale function $\phi$, data prediction model $\boldsymbol{x}_\theta$, number of numerical integration points $N$.

1: $\boldsymbol{x}_{t_0} \leftarrow \boldsymbol{x}_T$
2: $Q \leftarrow None$
3: $Q_d \leftarrow None$
4: **for** $i \leftarrow 1$ to $M$ **do**
5:      $r_i \leftarrow \frac{\phi(\sigma_{t_i})}{\phi(\sigma_{t_{i-1}})}$
6:      $\boldsymbol{z} \sim \mathcal{N}(\boldsymbol{0}, \mathbf{I})$
7:      $\boldsymbol{n}_{t_i} \leftarrow \sqrt{\sigma_{t_i}^2 - r_i^2 \sigma_{t_{i-1}}^2}\, \boldsymbol{z}$
8:      **if** $Q = None$ and $Q_d = None$ **then**
9:         $\boldsymbol{x}_{t_i} \leftarrow r_i \boldsymbol{x}_{t_{i-1}} + (1 - r_i)\boldsymbol{x}_\theta(\boldsymbol{x}_{t_{i-1}}, t_{i-1}) + \boldsymbol{n}_{t_i}$
10:      **else if** $Q \neq None$ and $Q_d = None$ **then**
11:         $\Delta_\sigma \leftarrow \frac{\sigma_{t_{i-1}} - \sigma_{t_i}}{N}$
12:         $S_i \leftarrow \sum_{k=0}^{N-1} \frac{\Delta_\sigma}{\phi(\sigma_{t_i} + k\Delta_\sigma)}$      ▷ Numerical integration
13:         $\mathbf{D}_i \leftarrow \frac{\boldsymbol{x}_\theta(\boldsymbol{x}_{t_{i-1}}, t_{i-1}) - \boldsymbol{x}_\theta(\boldsymbol{x}_{t_{i-2}}, t_{i-2})}{\sigma_{t_{i-1}} - \sigma_{t_{i-2}}}$
14:         $\boldsymbol{\delta}_{t_i} \leftarrow [\sigma_{t_i} - \sigma_{t_{i-1}} + S_i \phi(\sigma_{t_i})]\mathbf{D}_i$
15:         $\boldsymbol{x}_{t_i} \leftarrow r_i \boldsymbol{x}_{t_{i-1}} + (1 - r_i)\boldsymbol{x}_\theta(\boldsymbol{x}_{t_{i-1}}, t_{i-1}) + \boldsymbol{\delta}_{t_i} + \boldsymbol{n}_{t_i}$
16:         $Q_d \xleftarrow{\text{buffer}} \mathbf{D}_i$
17:      **else**
18:         $\Delta_\sigma \leftarrow \frac{\sigma_{t_{i-1}} - \sigma_{t_i}}{N}$
19:         $S_i \leftarrow \sum_{k=0}^{N-1} \frac{\Delta_\sigma}{\phi(\sigma_{t_i} + k\Delta_\sigma)}$
20:         $S_{d_i} \leftarrow \sum_{k=0}^{N-1} \frac{\sigma_{t_i} + k\Delta_\sigma - \sigma_{t_{i-1}}}{\phi(\sigma_{t_i} + k\Delta_\sigma)}\Delta_\sigma$
21:         $\mathbf{D}_i \leftarrow \frac{\boldsymbol{x}_\theta(\boldsymbol{x}_{t_{i-1}}, t_{i-1}) - \boldsymbol{x}_\theta(\boldsymbol{x}_{t_{i-2}}, t_{i-2})}{\sigma_{t_{i-1}} - \sigma_{t_{i-2}}}$
22:         $\boldsymbol{\delta}_{dt_i} \leftarrow \left[\frac{(\sigma_{t_i} - \sigma_{t_{i-1}})^2}{2} + S_{d_i}\phi(\sigma_{t_i})\right]\mathbf{U}_i$
23:         $\mathbf{U}_i \leftarrow \frac{\mathbf{D}_i - \mathbf{D}_{i-1}}{\frac{\sigma_{t_{i-1}} - \sigma_{t_{i-3}}}{2}}$
24:         $\boldsymbol{\delta}_{t_i} \leftarrow [\sigma_{t_i} - \sigma_{t_{i-1}} + S_i \phi(\sigma_{t_i})]\mathbf{D}_i$
25:         $\boldsymbol{x}_{t_i} \leftarrow r_i \boldsymbol{x}_{t_{i-1}} + (1 - r_i)\boldsymbol{x}_\theta(\boldsymbol{x}_{t_{i-1}}, t_{i-1}) + \boldsymbol{\delta}_{t_i} + \boldsymbol{\delta}_{dt_i} + \boldsymbol{n}_{t_i}$
26:         $Q_d \xleftarrow{\text{buffer}} \mathbf{D}_i$
27:      **end if**
28:      $Q \xleftarrow{\text{buffer}} \boldsymbol{x}_\theta(\boldsymbol{x}_{t_{i-1}}, t_{i-1})$
29: **end for**
30: **return** $\boldsymbol{x}_{t_M}$

---

---

**Algorithm 4** VP ER-SDE-Solver-1(Euler).

---

**Require:** initial value $\boldsymbol{x}_T$, time steps $\{t_i\}_{i=0}^M$, customized noise scale function $\phi$, data prediction model $\boldsymbol{x}_\theta$.

1: $\boldsymbol{x}_{t_0} \leftarrow \boldsymbol{x}_T$
2: **for** $i \leftarrow 1$ to $M$ **do**
3:      $r_i \leftarrow \frac{\phi(\lambda_{t_i})}{\phi(\lambda_{t_{i-1}})}$
4:      $r_{\alpha_i} \leftarrow \frac{\alpha_{t_i}}{\alpha_{t_{i-1}}}$
5:      $\boldsymbol{z} \sim \mathcal{N}(\boldsymbol{0}, \mathbf{I})$
6:      $\boldsymbol{n}_{t_i} \leftarrow \alpha_{t_i}\sqrt{\lambda_{t_i}^2 - r_i^2 \lambda_{t_{i-1}}^2}\, \boldsymbol{z}$
7:      $\tilde{\boldsymbol{x}}_0 \leftarrow \boldsymbol{x}_\theta(\boldsymbol{x}_{t_{i-1}}, t_{i-1})$
8:      $\boldsymbol{x}_{t_i} \leftarrow r_{\alpha_i} r_i \boldsymbol{x}_{t_{i-1}} + \alpha_{t_i}(1 - r_i)\tilde{\boldsymbol{x}}_0 + \boldsymbol{n}_{t_i}$
9: **end for**
10: **return** $\boldsymbol{x}_{t_M}$

---

---

**Algorithm 5** VP ER-SDE-Solver-2.

---

**Require:** initial value $\boldsymbol{x}_T$, time steps $\{t_i\}_{i=0}^M$, customized noise scale function $\phi$, data prediction model $\boldsymbol{x}_\theta$, number of numerical integration points $N$.

1: $\boldsymbol{x}_{t_0} \leftarrow \boldsymbol{x}_T, Q \leftarrow None$
2: **for** $i \leftarrow 1$ to $M$ **do**
3:      $r_i \leftarrow \frac{\phi(\lambda_{t_i})}{\phi(\lambda_{t_{i-1}})}, r_{\alpha_i} \leftarrow \frac{\alpha_{t_i}}{\alpha_{t_{i-1}}}, \boldsymbol{z} \sim \mathcal{N}(\mathbf{0}, \mathbf{I})$
4:      $\boldsymbol{n}_{t_i} \leftarrow \alpha_{t_i}\sqrt{\lambda_{t_i}^2 - r_i^2 \lambda_{t_{i-1}}^2}\boldsymbol{z}$
5:      **if** $Q = None$ **then**
6:          $\boldsymbol{x}_{t_i} \leftarrow r_{\alpha_i} r_i \boldsymbol{x}_{t_{i-1}} + \alpha_{t_i}(1 - r_i)\boldsymbol{x}_\theta(\boldsymbol{x}_{t_{i-1}}, t_{i-1}) + \boldsymbol{n}_{t_i}$
7:      **else**
8:          $\Delta_\lambda \leftarrow \frac{\lambda_{t_{i-1}} - \lambda_{t_i}}{N}, S_i \leftarrow \sum_{k=0}^{N-1} \frac{\Delta_\lambda}{\phi(\lambda_{t_i} + k\Delta_\lambda)}$          ▷ Numerical integration
9:          $\mathbf{D}_i \leftarrow \frac{\boldsymbol{x}_\theta(\boldsymbol{x}_{t_{i-1}}, t_{i-1}) - \boldsymbol{x}_\theta(\boldsymbol{x}_{t_{i-2}}, t_{i-2})}{\lambda_{t_{i-1}} - \lambda_{t_{i-2}}}$
10:         $\boldsymbol{\delta}_{t_i} \leftarrow \alpha_{t_i}[\lambda_{t_i} - \lambda_{t_{i-1}} + S_i\phi(\lambda_{t_i})]\mathbf{D}_i$
11:         $\boldsymbol{x}_{t_i} \leftarrow r_{\alpha_i} r_i \boldsymbol{x}_{t_{i-1}} + \alpha_{t_i}(1 - r_i)\boldsymbol{x}_\theta(\boldsymbol{x}_{t_{i-1}}, t_{i-1}) + \boldsymbol{\delta}_{t_i} + \boldsymbol{n}_{t_i}$
12:      **end if**
13:      $Q \xleftarrow{\text{buffer}} \boldsymbol{x}_\theta(\boldsymbol{x}_{t_{i-1}}, t_{i-1})$
14: **end for**
15: **return** $\boldsymbol{x}_{t_M}$

---

**Algorithm 6** VP ER-SDE-Solver-3.

---

**Require:** initial value $\boldsymbol{x}_T$, time steps $\{t_i\}_{i=0}^M$, customized noise scale function $\phi$, data prediction model $\boldsymbol{x}_\theta$, number of numerical integration points $N$.

1: $\boldsymbol{x}_{t_0} \leftarrow \boldsymbol{x}_T, Q \leftarrow None, Q_d \leftarrow None$
2: **for** $i \leftarrow 1$ to $M$ **do**
3:      $r_i \leftarrow \frac{\phi(\lambda_{t_i})}{\phi(\lambda_{t_{i-1}})}, r_{\alpha_i} \leftarrow \frac{\alpha_{t_i}}{\alpha_{t_{i-1}}}, \boldsymbol{z} \sim \mathcal{N}(\mathbf{0}, \mathbf{I})$
4:      $\boldsymbol{n}_{t_i} \leftarrow \alpha_{t_i}\sqrt{\lambda_{t_i}^2 - r_i^2 \lambda_{t_{i-1}}^2}\boldsymbol{z}$
5:      **if** $Q = None$ and $Q_d = None$ **then**
6:          $\boldsymbol{x}_{t_i} \leftarrow r_{\alpha_i} r_i \boldsymbol{x}_{t_{i-1}} + \alpha_{t_i}(1 - r_i)\boldsymbol{x}_\theta(\boldsymbol{x}_{t_{i-1}}, t_{i-1}) + \boldsymbol{n}_{t_i}$
7:      **else if** $Q \neq None$ and $Q_d = None$ **then**
8:          $\Delta_\lambda \leftarrow \frac{\lambda_{t_{i-1}} - \lambda_{t_i}}{N}, S_i \leftarrow \sum_{k=0}^{N-1} \frac{\Delta_\lambda}{\phi(\lambda_{t_i} + k\Delta_\lambda)}$          ▷ Numerical integration
9:          $\mathbf{D}_i \leftarrow \frac{\boldsymbol{x}_\theta(\boldsymbol{x}_{t_{i-1}}, t_{i-1}) - \boldsymbol{x}_\theta(\boldsymbol{x}_{t_{i-2}}, t_{i-2})}{\lambda_{t_{i-1}} - \lambda_{t_{i-2}}}$
10:         $\boldsymbol{\delta}_{t_i} \leftarrow \alpha_{t_i}[\lambda_{t_i} - \lambda_{t_{i-1}} + S_i\phi(\lambda_{t_i})]\mathbf{D}_i$
11:         $\boldsymbol{x}_{t_i} \leftarrow r_{\alpha_i} r_i \boldsymbol{x}_{t_{i-1}} + \alpha_{t_i}(1 - r_i)\boldsymbol{x}_\theta(\boldsymbol{x}_{t_{i-1}}, t_{i-1}) + \boldsymbol{\delta}_{t_i} + \boldsymbol{n}_{t_i}$
12:         $Q_d \xleftarrow{\text{buffer}} \mathbf{D}_i$
13:      **else**
14:          $\Delta_\lambda \leftarrow \frac{\lambda_{t_{i-1}} - \lambda_{t_i}}{N}, S_i \leftarrow \sum_{k=0}^{N-1} \frac{\Delta_\lambda}{\phi(\lambda_{t_i} + k\Delta_\lambda)}, S_{d_i} \leftarrow \sum_{k=0}^{N-1} \frac{\lambda_{t_i} + k\Delta_\lambda - \lambda_{t_{i-1}}}{\phi(\lambda_{t_i} + k\Delta_\lambda)}\Delta_\lambda$
15:          $\mathbf{D}_i \leftarrow \frac{\boldsymbol{x}_\theta(\boldsymbol{x}_{t_{i-1}}, t_{i-1}) - \boldsymbol{x}_\theta(\boldsymbol{x}_{t_{i-2}}, t_{i-2})}{\lambda_{t_{i-1}} - \lambda_{t_{i-2}}}$
16:          $\mathbf{U}_i \leftarrow \frac{\mathbf{D}_i - \mathbf{D}_{i-1}}{\frac{\lambda_{t_{i-1}} - \lambda_{t_{i-3}}}{2}}$
17:          $\boldsymbol{\delta}_{t_i} \leftarrow \alpha_{t_i}[\lambda_{t_i} - \lambda_{t_{i-1}} + S_i\phi(\lambda_{t_i})]\mathbf{D}_i$
18:          $\boldsymbol{\delta}_{dt_i} \leftarrow \alpha_{t_i}\left[\frac{(\lambda_{t_i} - \lambda_{t_{i-1}})^2}{2} + S_{d_i}\phi(\lambda_{t_i})\right]\mathbf{U}_i$
19:          $\boldsymbol{x}_{t_i} \leftarrow r_{\alpha_i} r_i \boldsymbol{x}_{t_{i-1}} + \alpha_{t_i}(1 - r_i)\boldsymbol{x}_\theta(\boldsymbol{x}_{t_{i-1}}, t_{i-1}) + \boldsymbol{\delta}_{t_i} + \boldsymbol{\delta}_{dt_i} + \boldsymbol{n}_{t_i}$
20:          $Q_d \xleftarrow{\text{buffer}} \mathbf{D}_i$
21:      **end if**
22:      $Q \xleftarrow{\text{buffer}} \boldsymbol{x}_\theta(\boldsymbol{x}_{t_{i-1}}, t_{i-1})$
23: **end for**
24: **return** $\boldsymbol{x}_{t_M}$

---

# C IMPLEMENTATION DETAILS

We test our sampling method on VE-type and VP-type pretrained diffusion models. For VE-type, we select EDM (Karras et al., 2022) as the pretrained diffusion model. Although EDM differs slightly from the commonly used VE-type diffusion model (Song et al., 2021b), its forward process still follows the VE SDE described in Eq.(2). Therefore, it can be regarded as a generalized VE-type diffusion model, which is proved in C.3. Furthermore, EDM provides a method that can be converted into VP-type diffusion models, facilitating a fair comparison between VE SDE solvers and VP SDE solvers using the same model weights. For VP-type pretrained diffusion models, we choose widely-used Guided-diffusion (Dhariwal & Nichol, 2021). Different from EDM, Guided-diffusion provides pretrained models on high-resolution datasets, such as ImageNet $128 \times 128$ (Deng et al., 2009), LSUN $256 \times 256$ (Yu et al., 2015) and so on. For all experiments, we evaluate our method on NVIDIA GeForce RTX 3090 GPUs.

## C.1 STEP SIZE SCHEDULE

With EDM pretrained model, the time step size of ER-SDE-Solvers and other comparison methods are aligned with EDM (Karras et al., 2022). The specific time step size is

$$t_{i<M} = \left[ \sigma_{\max}^{\frac{1}{\rho}} + \frac{i}{M-1} (\sigma_{\min}^{\frac{1}{\rho}} - \sigma_{\max}^{\frac{1}{\rho}}) \right]^{\rho}, \tag{59}$$

where $\sigma_{min} = 0.002, \sigma_{max} = 80, \rho = 7$.

With Guided-diffusion pretrained model (Dhariwal & Nichol, 2021), we use the uniform time steps for our method and other comparison methods, i.e.,

$$t_{i<M} = 1 + \frac{i}{M-1}(\epsilon - 1). \tag{60}$$

In theory, we would need to solve the ER SDE from time $T$ to time $0$ to generate samples. However, taking inspiration from Lu et al. (2022a), to circumvent numerical issues near $t = 0$, the training and evaluation of the data prediction model $\boldsymbol{x}_\theta (\boldsymbol{x}_t, t)$ typically span from time $T$ to a small positive value $\epsilon$, where $\epsilon$ is a hyperparameter (Song et al., 2021b).

## C.2 NOISE SCHEDULE

In EDM, the noise schedule is equal to the time step size schedule, i.e.,

$$\sigma_{i<M} = \left[ \sigma_{\max}^{\frac{1}{\rho}} + \frac{i}{M-1} (\sigma_{\min}^{\frac{1}{\rho}} - \sigma_{\max}^{\frac{1}{\rho}}) \right]^{\rho}. \tag{61}$$

In Guided-diffusion, two commonly used noise schedules are provided (the linear and the cosine noise schedules). For the linear noise schedule, we have

$$\alpha_t = e^{-\frac{1}{4}t^2(\beta_{\max}-\beta_{\min})-\frac{1}{2}t\beta_{\min}}, \tag{62}$$

where $\beta_{\min} = 0.1$ and $\beta_{\max} = 20$, following Song et al. (2021b). For the cosine noise schedule, we have

$$\alpha_t = \frac{f(t)}{f(0)}, \quad f(t) = \cos\left(\frac{t/T + s}{1 + s} \cdot \frac{\pi}{2}\right)^2, \tag{63}$$

where $s = 0.008$, following Nichol & Dhariwal (2021). Since Guided-diffusion is a VP-type pretrained model, it satisfies $\alpha_t^2 + \sigma_t^2 = 1$.

## C.3 RELATIONSHIP BETWEEN EDM AND VE,VP

EDM describes the forward process as follows:

$$\boldsymbol{x}(t) = s(t)\boldsymbol{x}_0 + s(t)\sigma(t)\boldsymbol{z}, \quad \boldsymbol{z} \sim \mathcal{N}(\mathbf{0}, \mathbf{I}). \tag{64}$$

It is not difficult to find that $\alpha_t = s(t)$ and $\sigma_t = s(t)\sigma(t)$ in VP. In addition, the expression for $s(t)$ and $\sigma(t)$ w.r.t $t$ is

$$\sigma(t) = \sqrt{e^{\frac{1}{2}\beta_d t^2 + \beta_{\min} t} - 1} \tag{65}$$

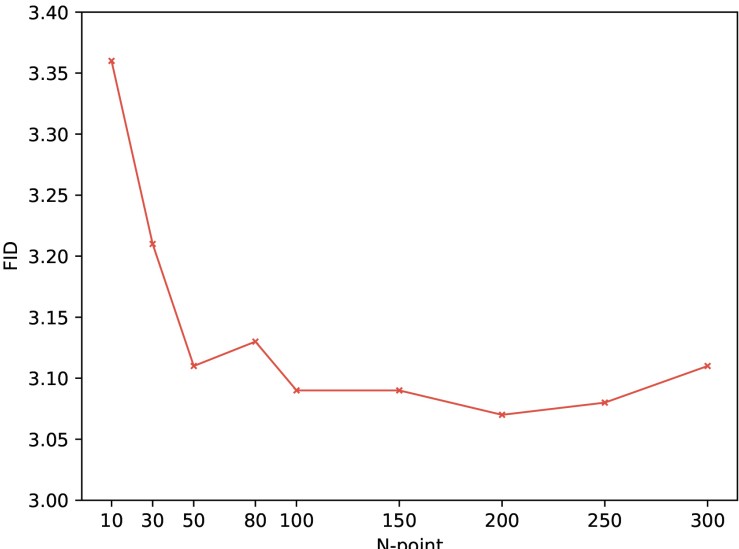

Figure 4: FID (NFE=20) on CIFAR-10 with the pretrained EDM, varying with the number of integration points. As the number of integration points $N$ increases, FID scores initially show a decreasing trend, reaching a minimum at $N = 200$. Subsequently, FID scores slowly increase, indicating a decrease in image generation quality.

and

$$s(t) = \frac{1}{\sqrt{e^{\frac{1}{2}\beta_d t^2 + \beta_{\min} t}}}. \tag{66}$$

where $\beta_d = 19.9$, $\beta_{\min} = 0.1$ and $t$ follows the uniform time size in Eq.(60). However, in order to match the noise schedule of EDM in Eq.(61), we let

$$\sigma(\epsilon) = \sigma_{\min}, \quad \sigma(1) = \sigma_{\max}. \tag{67}$$

Thus, we can get the corresponding $\beta_d$ and $\beta_{\min}$. The corresponding time step can also be obtained by using the inverse function of $\sigma(t)$.

Similarly, $s(t) = 1$ and $\sigma_t = \sigma(t)$ in VE, so we can directly use the EDM as VE-type pretrained diffusion models to match the noise schedule. Through the inverse function of $\sigma(t) = \sqrt{t}$, we can get the corresponding VE-type time step, following Karras et al. (2022).

### C.4    ABLATING NUMERICAL INTEGRATION POINTS $N$

As mentioned before, the terms $\int_{\sigma_{t_i}}^{\sigma_{t_{i-1}}} \frac{(\sigma - \sigma_{t_{i-1}})^{n-1}}{(n-1)!\phi(\sigma)} d\sigma$ in Eq.(11) and $\int_{\lambda_{t_i}}^{\lambda_{t_{i-1}}} \frac{(\lambda - \lambda_{t_{i-1}})^{n-1}}{(n-1)!\phi(\lambda)} d\lambda$ in Eq.(15) lack analytical expressions and need to be estimated using $N$-point numerical integration. In this section, we conduct ablation experiments to select an appropriate number of integration points. As Fig.4 shows, FID oscillations decrease with an increase in the number of points $N$, reaching a minimum at $N = 200$. Subsequently, image quality deteriorates as the number of points increases further. Additionally, a higher number of integration points leads to slower inference speed. Therefore, to strike a balance between efficiency and performance, we choose $N = 100$ for all experiments in this paper.

### C.5    CODE IMPLEMENTATION

Our code is available in the supplementary material, which is implemented with PyTorch. The implementation code for the pretrained model EDM is accessible at https://github.com/NVlabs/edm.

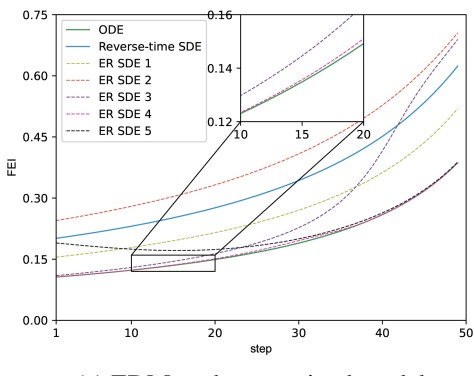
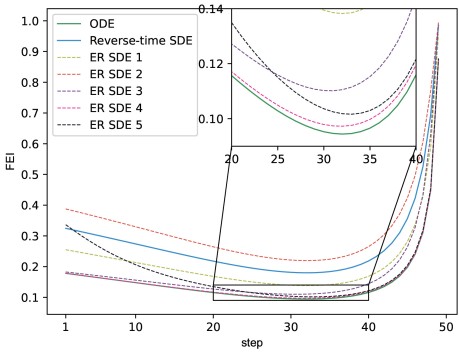

(a) EDM as the pretrained model      (b) Guided-diffusion as the pretrained model

Figure 5: FEI coefficients with the pretrained model EDM (a) and Guided-diffusion (b) (linear noise schedule), varying with the steps. Different step size schedules and noise schedules used in the pretrained models lead to variations in the shape of the FEI-step curves.

The implementation code for the pretrained model Guided-diffusion is accessible at https://github.com/openai/guided-diffusion and the code license is MIT License.

## D  EXPERIMENT DETAILS

We list all the experimental details and experimental results in this section.

Firstly, we have observed that the shape of the FEI-step curve is closely related to the choice of step size schedule and noise schedule within the pretrained diffusion models. Fig.5 illustrates two distinct FEI-step curves using EDM (Karras et al., 2022) and Guided-diffusion (Dhariwal & Nichol, 2021) as pretrained models.

Table 4 and Table 5 present the image generation quality of different noise scale functions on ImageNet $64 \times 64$ (Deng et al., 2009) and CIFAR-10 (Krizhevsky et al., 2009), described by FID scores (Heusel et al., 2017). Due to the discretization errors of ER SDE 4 being sufficiently close to the minimal error (see Fig.3), it exhibits outstanding FID scores. Despite the initial significant discretization error in ER SDE 5, its later-generated image quality is comparable to ER SDE 4. Particularly on the ImageNet $64 \times 64$ dataset, ER SDE 5 even surpasses ER SDE 4 after 20 NFE. This implies that practitioners may need to experiment with different noise scale functions for various scenarios. This paper aims to generate images that combine efficiency and high-quality, hence selecting ER SDE 5 as the noise scale function for ER-SDE-Solvers.

Table 4: Sample quality measured by FID↓ on ImageNet $64 \times 64$ for different noise scale functions with VE ER-SDE-Solver-3, varying the NFE. The pretrained model is EDM.

| Noise scale functions\NFE | 10 | 15 | 20 | 25 | 30 | 40 | 50 |
|---|---|---|---|---|---|---|---|
| ER SDE 4 | **10.79** | **4.99** | **3.45** | 2.96 | 2.71 | 2.43 | 2.38 |
| ER SDE 5 | 11.46 | 5.08 | **3.45** | **2.92** | **2.58** | **2.37** | **2.24** |

Table 5: Sample quality measured by FID↓ on CIFAR-10 for different noise scale functions with VE ER-SDE-Solver-3, varying the NFE. The pretrained model is EDM.

| Noise scale functions\NFE | 10 | 15 | 20 | 25 | 30 | 40 | 50 |
|---|---|---|---|---|---|---|---|
| ER SDE 4 | **9.28** | **4.37** | **3.01** | **2.52** | **2.25** | **2.07** | 1.97 |
| ER SDE 5 | 9.86 | 4.57 | 3.09 | 2.54 | 2.29 | 2.10 | **1.97** |

Table 6 shows how FID scores change with NFE for different stages of VE ER-SDE-Solvers and VP ER-SDE-Solvers on CIFAR-10. Consistent with the findings in Table 1, VE ER-SDE-Solvers and VP ER-SDE-Solvers demonstrate similar image generation quality. Furthermore, as the stage increases, convergence becomes faster with the decreasing discretization errors.

Table 6: Sample quality measured by FID↓ on CIFAR-10 for different stages of VE and VP ER-SDE-Solvers with the pretrained model EDM, varying the NFE. VE(P)-x denotes the x-th stage VE(P) ER-SDE-Solver.

| Method\NFE | 10 | 20 | 30 | 50 | Method\NFE | 10 | 20 | 30 | 50 |
|---|---|---|---|---|---|---|---|---|---|
| VE-2 | 10.33 | 3.33 | 2.39 | 2.03 | VE-3 | 9.86 | 3.13 | 2.31 | 1.97 |
| VP-2 | 10.12 | 3.28 | 2.41 | 2.08 | VP-3 | 9.77 | 3.02 | 2.29 | 1.96 |

Table 7 and Table 8 provide comparisons between ER-Solvers and other training-free samplers on CIFAR-10 and FFHQ $64 \times 64$ (Karras et al., 2019) with the same pretrained model EDM (Karras et al., 2022). The results consistently highlight ER-SDE-Solvers as the most efficient stochastic sampler. Furthermore, Table 7 and Table 8 reinforce our conclusion in Sec.4.1 that stochastic samplers exhibit lower image generation efficiency compared to deterministic samplers.

Table 7: Sample quality measured by FID↓ on CIFAR-10 with the pretrained model EDM, varying the NFE. The upper right $-$ indicates a reduction of NFE by one, while $*$ represents the FID obtained at NFE $= 35$.

|  | Sampling method\NFE | 10 | 20 | 30 | 50 |
|---|---|---|---|---|---|
| Stochastic Sampling | DDIM($\eta = 1$) (Song et al., 2021a) | 41.25 | 21.17 | 13.34 | 7.53 |
|  | SDE-DPM-Solver++(2M) (Lu et al., 2022b) | 14.88 | 4.67 | 3.13 | 2.39 |
|  | EDM-Stochastic (Karras et al., 2022) | $50.20^-$ | $4.59^-$ | $2.61^-$ | $2.25^-$ |
|  | Ours(ER-SDE-Solver-3) | **9.86** | **3.09** | **2.29** | **1.97** |
| Deterministic Sampling | DDIM (Song et al., 2021a) | 14.64 | 5.77 | 3.91 | 2.81 |
|  | EDM-Deterministic (Karras et al., 2022) | $35.40^-$ | $2.65^-$ | **$1.90^-$** | $1.84^*$ |
|  | DPM-Solver-3 (Lu et al., 2022a) | 5.52 | 2.47 | 2.10 | 1.94 |
|  | DPM-Solver++(2M) (Lu et al., 2022b) | **4.83** | **2.19** | 1.99 | 1.91 |

Table 8: Sample quality measured by FID↓ on FFHQ $64 \times 64$ with the pretrained model EDM, varying the NFE. The upper right $-$ indicates a reduction of NFE by one.

|  | Sampling method\NFE | 10 | 20 | 30 | 50 |
|---|---|---|---|---|---|
| Stochastic Sampling | DDIM($\eta = 1$) (Song et al., 2021a) | 57.06 | 33.26 | 22.93 | 13.61 |
|  | SDE-DPM-Solver++(2M) (Lu et al., 2022b) | 25.39 | 10.41 | 6.18 | 3.83 |
|  | EDM-Stochastic (Karras et al., 2022) | $94.16^-$ | $10.41^-$ | $4.79^-$ | $3.29^-$ |
|  | Ours(ER-SDE-Solver-3) | **14.77** | **4.67** | **3.34** | **2.78** |
| Deterministic Sampling | DDIM (Song et al., 2021a) | 21.19 | 8.73 | 5.91 | 4.11 |
|  | EDM-Deterministic (Karras et al., 2022) | $56.91^-$ | $5.43^-$ | $3.07^-$ | $2.65^-$ |
|  | DPM-Solver-3 (Lu et al., 2022a) | 8.13 | 3.55 | 2.94 | 2.67 |
|  | DPM-Solver++(2M) (Lu et al., 2022b) | **6.69** | **3.16** | **2.83** | **2.65** |

With EDM pretrained model used in Table 1 - 2 and 4 - 8, we evaluate all the methods using the same pretrained checkpoint provided by Karras et al. (2022). For a fair comparison, all the methods in our evaluation follow the noise schedule in Eq.(61). Although many techniques, such as *thresholding methods* (Lu et al., 2022b) and *numerical clip alpha* (Nichol & Dhariwal, 2021; Lu et al., 2022a; Dhariwal & Nichol, 2021), can help reduce FID scores, they are challenging to apply to all sampling methods equally. For example, *thresholding methods* are usually used in the

data prediction model or where it comes to prediction data, but they cannot directly be used in the noise prediction model. Therefore, for the purpose of fair comparison and evaluating the true efficacy of these sampling methods, none of the FID scores we report use these techniques. That is why they may appear slightly higher than the FID scores in the original paper. For experiments involving the ImageNet $64 \times 64$ dataset, we use the checkpoint edm-imagenet-64x64-cond-adm.pkl. For experiments involving the CIFAR-10 dataset, we use the checkpoint edm-cifar10-32x32-cond-ve.pkl. For experiments involving the FFHQ $64 \times 64$ dataset, we use the checkpoint edm-ffhq-64x64-uncond-ve.pkl. For the three categories of experiments mentioned above, we calculate FID scores using random seeds 100000-149999, following Karras et al. (2022). For comparison method EDM-Stochastic, we use the optimal settings mentioned in Karras et al. (2022), which are $S_{churn} = 80, S_{min} = 0.05, S_{max} = 1, S_{noise} = 1.007$ in Table 7 and $S_{churn} = 40, S_{min} = 0.05, S_{max} = 50, S_{noise} = 1.003$ in Table 2 and Table 8.

Next, we apply ER-SDE-Solvers to high-resolution image generation. Table 9 - Table 10 provide comparative results on ImageNet $128 \times 128$ (Deng et al., 2009) and LSUN $256 \times 256$ (Yu et al., 2015), using Guided-diffusion (Dhariwal & Nichol, 2021) as the pretrained model. The results demonstrate that ER-SDE-Solvers can also accelerate the generation of high-resolution images, significantly surpassing deterministic samplers in 20 NFE.

Table 9: Sample quality measured by FID↓ on class-conditional ImageNet $128 \times 128$ with the pretrained model Guided-diffusion (without classifier guidance, linear noise schedule), varying the NFE.

|  | Sampling method\NFE | 10 | 20 | 30 | 50 |
|---|---|---|---|---|---|
| Stochastic Sampling | DDIM($\eta = 1$) (Song et al., 2021a) | 40.60 | 21.23 | 14.95 | 10.42 |
|  | SDE-DPM-Solver++(2M) (Lu et al., 2022b) | 16.31 | 9.73 | 8.27 | 7.83 |
|  | Ours(ER-SDE-Solver-3) | **11.95** | **8.33** | **7.84** | **7.76** |
| Deterministic Sampling | DDIM (Song et al., 2021a) | 19.86 | 11.49 | 9.79 | **8.65** |
|  | DPM-Solver-3 (Lu et al., 2022a) | **11.77** | **9.55** | **9.39** | 9.02 |
|  | DPM-Solver++(2M) (Lu et al., 2022b) | 12.32 | 10.50 | 10.04 | 9.61 |

Table 10: Sample quality measured by FID↓ on unconditional LSUN Bedrooms $256 \times 256$ with the pretrained model Guided-diffusion (linear noise schedule), varying the NFE.

|  | Sampling method\NFE | 10 | 20 | 30 | 50 |
|---|---|---|---|---|---|
| Stochastic Sampling | DDIM($\eta = 1$) (Song et al., 2021a) | 38.01 | 19.80 | 13.76 | 8.68 |
|  | SDE-DPM-Solver++(2M) (Lu et al., 2022b) | 13.70 | 6.53 | 4.53 | 3.31 |
|  | Ours(ER-SDE-Solver-3) | **9.34** | **4.96** | **3.55** | **2.71** |
| Deterministic Sampling | DDIM (Song et al., 2021a) | 12.33 | 6.39 | 4.77 | 3.67 |
|  | PNDM (Liu et al., 2022) | 10.20 | 5.68 | — | — |
|  | DPM-Solver-3 (Lu et al., 2022a) | 8.20 | 4.51 | **3.45** | **2.71** |
|  | DPM-Solver++(2M) (Lu et al., 2022b) | 10.43 | 5.70 | 4.02 | 3.15 |
|  | $t$AB-DEIS (Zhang & Chen, 2023) | **8.18** | **4.10** | 3.54 | 3.12 |

With Guided-diffusion pretrained model used in Table 3, and Table 9 - Table 14, we evaluate all the methods using the same pretrained checkpoint provided by Dhariwal & Nichol (2021). Similarly, for the purpose of fair comparison and evaluating the true efficacy of these sampling methods, none of the FID scores we report use techniques like *thresholding methods* or *numerical clip alpha*. That is why they may appear slightly higher than the FID scores in the original paper. For experiments involving the ImageNet $128 \times 128$ dataset, we use the checkpoint 128x128_diffusion.pt. All the methods in our evaluation follow the linear schedule and uniform time steps in Eq. 60. Although DPM-Solver (Lu et al., 2022a) introduces some types of discrete time steps, we do not use them in our experiment and just follow the initial settings in Guided-diffusion. We do not use the classifier guidance but evaluate all methods on class-conditional. For experiments involving the LSUN $256 \times 256$ dataset, we use the checkpoint lsun_bedroom.pt. All the methods in our evaluation follow

the linear schedule and uniform time steps in Eq. 60. For experiments involving the ImageNet $256 \times 256$ dataset, we use the diffusion checkpoint 256x256_diffusion.pt and the classifier checkpoint 256x256_classifier.pt. The classifier guidance scale is equal to 2.0. All the methods in our evaluation follow the linear schedule and uniform time steps in Eq. 60.

When the random seed is fixed, Fig.6 - Fig.10 compare the sampling results between stochastic samplers and deterministic samplers on CIFAR-10 $32 \times 32$ (Krizhevsky et al., 2009), FFHQ $64 \times 64$ (Karras et al., 2019), ImageNet $128 \times 128$, ImageNet $256 \times 256$ (Deng et al., 2009) and LSUN $256 \times 256$ (Yu et al., 2015) datasets. As stochastic samplers introduce stochastic noise at each step of the sampling process, the generated images exhibit greater variability, which becomes more pronounced in higher-resolution images. For lower-resolution images, such as CIFAR-10 $32 \times 32$, stochastic samplers may not introduce significant variations within a limited number of steps. However, this does not diminish the value of stochastic samplers, as low-resolution images are becoming less common with the advancements in imaging and display technologies. Furthermore, we also observe that stochastic samplers and deterministic samplers diverge towards different trajectories early in the sampling process, while samplers belonging to the same category exhibit similar patterns of change. Further exploration of stochastic samplers and deterministic samplers is left for future work.

Finally, we also provide samples generated by ER-SDE-Solvers using different pretrained models on various datasets, as illustrated in Fig.11 - Fig.16.

# E EXPERIMENTAL RESULTS ON OTHER METRICS

In the previous sections, we employ FID as our default metric for comprehensive sample quality comparisons due to its ability to capture both diversity and fidelity, and its established position as the standard metric in state-of-the-art generative modeling work (Ho et al., 2020; Song et al., 2021a;b). To further showcase the superiority of ER-SDE-Solvers, we conduct quantitative evaluations using additional metrics, including Inception Score (Salimans et al., 2016), Improved Precision and Recall (Kynkäänniemi et al., 2019), and sFID (Nash et al., 2021).

Followed by Dhariwal & Nichol (2021), we use IS and Precision to measure fidelity, and Recall as well as sFID to measure diversity or distribution coverage, as shown in Table 11 – Table 14. It can be observed that ER-SDE-Solvers consistently perform optimally across all metrics, demonstrating superior fidelity across all stochastic samplers.

Table 11: Sample quality measured by IS↑ on class-conditional ImageNet $128 \times 128$ with the pretrained model Guided-diffusion (without classifier guidance, linear noise schedule), varying the NFE.

| | Sampling method\NFE | 10 | 20 | 30 | 50 |
|---|---|---|---|---|---|
| Stochastic Sampling | DDIM($\eta = 1$) (Song et al., 2021a) | 33.64 | 54.80 | 67.44 | 76.73 |
| | Ours(ER-SDE-Solver-3) | **71.42** | **80.84** | **82.83** | **84.49** |
| Deterministic Sampling | DDIM (Song et al., 2021a) | 57.01 | 70.16 | 76.31 | **79.07** |
| | DPM-Solver-3 (Lu et al., 2022a) | **70.49** | **75.69** | **77.92** | 78.36 |

Table 12: Sample quality measured by Precision↑ on class-conditional ImageNet $128 \times 128$ with the pretrained model Guided-diffusion (without classifier guidance, linear noise schedule), varying the NFE.

| | Sampling method\NFE | 10 | 20 | 30 | 50 |
|---|---|---|---|---|---|
| Stochastic Sampling | DDIM($\eta = 1$) (Song et al., 2021a) | 0.39 | 0.53 | 0.60 | 0.64 |
| | Ours(ER-SDE-Solver-3) | **0.64** | **0.67** | **0.68** | **0.68** |
| Deterministic Sampling | DDIM (Song et al., 2021a) | 0.54 | 0.62 | 0.64 | 0.66 |
| | DPM-Solver-3 (Lu et al., 2022a) | **0.64** | **0.66** | **0.66** | **0.67** |

Table 13: Sample quality measured by Recall↑ on class-conditional ImageNet $128 \times 128$ with the pretrained model Guided-diffusion (without classifier guidance, linear noise schedule), varying the NFE.

| | Sampling method\NFE | 10 | 20 | 30 | 50 |
|---|---|---|---|---|---|
| Stochastic Sampling | DDIM($\eta = 1$) (Song et al., 2021a) | 0.46 | 0.54 | 0.58 | 0.61 |
| | Ours(ER-SDE-Solver-3) | **0.62** | **0.64** | **0.66** | **0.66** |
| Deterministic Sampling | DDIM (Song et al., 2021a) | 0.61 | 0.63 | 0.65 | 0.65 |
| | DPM-Solver-3 (Lu et al., 2022a) | **0.64** | **0.65** | **0.66** | **0.66** |

Table 14: Sample quality measured by sFID↓ on class-conditional ImageNet $128 \times 128$ with the pretrained model Guided-diffusion (without classifier guidance, linear noise schedule), varying the NFE.

| | Sampling method\NFE | 10 | 20 | 30 | 50 |
|---|---|---|---|---|---|
| Stochastic Sampling | DDIM($\eta = 1$) (Song et al., 2021a) | 49.89 | 27.77 | 19.62 | 13.30 |
| | Ours(ER-SDE-Solver-3) | **8.88** | **5.90** | **5.16** | **4.88** |
| Deterministic Sampling | DDIM (Song et al., 2021a) | 14.42 | 7.88 | 6.24 | 5.27 |
| | DPM-Solver-3 (Lu et al., 2022a) | **7.44** | **5.75** | **5.30** | **5.06** |

## F    ADDITIONAL DISCUSSION

**Limitations:**

Despite the promising acceleration capabilities, ER-SDE-Solvers are designed for rapid sampling, which may not be suitable for accelerating the likelihood evaluation of DMs. Furthermore, compared to commonly used GANs (Goodfellow et al., 2014), flow-based generative models (Kingma & Dhariwal, 2018), and techniques like distillation for speeding up sampling (Salimans & Ho, 2022; Song et al., 2023), DMs with ER-SDE-Solvers are still not fast enough for real-time applications.

**Future work:**

This paper introduces a unified framework for DMs, and there are several aspects that merit further investigation in future work. For instance, this paper maintains consistency between the time step mentioned in Proposition 3, 5 and the pretrained model (see Appendix C.1). In fact, many works (Lu et al., 2022a; Jolicoeur-Martineau et al., 2021) have carefully designed time steps tailored to their solvers and achieved good performance. Although our experimental results demonstrate that ER-SDE-Solvers can achieve outstanding performance even without any tricks, further exploration of time step adjustments may potentially enhance the performance. Additionally, this paper only explores some of the noise scale functions for the reverse process, providing examples of excellent performance (such as ER SDE 4 and ER SDE 5). Whether an optimal choice exists for the noise scale function is worth further investigation. Lastly, applying ER-SDE-Solvers to other data modalities, such as speech data (Kong et al., 2020), would be an interesting avenue for future research.

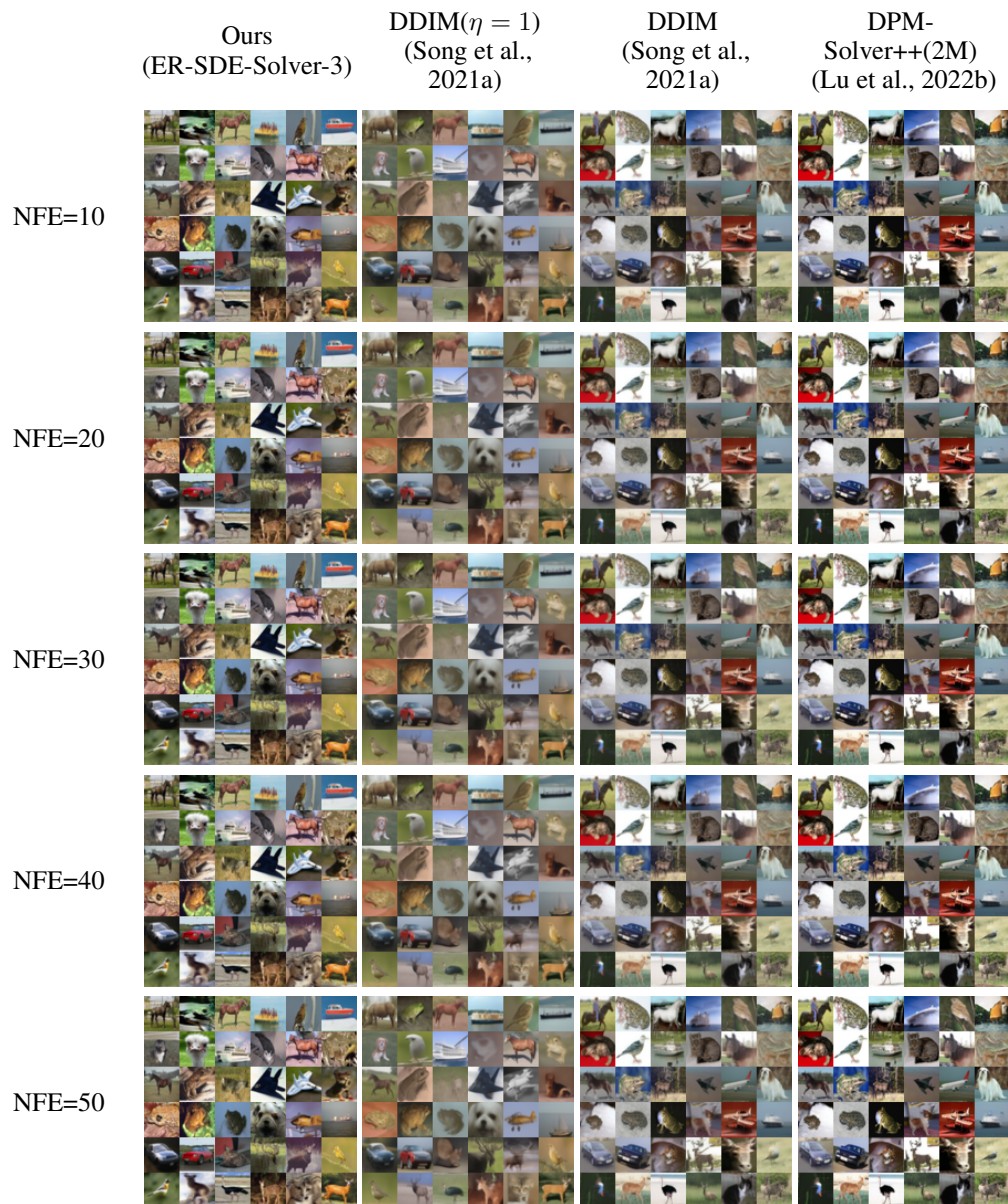

Figure 6: Samples by stochastic samplers (DDIM($\eta = 1$), ER-SDE-Solver-3 (ours)) and deterministic samplers (DDIM, DPM-Solver++(2M)) with 10, 20, 30, 40, 50 number of function evaluations (NFE) with the same random seed (666), using the pretrained EDM (Karras et al., 2022) on CIFAR-10 $32 \times 32$.

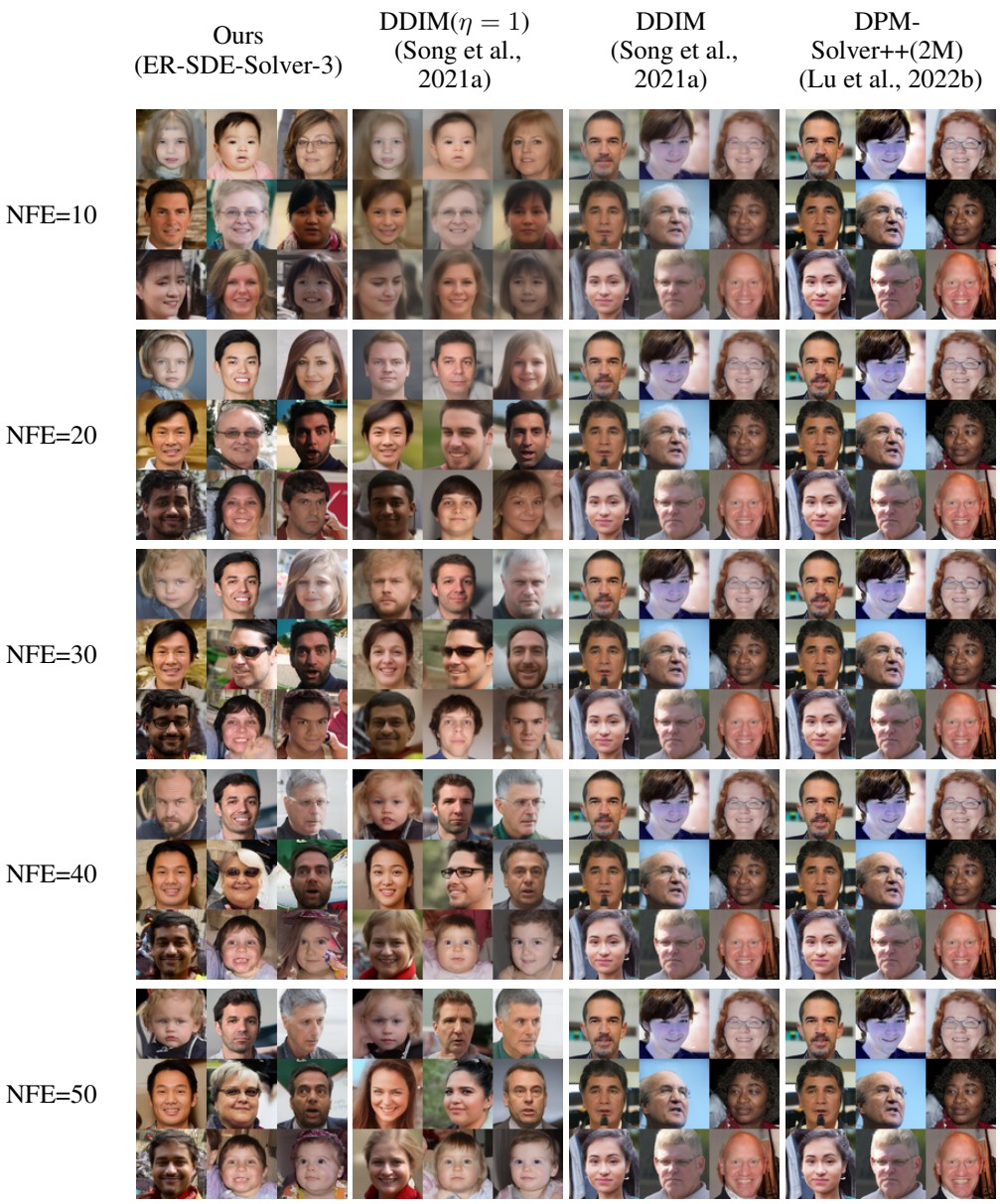

Figure 7: Samples by stochastic samplers (DDIM($\eta = 1$), ER-SDE-Solver-3 (ours)) and deterministic samplers (DDIM, DPM-Solver++(2M)) with 10, 20, 30, 40, 50 number of function evaluations (NFE) with the same random seed (666), using the pretrained EDM (Karras et al., 2022) on FFHQ $64 \times 64$.

DPM-Solver-3 (Lu et al., 2022a)          Ours (ER-SDE-Solver-3)

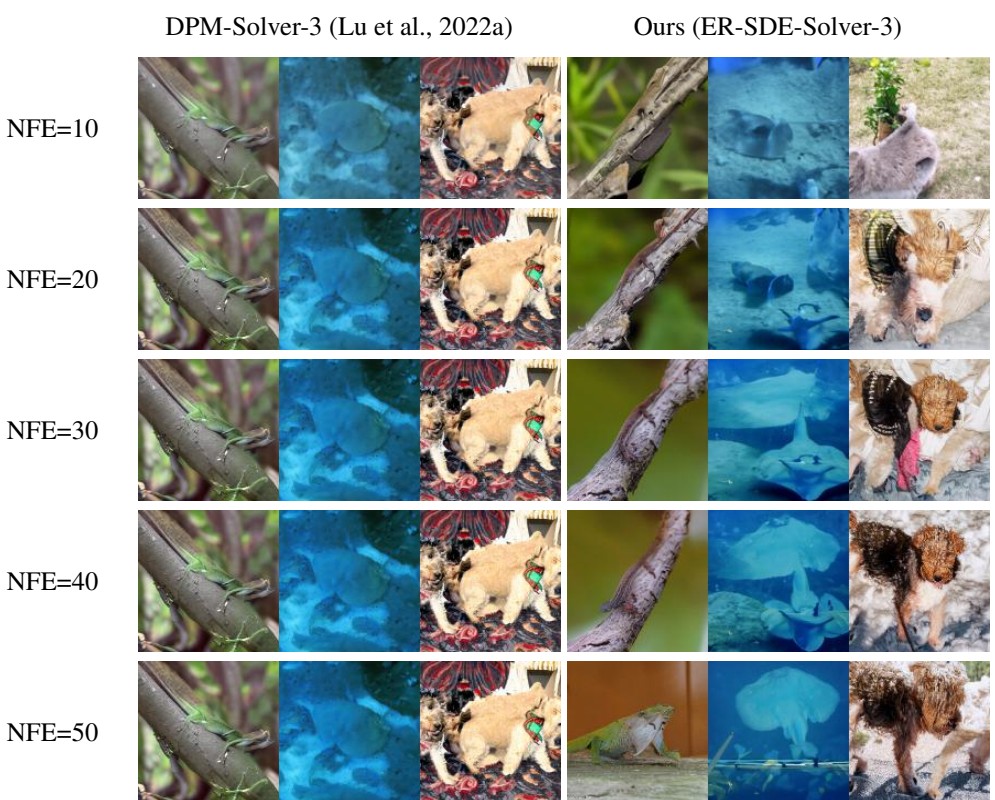

Figure 8:   Samples by stochastic sampler (ER-SDE-Solver-3 (ours)) and deterministic sampler (DPM-Solver-3) with 10, 20, 30, 40, 50 number of function evaluations (NFE) with the same random seed (999), using the pretrained Guided-diffusion (Dhariwal & Nichol, 2021) on ImageNet $128 \times 128$ without classifier guidance.

DPM-Solver-3 (Lu et al., 2022a)      Ours (ER-SDE-Solver-3)

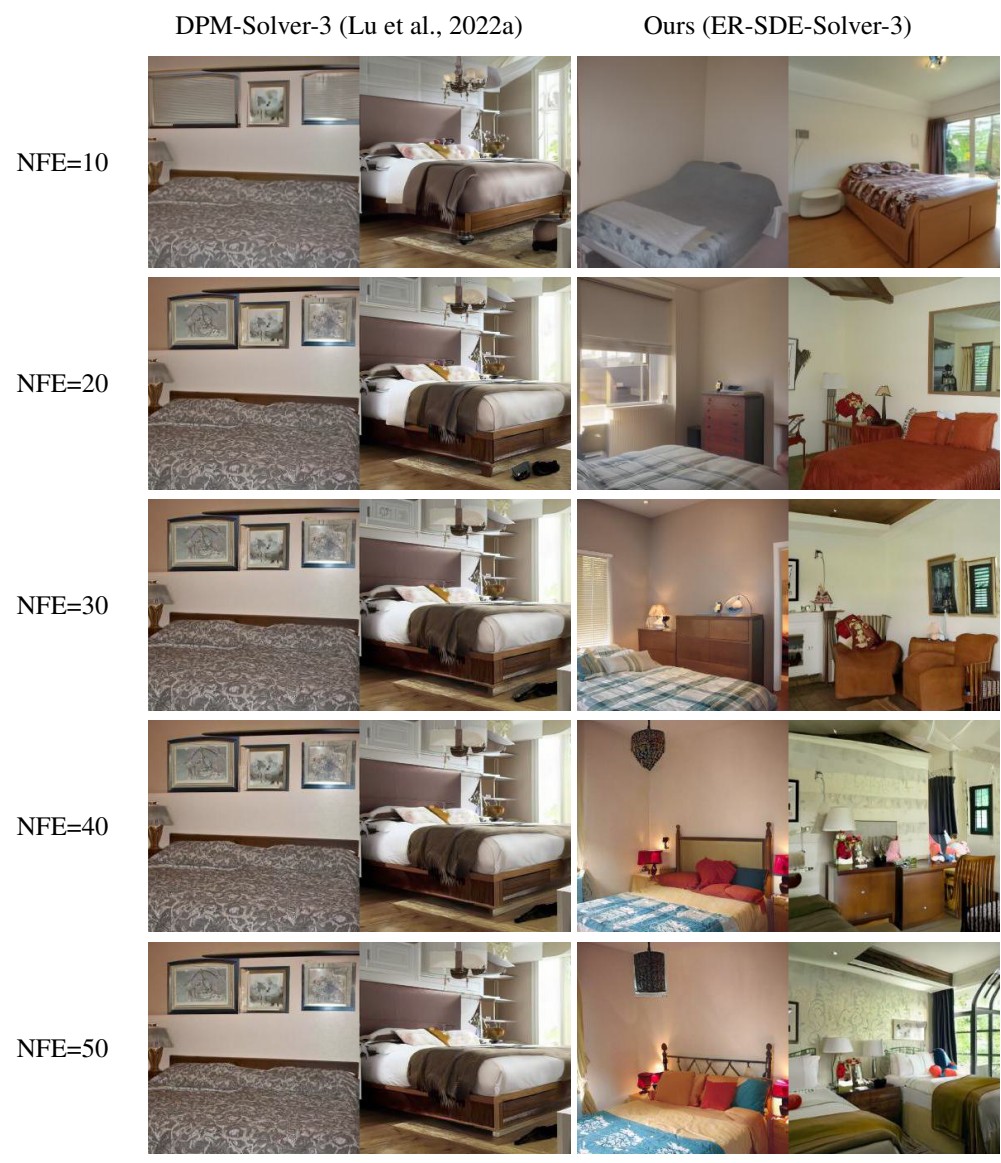

Figure 9: Samples by stochastic sampler (ER-SDE-Solver-3 (ours)) and deterministic sampler (DPM-Solver-3) with 10, 20, 30, 40, 50 number of function evaluations (NFE) with the same random seed (666), using the pretrained Guided-diffusion (Dhariwal & Nichol, 2021) on LSUN Bedrooms $256 \times 256$.

DPM-Solver-3 (Lu et al., 2022a)          Ours (ER-SDE-Solver-3)

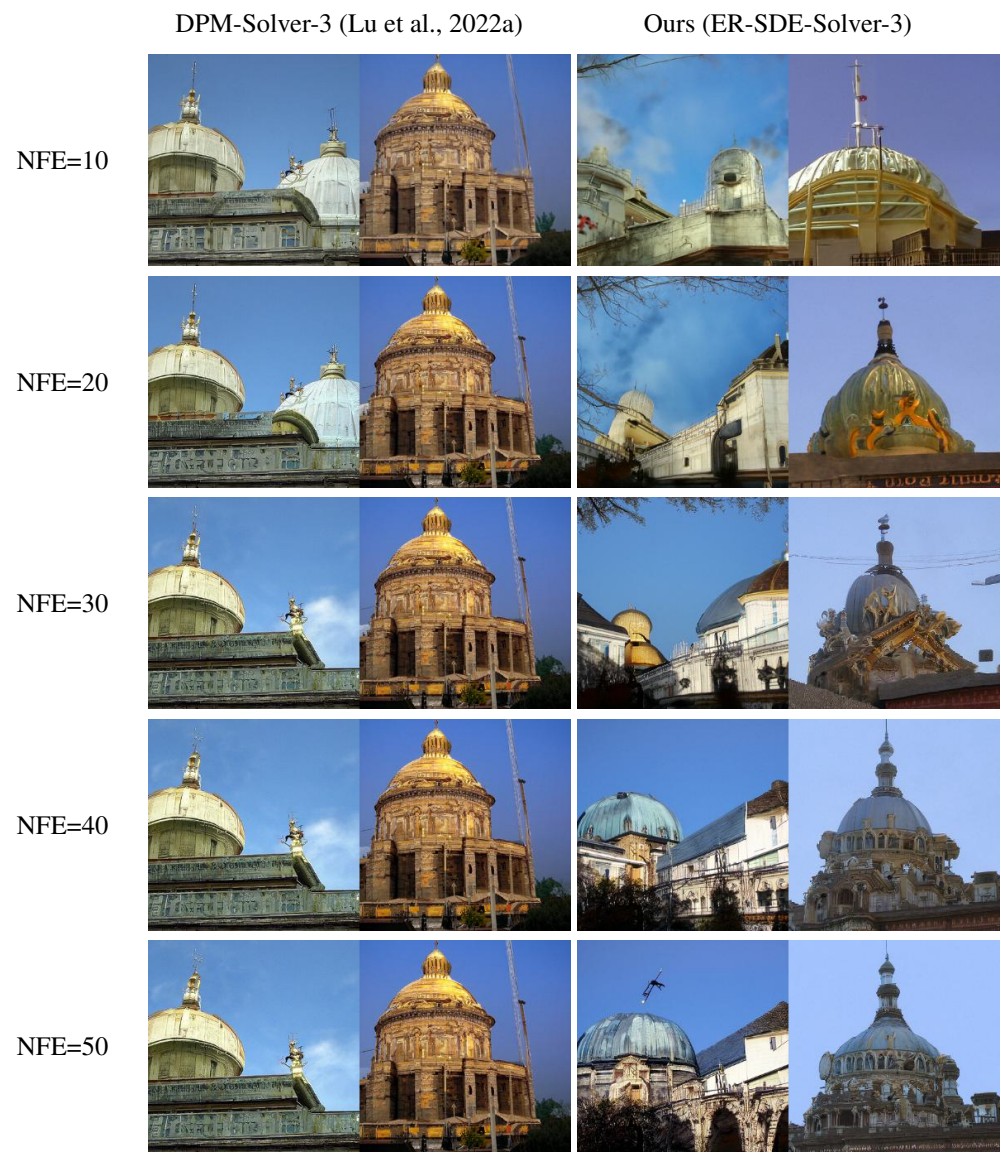

Figure 10:   Samples by stochastic sampler (ER-SDE-Solver-3 (ours)) and deterministic sampler (DPM-Solver-3) with 10, 20, 30, 40, 50 number of function evaluations (NFE) with the same random seed (999), using the pretrained Guided-diffusion (Dhariwal & Nichol, 2021) on ImageNet $256 \times 256$. The class is fixed as dome and classifier guidance scale is 2.0.

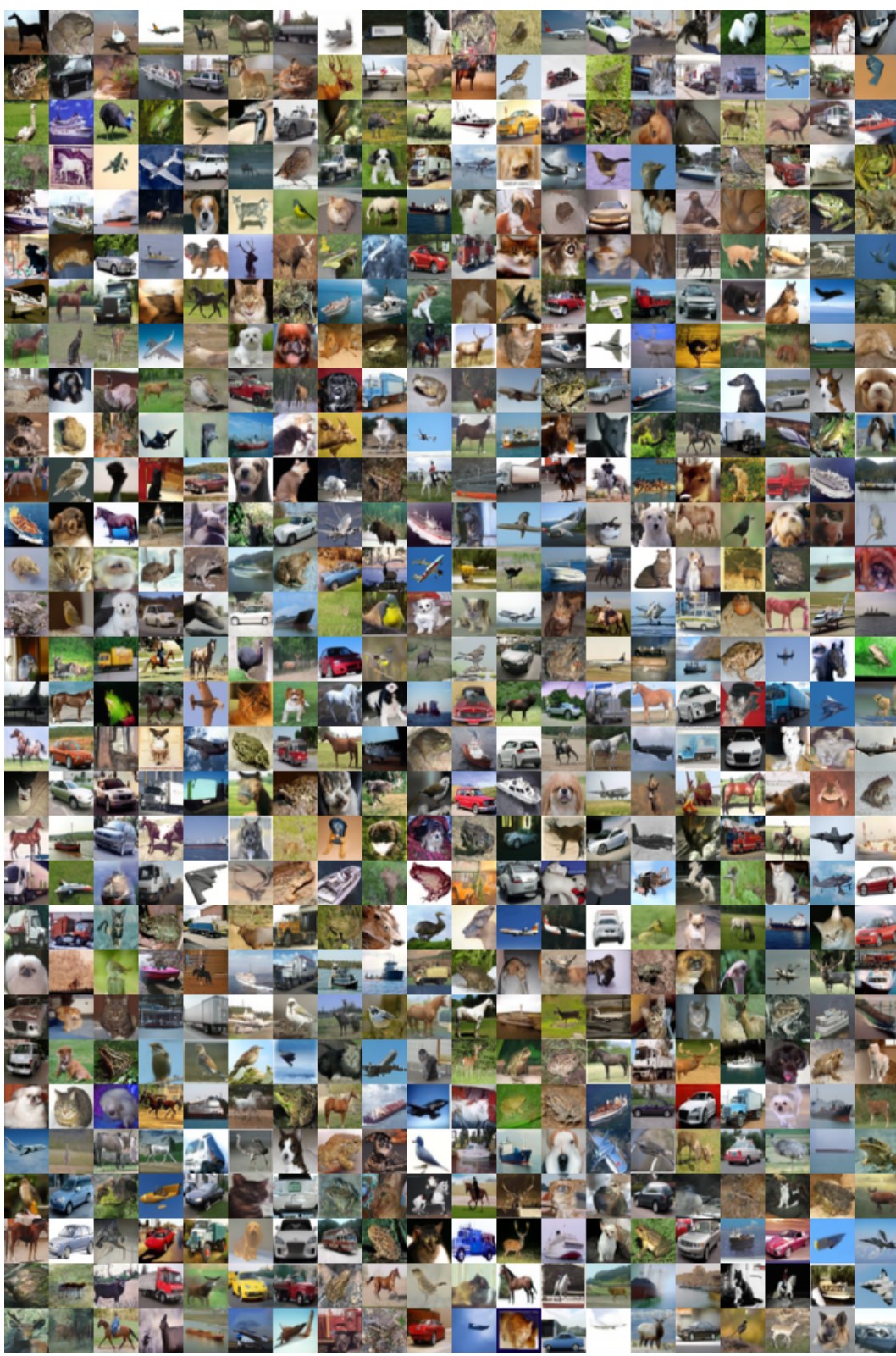

Figure 11: Generated images with ER-SDE-Slover-3 (ours) on CIFAR-10 (NFE=20). The pretrained model is EDM (Karras et al., 2022).

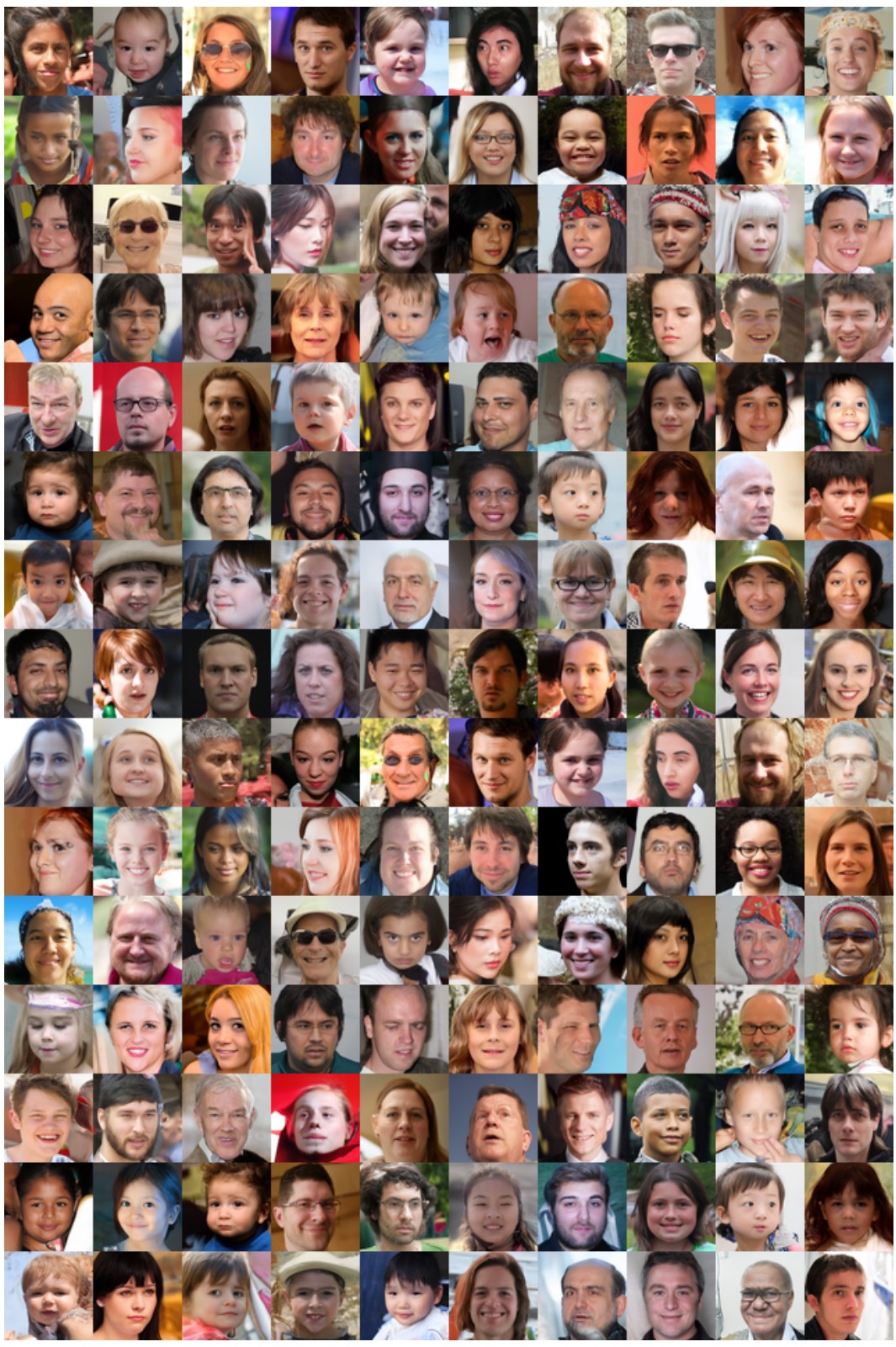

Figure 12: Generated images with ER-SDE-Slover-3 (ours) on FFHQ $64 \times 64$ (NFE=20). The pretrained model is EDM (Karras et al., 2022).

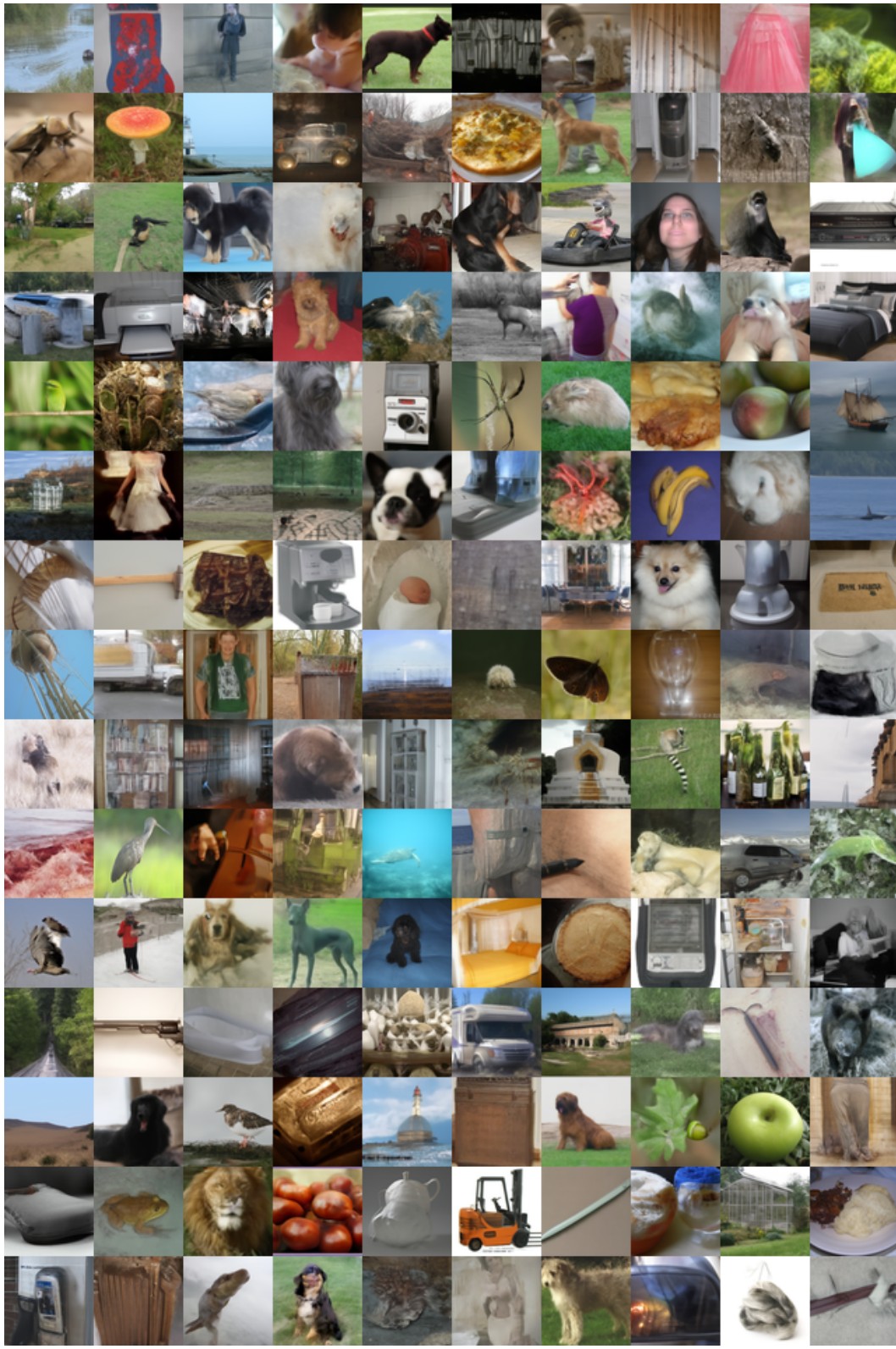

Figure 13: Generated images with ER-SDE-Slover-3 (ours) on ImageNet $64 \times 64$ (NFE=20). The pretrained model is EDM (Karras et al., 2022).

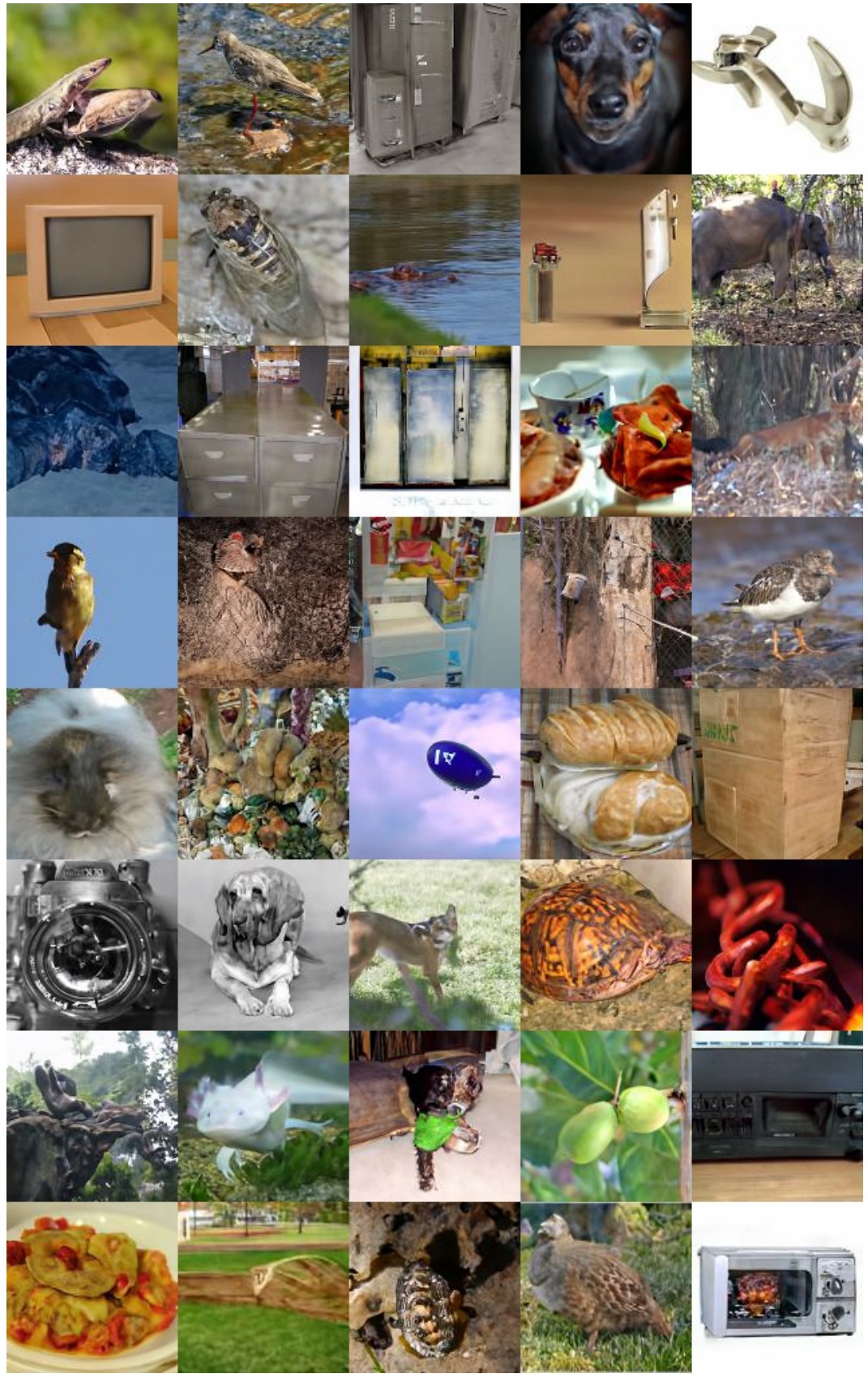

Figure 14: Generated images with ER-SDE-Slover-3 (ours) on ImageNet $128 \times 128$ (NFE=20). The pretrained model is Guided-diffusion (Dhariwal & Nichol, 2021) (without classifier guidance).

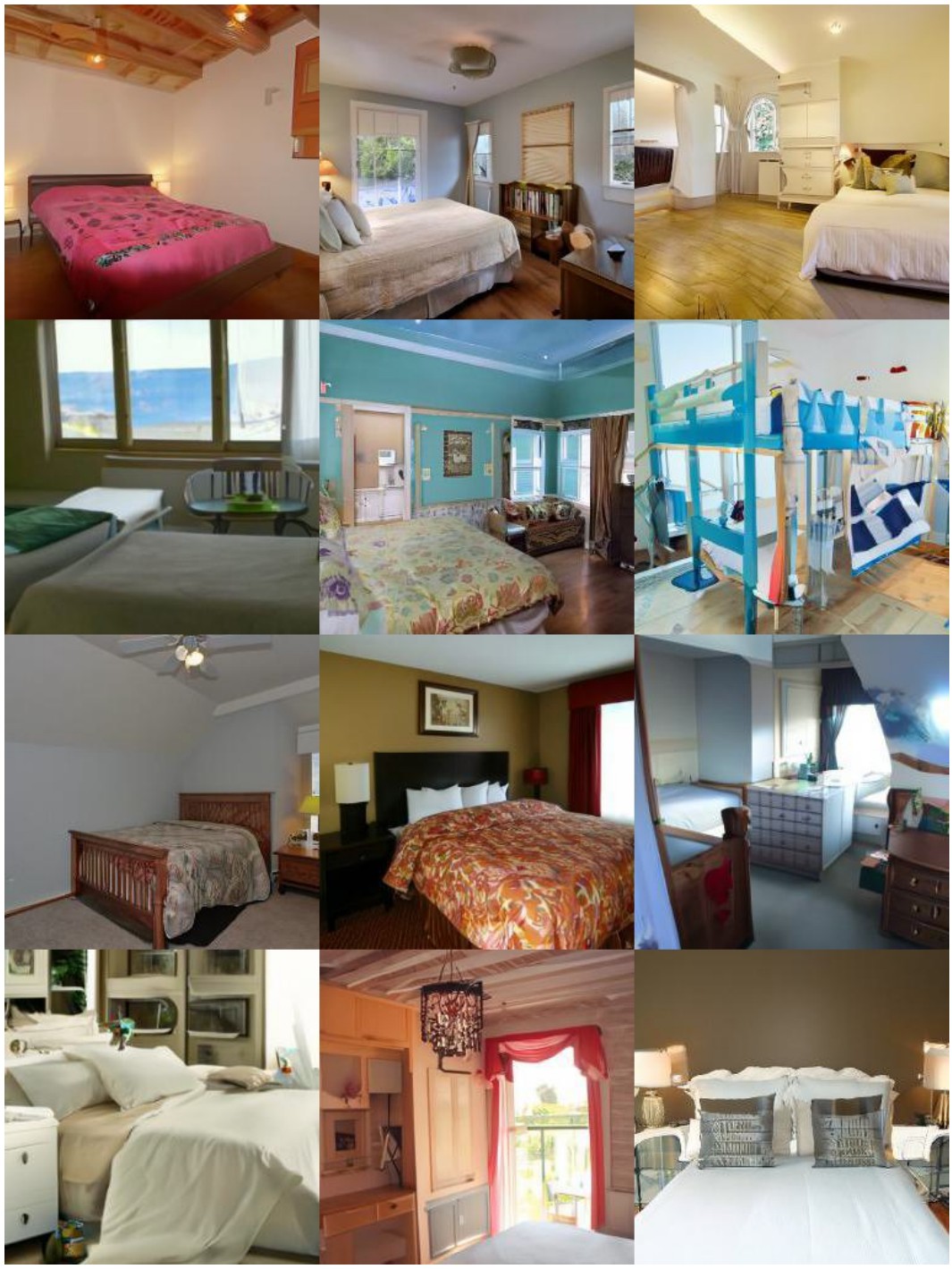

Figure 15: Generated images with ER-SDE-Slover-3 (ours) on LSUN Bedrooms $256 \times 256$ (NFE=20). The pretrained model is Guided-diffusion (Dhariwal & Nichol, 2021).

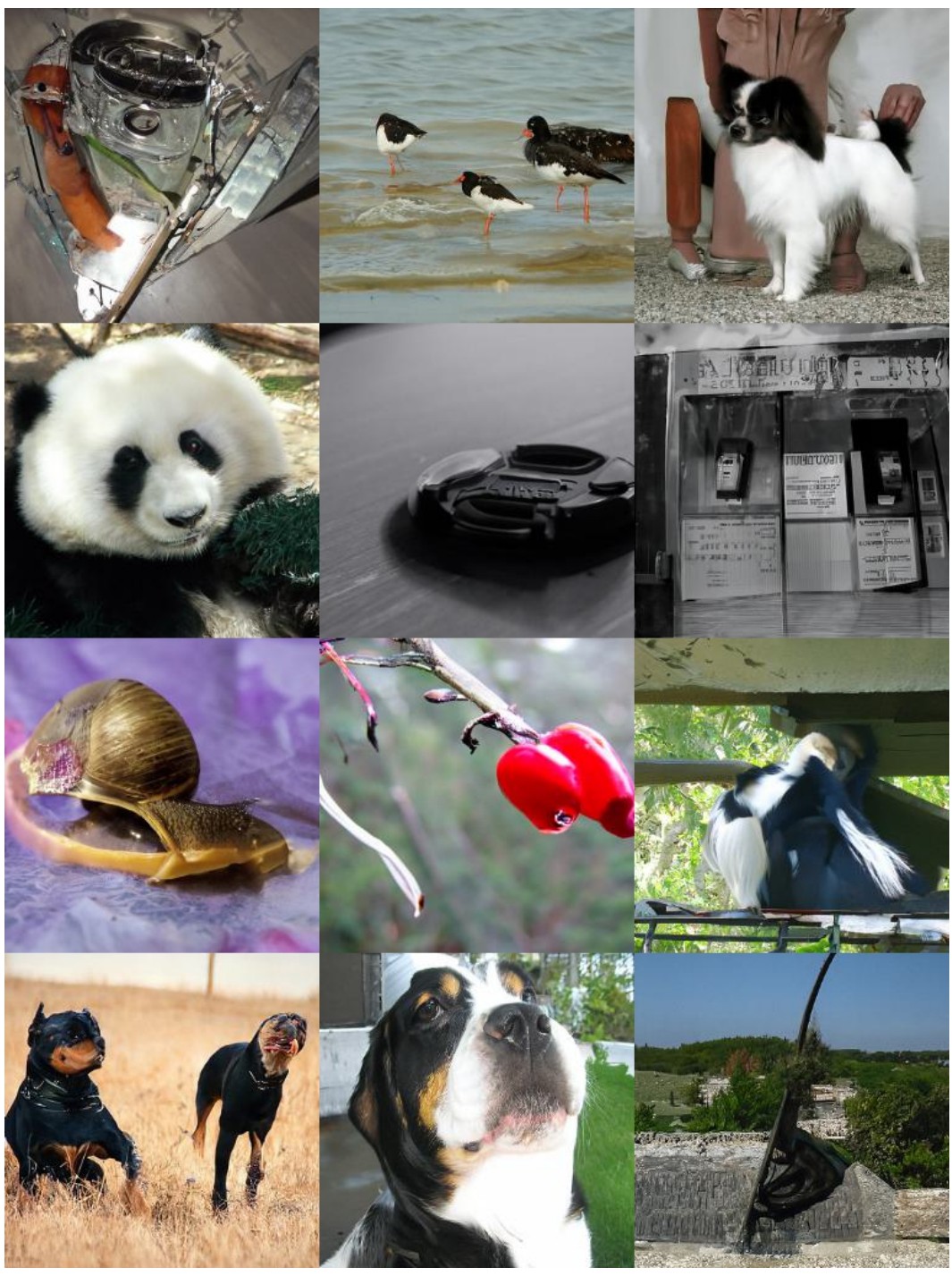

Figure 16: Generated images with ER-SDE-Slover-3 (ours) on ImageNet $256 \times 256$ (NFE=20). The pretrained model is Guided-diffusion (Dhariwal & Nichol, 2021) (classifier guidance scale=2.0).

