# OpenReview forum: "Elucidating the Solution Space of Extended Reverse-Time SDE for Diffusion Models"
_ICLR.cc/2024/Conference — ICLR 2024 Conference Withdrawn Submission_

### Official Review · Reviewer_bLLA · 2023-10-29

**Soundness:** 3 good
**Presentation:** 3 good
**Contribution:** 3 good
**Rating:** 6
**Confidence:** 4

**Summary:**

To speed the sampling in diffusion models, this paper proposes an extended reverse-time SDE (ER SDE). Contrary to the usual order of treatment, they unify prior explorations into ODEs and SDEs, wherein avoiding the lower diversity caused by neural ODEs. Moreover, some mathematical insights is presented to elucidate the fast reason of ODE solvers compared to SDE solvers. Importantly, the experiments show remarkably performance on all stochastic samplers.

**Strengths:**

The proposed method is novel and interesting to speed the sampling in diffusion models, which also avoid the lower diversity caused by ODE solvers. In my humble opinion, the mathematical analysis is rigorous and can support the improvements on the experiment results. Furthermore, the SOTA performance on stochastic samplers contributes to diffusion models to practice applications.

**Weaknesses:**

1.Though it is know that the diversity of generated images will be increased while using SDE solvers, it is better to use some metrics to demonstrate it, such as Inception Score and Precision. I guess it will further demonstrate the superior of ER SDE in community.

2.The $\phi (x)$ is a hyper-parameter, can it be adaptive implement on various diffusion models? since it is just set it manually in this paper.

**Questions:**

The same as Weaknesses.

---

> ### Author Response · Authors · 2023-11-14
> **Reply to your valuable comments**
>
> We extend our sincere appreciation to Reviewer bLLA for the thoughtful consideration of our manuscript and the valuable insights provided. We address each query in detail as outlined below.
> ## **Q1：Use some metrics to demonstrate diversity, such as Inception Score and Precision**
> Thanks for your constructive suggestion! In order to demonstrate diversity, Fig.6 - Fig.10 in Appendix D compare the sampling results between stochastic samplers（ER-SDE-Solvers） and deterministic samplers on various datasets when fixing the random seed. As stochastic samplers introduce stochastic noise at each step of the sampling process, the generated images exhibit greater diversity, which becomes more pronounced in higher-resolution images. \
> In addition to the FID metric already presented in our paper, we appreciate your suggestion and have included additional metrics to further showcase the superiority of ER-SDE-Solvers. Specifically, we have chosen four metrics: **Inception Score, Precision, Recall, and sFID**. The relevant results are provided in Appendix E. **Experimental findings consistently indicate that ER-SDE-Solvers perform optimally across all metrics, highlighting superior diversity and fidelity across all stochastic samplers.**
> ## **Q2：Could $\phi(x)$ be adaptive rather than manual?**
> Thank you for this constructive suggestion! Currently, there lacks a comprehensive theoretical explanation regarding the selection of the noise schedules. Previous studies [1][2][3] have predominantly relied on manually designed noise schedules. [4] argued that the implicit choice for noise schedules enjoys no special properties and the optimal amount of stochasticity should be determined empirically.\
> **Compared to entirely empirical choices, we have taken a significant stride in the study of noise schedule principles.** Our criterion for selecting the noise schedules (noise scale functions) is based on the analysis of the First-order Euler Integral (FEI) coefficient, coupled with relevant experimental observations, rendering it practical (see Appendix A.8). The customizable $\phi(x)$ provides practitioners with a broad spectrum of choices, allowing them to make diverse selections based on specific application scenarios. This flexibility enables a fine balance between the efficiency of sampling and the diversity of generated images.\
> Different noise scale functions $\phi(x)$ correspond to different solutions, collectively forming the solution space of the ER SDE. **This aligns with the focus of this study, which is to elucidate the solution space of the ER SDE**. Therefore, this paper only explores some of the noise scale functions for the reverse process, providing examples of excellent performance (such as ER SDE 4 and ER SDE 5). Of course, whether an optimal and adaptive $\phi(x)$ exists is worth further investigation.
>
> We sincerely hope that our responses align with your expectations, and we would be extremely grateful if these clarifications contribute positively to the assessment of our manuscript.
>
> [1]Denoising diffusion probabilistic models, Ho, J. et al.\
> [2]Improved techniques for training score-based generative models, Song, Y. and Ermon, S.\
> [3] gDDIM: Generalized denoising diffusion implicit models, Zhang et al.\
> [4] Elucidating the design space of diffusion-based generative models, Karras et al.

---

> > ### Author Response · Authors · 2023-11-22
> > **Sincerely looking forward to the further discussions**
> >
> > Dear reviewer,
> >
> > We sincerely hope that our recent response and revisions have successfully addressed your concerns. Your feedback is invaluable to us, and if you find that our efforts have met your expectations, we would be immensely grateful if you could reconsider and possibly elevate the score for our work.
> >
> > If you have any additional questions or suggestions, we would be happy to have further discussions.
> >
> > Best regards,
> >
> > The Authors

---

### Official Review · Reviewer_f7G3 · 2023-10-30

**Soundness:** 3 good
**Presentation:** 3 good
**Contribution:** 2 fair
**Rating:** 5
**Confidence:** 4

**Summary:**

This paper formulates the sampling process as extended reverse-time SDE (ER SDE) for both VP and VE SDE, unifying previous diffusion ODEs and SDEs. Based on ER SDE and its semi-linear structure, the authors derive an analytical solution of ER SDE and then devise training-free SDE samplers. The authors then test their proposed methods on several image-generation experiments.

**Strengths:**

1. The paper is overall well-written and easy to follow.
2. This paper tested several hand-crafted noise schedules $\phi(x)$ and the numerical experiments show improvements compared to other baseline solvers.

**Weaknesses:**

1. The idea of extended reverse-time SDE with noise schedules uncorrelated with $g(t)$ has appeared in several papers, e.g. [1][2]. I think it would be beneficial to discuss them to motivate this paper better.
2. Proposition 3 and Proposition 5 in this paper claim it achieves arbitrary order approximations for SDEs, which is incorrect. The main fault is while deriving approximation error for SDEs, Ito-Taylor expansion rather than Taylor expansion should be employed. The authors may refer to [2][3] for more details.
3. The idea of utilizing the semi-linear structure in reverse diffusion process and analytical solutions has been well-established for ODEs, e.g. [4], and for SDEs, e.g. [2][3][5], which may lower the novelty of this paper.
4. The authors tested on different hand-crafted noise schedules $\phi(x)$. It would be better for authors to do some comprehensive experiments and design theory-motivated noise schedule principles.




[1] Elucidating the design space of diffusion-based generative models, Karras et al.

[2] SA-Solver: Stochastic Adams Solver for Fast Sampling of Diffusion Models, Xue et al.

[3] SEEDS: Exponential SDE Solvers for Fast High-Quality Sampling from Diffusion Models, Gonzalez et al.

[4] DPM-Solver: A Fast ODE Solver for Diffusion Probabilistic Model Sampling in Around 10 Steps, Lu et al.

[5] DPM-Solver++: Fast Solver for Guided Sampling of Diffusion Probabilistic Models, Lu et al.

**Questions:**

I have no other questions, see the weaknesses.

---

> ### Author Response · Authors · 2023-11-12
> **Reply to your valuable comments**
>
> We sincerely appreciate you for the thoughtful consideration and valuable insights.We address each query in detail as outlined below.
> # Q1:Discuss SDE with uncorrelated g(t) in [1,2]
> Thanks for highlighting the relevance of SDE with uncorrelated g(t) in [1,2].Our paper differs in the following aspects:
> 1. Though [2] used $\tau(t)g(t)$ when modeling the reverse process, they took $\tau(t)=\tau$to a constant function for computational convenience in experiments(Sec.6).This implies **[2] essentially introduced a parameter to control the extent of noise.** However, the form of the noise scale is associated with g(t). **In contrast, ER SDE proposes an entirely new noise scale h(t) for the reverse process**, suggesting h(t) may not necessarily correlate with g(t).
> 2. Due to the challenges in solving ER SDE,**[1,2] didn't offer specific implementable solvers. To our knowledge, our paper is the first to offer exact analytical solution for ER SDE.**
> 3. While [1,2] generalized SDE and ODE,as noted by Reviewer yrnV,our paper tackles the ratio of how the solver should work as an SDE or ODE solver.
>
> In the revised version,Sec.3.1 provides discussion with[1,2],significantly enhance the motivation for our paper.
> # Q2:Ito-Taylor expansion rather than Taylor expansion
> Thanks for pointing it out. It is appropriate to use Ito-Taylor expansion when deriving the approximation error for SDE.The key difference between Ito-Taylor and Taylor expansion is the infinitesimal operators. However, we’ve dropped the high-order terms,aligning with common practice in[2,3].
>
> We sincerely apologize for the oversight. It is noteworthy the approximated solutions derived from Ito-Taylor and Taylor expansion are identical after dropping residual terms.As a result, our conclusions remain valid, supported by extensive experiments.Corrections have been made in Appendix A.3&A.5.We deeply appreciate your feedback again.
> # Q3:Utilizing the existing semi-linear structure lowers the novelty
> Though the utilization of the semi-linear structure in analytical solutions has been established for ODE[4] and SDE[2,3,5],distinctions are as follows:
> 1. Equations being solved are different.[4] is based on ODE and[2,5] are based on the original SDE, while we use the semi-linear structure to solve ER SDE. Compared to the former, solving ER SDE is more challenging[2].Although [3] and our paper are contemporaneous, we are open to comparisons. SDE in [3] is $dx_t=[A(t)x_t+b(t)\nabla_x\log p_t(x_t)]dt+g(t)dw_t$, while ER SDE is $dx_t=[f(t)x_t-\frac{g^2(t)+h^2(t)}{2}\nabla_{x}\log p_t(x_t)]dt+h(t)dw_t$.We can see these two SDEs are distinct.
> 2. Using the semi-linear structure to solve differential equations is common, dating back to [6].However,**we aim at elucidating more comprehensive general solutions for SDEs.** Based on it,we find the superior performance of ODE solvers for fast sampling over SDE solvers, and the equal performance of VP solvers with VE solvers.
>
> We hope our response clearly articulates the unique contributions of our paper,addressing your concerns effectively.
> # Q4:Design theory-motivated noise schedule principles
> Thanks for constructive suggestion!Now, there lacks theoretical explanations for selecting noise schedules.Previous studies[7,8,9] have mainly relied on manually designed noise schedules.[1] argued the choice for noise schedules enjoys no special properties and the optimal amount of stochasticity should be determined empirically.\
> **Compared to purely empirical choices,we've taken a significant stride in the study of noise schedule principles.** Our criterion for selecting noise schedules is based on FEI coefficient and experimental observations, which is practical.Customizable $\phi(x)$ offers practitioners diverse options,enabling a balance between sampling efficiency and image diversity tailored to specific application scenarios.\
> Different noise schedules $\phi(x)$ correspond to different solutions, collectively forming the solution space of the ER SDE,**which aligns with the focus of our paper: elucidating the solution space of ER SDE.** Therefore,we only explore some of $\phi(x)$ for the reverse process (ER SDE 4&5).The optimal $\phi(x)$ is worth further investigation.\
> We hope our responses meet your expectations,and we'd be grateful if these clarifications positively contribute to assessing our paper.
>
> [1]Elucidating the design space of diffusion-based generative models.\
> [2]SA-Solver:Stochastic Adams Solver for Fast Sampling of Diffusion Models.\
> [3]SEEDS:Exponential SDE Solvers for Fast High-Quality Sampling from Diffusion.\
> [4]DPM-Solver:A Fast ODE Solver for Diffusion Probabilistic Model Sampling in Around 10 Steps.\
> [5]DPM-Solver++:Fast Solver for Guided Sampling of Diffusion Probabilistic Models.\
> [6]Numerical Solution of Stochastic Differential Equations.\
> [7]Denoising diffusion probabilistic models.\
> [8]Improved techniques for training score-based generative models.\
> [9]gDDIM:Generalized denoising diffusion implicit models.

---

> > ### Comment · Reviewer_f7G3 · 2023-11-21
> >
> > Thank you for the detailed response. The authors' response addressed most of my issues. However, I still have questions about the convergence order of the proposed algorithm. In the revised version of the paper, Proposition 3 and Proposition 5 claim that the proposed scheme achieves arbitrary-order approximations for the SDE, which I highly disagree with. In the proof of Proposition 3 and 5, which is Appendix A.3 and A.5, the omitted term $R_k$ is of order $h^{\frac{3}{2}}$ for which the proposed scheme has a global error of at least $O(h)$. I will consider raising my score if the authors could revise it.

---

> > > ### Author Response · Authors · 2023-11-21
> > > **Reply to your valuable comments**
> > >
> > > Thank you for pointing it out!  We have carefully reviewed [1], and there is indeed an error in the claims made in Proposition 3 and Proposition 5 regarding the proposed scheme achieving arbitrary-order approximations for the SDE. We sincerely apologize for this mistake.
> > >
> > >
> > > In the latest revised version, we refer to the case where $k = 1$ as a "first-order solver" because the omitted term $R_k$ is of order $h^{\frac{3}{2}}$ for which the proposed scheme has a global error of at least $O(h)$. When $k ≥ 2$, we designate it as a kth-stage solver in accordance with the statement in [1].
> > >
> > >
> > > Thank you again for your time and thorough review. We would be extremely grateful if these clarifications contribute positively to the assessment of our manuscript.
> > >
> > >
> > > [1] SEEDS: Exponential SDE Solvers for Fast High-Quality Sampling from Diffusion Models, Gonzalez et al.

---

> > > > ### Comment · Reviewer_f7G3 · 2023-11-21
> > > >
> > > > Thanks for your response. I will raise my score to 5.

---

> > > > > ### Author Response · Authors · 2023-11-22
> > > > > **Thank you for raising the score!**
> > > > >
> > > > > Thank you so much for raising the score. We believe that the practical application of ER-SDE-Solver will have a high impact on the community, and the elucidation of  ER SDE solution space will inspire people to continue exploring the selection of optimal noise scale functions.

---

### Official Review · Reviewer_RiM8 · 2023-11-01

**Soundness:** 2 fair
**Presentation:** 2 fair
**Contribution:** 2 fair
**Rating:** 3
**Confidence:** 4

**Summary:**

This paper proposes Extended Reverse-time SDE (ER-SDE) to model the sampling process of diffusion models, which can unify ODE-based and SDE-based sampling. Based on the approximation of the exact solution of ER-SDE, the authors propose ER-SDE-solver, a fast stochastic sampler for diffusion models. The experimental results on various datasets show that ER-SDE-solver achieves great sample quality within 20-50 NFE.

**Strengths:**

1. The proposed method is clear and complete.
2. RE-SDE-solver consistently outperforms other stochastic samplers within 50 NFE on different models and various datasets.
3. Extensive ablation studies are provided to understand the design component of ER-SDE-solver.

**Weaknesses:**

1. The discretization error discussed in Section 4.1 needs further clarification. If I'm not mistaken, the discretization error is defined to be the remainder of the k-th order Taylor approximation. I do not see how the remainder is related to FEI and how you can control FEI to reduce the error.
2. ER-SDE-solver needs to tune the hyperparameter for N-point numerical integration and design the noise scale function by monitoring the FID metric, which could be tricky to find the right balance for different models in practice.
3. Section 4.2 discusses the results of ER SDE 4, ER SDE 5, and ODE. However, it is difficult to see the difference from Figure 3 (b).
4. What is the order of the algorithm used to report Figure 3? The basic detail about the experiments in Figure 3 is missing in Section 4.2.
5. The claim of outperforming all other stochastic samplers is not well supported. For example, EDM sampler [1] can achieve FID 1.55 on ImageNet64 but the best FID reported in this paper is 2.24.
6. Typo: in the last second paragraph of introduction, "... theoretically establish that the VP SDE solves yield image quality .. " -> "theoretically establish that the VP SDE solvers yield image ..."

[1]: Karras, Tero, Miika Aittala, Timo Aila, and Samuli Laine. "Elucidating the design space of diffusion-based generative models." _Advances in Neural Information Processing Systems_ 35 (2022): 26565-26577.

**Questions:**

1. Regarding Table 2, can ER-SDE-solver further improve sample quality by increasing NFE? For example, in Figure 4(c) of EDM paper [1], EDM-solver can achieve FID 1.55 with more NFE.

---

> ### Author Response · Authors · 2023-11-14
> **Reply to your valuable comments[1/2]**
>
> We extend our sincere appreciation to Reviewer RiM8 for the thoughtful consideration of our manuscript and the valuable insights provided. We address each query in detail as outlined below.
> ## **Q1：The discretization error discussed in Sec.4.1 needs further clarification**
> We sincerely apologize for confusion! Sec.3 demonstrates that the exact solution of the ER SDE comprises three components: a linear function of the data variables, a non-linear function parameterized by neural networks and a noise term. The linear and noise terms can be precisely computed, while discretization errors are present in the non-linear term. Due to the decreasing error as the order increases (see Table1), the first-order error predominantly influences the overall error. Therefore, we exemplify the case with order k = 1 for error analysis.\
> It can be observed from Eq.(16) and Eq. (17) that the error arises from the second term$\Big{[}1-\frac{\phi(\sigma_{t_{i}})}{\phi(\sigma_{t_{i-1}})}\Big{]} x_\theta(\tilde{x} _ {\sigma_{t_{i-1}}},\sigma_{t_{i-1}})$, as $x_\theta(\tilde{x} _ {\sigma_{t_{i-1}}}, \sigma_{t_{i-1}})$ is a neural network trained in the forward process to estimate data, whose precision is fixed. To reduce discretization error, it is necessary to minimize the coefficient $1 - \frac{\phi(x_t)}{\phi(x_s)}$, referring as the First-order Euler Integral (FEI) coefficient. **Therefore, when using the same model weights, a lower FEI coefficient leads to smaller discretization error. In turn, the images generated in the reverse process more similar to real images in the forward process, resulting in higher image quality and a smaller FID**. We hope our explanation addresses your concerns.
> ## **Q2：It is tricky to find N-point and noise scale function for different models in practice**
> Thank you for your thorough review! We conducted ablation experiments in Appendix C.3 to investigate the choice of the numerical integration points \(N\). To strike a balance between efficiency and performance, we fixed \(N = 100\) for all experiments in our paper. Although the ablation experiments were conducted on CIFAR-10 dataset using the pretrained EDM, our experimental results on other datasets such as FFHQ, ImageNet, and LUSN, using Guided-diffusion as the pretrained model, indicate that the chosen integration point \(N=100\) still performs well. **Therefore, practitioners do not need to excessively adjust the number of numerical integration points in practice**.\
> Regarding the noise scale function, our experiments were conducted using the pretrained EDM on CIFAR-10 dataset (See Fig.3). Extensive experimental results indicate that our designed ER SDE 5 can generalize to all datasets and pretrained diffusion models. However, we still provide practitioners with the opportunity to customize the noise scale function according to specific application scenarios. **Compared to entirely empirical choices [1][2][3], we have taken a significant stride in the study of noise schedule principles**.
> Different noise scale functions $\phi(x)$ correspond to different solutions, collectively forming the solution space of the ER SDE. **This aligns with the focus of our study, which is to elucidate the solution space of the ER SDE**. Therefore, we only explores some of the noise scale functions for the reverse process, providing examples of excellent performance (such as ER SDE 4 and ER SDE 5). Of course, whether an optimal and adaptive $\phi(x)$ exists is worth further investigation.
> ## **Q3：It is difficult to see the difference from Fig. 3 (b)**
> We apologize for the lack of clarity in the figures. We have provided enlarged details for Fig. 3(a), Fig. 3(b), Fig. 5(a) and Fig. 5(b) in the revised version. As observed, the ODE solver exhibits minimal discretization error. ER SDE 4 exhibits discretization error that closely adheres to the behavior of the ODE. ER SDE 5 demonstrates elevated error in the initial 100 steps and gradually converges to the ODE’s error profile. Both ER SDE 4 and ER SDE 5 exhibit comparable efficiency to the optimal ODE solver. Image quality deteriorates for ill-suited noise scale functions (like ER SDE 2).
>
>
> We sincerely hope that our responses align with your expectations, and we would be extremely grateful if these clarifications contribute positively to the assessment of our manuscript.
>
> [1]Denoising diffusion probabilistic models, Ho, J. et al.\
> [2]Improved techniques for training score-based generative models, Song, Y. and Ermon, S.\
> [3] gDDIM: Generalized denoising diffusion implicit models, Zhang et al.

---

> > ### Comment · Reviewer_RiM8 · 2023-11-21
> >
> > Thanks for adding details to the figures.
> >
> > Regarding the discretization error, it seems you are discussing a different error than the one in my review. Can you precisely define what you mean by discretization error? I think what really matters here is the local error $\tilde{\boldsymbol{x}}_t - \boldsymbol{x}_t$, which will accumulate over time. I still don't see how FEI can control the error.
> >
> > Thanks for your elaboration on N-point integration and noise scale function. Assume your claim about the relation between FEI and the discretization error holds. Why does ER-SDE 5 have better results than ER-SDE 4 on CIFAR10 and ImageNet64?
> >
> > Another point that confuses the reviewer is that the authors claim that ODE is the optimal solution of ER SDE as it minimizes the FEI, as shown in figure (b). Then why not just do ODE instead of ER SDE? The experimental results also show that the proposed ER SDE solvers do not consistently outperform ODE sovler.

---

> > > ### Comment · Reviewer_RiM8 · 2023-11-21
> > >
> > > BTW, what do you mean by Eq (9) is linear in the narrow sense? $\boldsymbol{x}_{\theta}$ is nonlinear so Eq(9) is a nonlinear SDE, right?

---

> ### Author Response · Authors · 2023-11-14
> **Reply to your valuable comments[2/2]**
>
> We extend our sincere appreciation to Reviewer RiM8 for the thoughtful consideration of our manuscript and the valuable insights provided. We address each query in detail as outlined below.
> ## **Q4：What is the order of the algorithm used to report Fig.3?**
> We sincerely apologize for confusion. In the experiments depicted in Fig. 3, we utilize the first-order ER SDE Solvers. The relevant experimental details have been supplemented in the legend of Fig. 3. Thank you for pointing it out!
> ## **Q5：The claim of outperforming all other stochastic samplers is not well supported？Can ER-SDE-solver further improve sample quality by increasing NFE?**
> Thanks for your question! The purpose of our paper is to accelerate sampling for diffusion models, so we limit our experiments to a finite number of steps (<=50 NFE), which is consistent with many works in the field [4][5][6]. For a fair comparison, FID metrics should be compared under the condition of fixed NFE. When NFE=50, ER SDE 5 achieves 2.24 FID on ImageNet64, while EDM-Stochastic only achieves 2.49. Therefore, **we claim that ER-SDE-Solvers achieve state-of-the-art performance across all stochastic samplers in the context of rapid sampling**.\
> In fact, ER-SDE-Solver can enhance sample quality by increasing NFE. Experimental findings indicate that judiciously introducing additional stochastic noise can effectively rectify errors in the early sampling steps, thereby generating high-quality samples in prolonged sampling scenarios. However, blindly increasing NFE for better FID might not be practical in real-world applications. Therefore, **our paper focuses on rapid sampling rather than achieving absolute high-quality samples**. Leveraging ER-SDE-Solvers for high-quality sampling can be considered for future work.
> ## **Q6：Spelling correction**
> We sincerely apologize for the spelling error. We have rectified it in the revised version. Thank you for pointing it out!
>
> We sincerely hope that our responses align with your expectations, and we would be extremely grateful if these clarifications contribute positively to the assessment of our manuscript.
>
> [4] Fast sampling of diffusion models with exponential integrator, Zhang et al.\
> [5] DPM-Solver: A Fast ODE Solver for Diffusion Probabilistic Model Sampling in Around 10 Steps, Lu et al.\
> [6] DPM-Solver++: Fast Solver for Guided Sampling of Diffusion Probabilistic Models, Lu et al.

---

> ### Author Response · Authors · 2023-11-22
> **Reply to your valuable comments**
>
> We extend our sincere appreciation for your valuable insights. We address each query in detail as outlined below.
> ## **Q1：Precisely define discretization error**
> **The discretization error as defined by us does not refer to the local error $\tilde{x}_t-x_t$, but pertains to the error introduced when discretizing the nonlinear integral term**  $\int_{\sigma_t}^{\sigma_s}\frac{\phi^{(1)}(\sigma)}{\phi^2(\sigma)}x_\theta(x_\sigma,\sigma)d \sigma$. The discretization error we define directly influences the local error, as Section 3 demonstrates that the exact solution of ER SDE consists of three components: a linear function of the data variables(1st term), a non-linear function parameterized by neural networks(2nd term), and a noise term(3rd term), i.e.,
> $$
> x_t=\frac{\phi(\sigma_t)}{\phi(\sigma_s)}x_s+\phi(\sigma_t) \int_{\sigma_t}^{\sigma_s}\frac{\phi^{(1)}(\sigma)}{\phi^2(\sigma)}x_\theta(x_\sigma, \sigma)d \sigma+\sqrt{\sigma_t^2-\sigma_s^2[\frac{\phi(\sigma_t)}{\phi(\sigma_s)}]^2 }z_s
> $$
> **The linear and noise terms can be precisely computed, while local errors are present in the non-linear term**. Due to the decreasing error as the order increases (Table1), the first-order error predominantly influences the overall error. Therefore, we exemplify the case with order k = 1 for error analysis.
> It can be observed that the discretization error arises from the second term is a neural network trained in the forward process to estimate data, whose precision is fixed. To reduce discretization error, it is necessary to minimize the coefficient $1 - \frac{\phi(x_t)}{\phi(x_s)}$, referring as the First-order Euler Integral (FEI) coefficient. **Therefore, when using the same model weights, a lower FEI coefficient leads to smaller discretization error**. We hope our explanation addresses your concerns.
> ## **Q2：Why does ER-SDE 5 have better results than ER-SDE 4 on CIFAR10 and ImageNet64?**
> Thanks for your question! In fact, **the sampling performance of ER-SDE 5 is inferior to that of ER-SDE 4 when NFE is relatively small**. Table 4 and Table 5 in Appendix D present the image generation quality of different noise scale functions on ImageNet 64 × 64 and CIFAR-10. We cite here to demonstrate clearly:
>
> Table 4: Sample quality measured by FID$\downarrow$ on ImageNet $64\times64$ for different noise scale functions with VE ER-SDE-Solver-3.
> |Noise scale functions\NFE|10|15|20|25|30|40|50|
> |----------------------|----|----|----|----|----|----|----|
> |ER SDE 4|**10.79**|**4.99**|**3.45**|2.96|2.71|2.43|2.38|
> |ER SDE 5|11.46|5.08|**3.45**|**2.92**|**2.58**|**2.37**|**2.24**|
>
> Table 5: Sample quality measured by FID$\downarrow$ on CIFAR-10 for different noise scale functions with VE ER-SDE-Solver-3.
>
> |Noise scale functions\NFE|10|15|20|25|30|40|50|
> |----------------------|----|----|----|----|----|----|----|
> |ER SDE 4|**9.28**|**4.37**|**3.01**|**2.52**|**2.25**|**2.07**|**1.97**|
> |ER SDE 5|9.86|4.57|3.09|2.54|2.29|2.10|**1.97**|
>
> Due to the discretization errors of ER-SDE 4 being sufficiently close to the minimal error (see Fig. 3), it exhibits outstanding FID scores on CIFAR-10 and ImageNet 64×64 (NFE≤20). This result once again confirms the validity of the relationship between FEI and discretization error.
> In the experiments on ImageNet 64×64, despite the initial significant discretization error in ER-SDE 5, its later-generated image quality (NFE≥20) is comparable to ER-SDE 4. This is attributed to the fact that ER SDE involves implicit Langevin diffusion, which can effectively correct any errors in the early sampling steps [1]. Consequently, the early errors in ER-SDE 5 are rectified in the later stages.
> ## **Q3：Why not just do ODE instead of ER SDE?**
>
> Thanks for your insightful question! As highlighted in Sec.1, **SDE-based stochastic samplers exhibit an enhanced capability to generate diverse images by injecting additional noise into the data state at each sampling step**. This conclusion is further supported by Fig.6-Fig.10 in Appendix D, where stochastic samplers demonstrate a greater variability in generated images when the random seed is fixed. This advantageous feature becomes particularly prominent in the context of high-resolution image generation, effectively meeting the demands of industries such as art creation and game development.\
> In addition, **stochastic samplers with classifier guidance exhibit superior performance in rapid sampling compared to deterministic samplers(see Table 3)**.
> This is attributed to the customized noise injected into the sampling process, which mitigates the inaccuracies in data estimation introduced by classifier gradient guidance.
>
> We sincerely hope that our responses align with your expectations, and we would be extremely grateful if these clarifications contribute positively to the assessment of our manuscript. Thank you for your time and consideration.\
> [1] Elucidating the design space of diffusion-based generative models, Karras et al.

---

> ### Author Response · Authors · 2023-11-22
> **Reply to your valuable comments**
>
> We extend our sincere appreciation for your valuable insights. We address each query in detail as outlined below.
>
> ## **Q4：Why Eq.(9) is linear in the narrow sense?**
>
> Thanks for your question! Although $x_\theta$ is a nonlinear neural network, it is pre-trained during the forward process. **Eq.(9) represents a stochastic differential equation with the data state $x_t$ in the reverse process as the variable**. According to the discussion in Section 4.2 of [2] regarding Eq.(2.3), we refer to Eq.(9) as linear in the narrow sense.
>
> We sincerely hope that our responses align with your expectations, and we would be extremely grateful if these clarifications contribute positively to the assessment of our manuscript. Thank you for your time and consideration.
>
> [2] Numerical Solution of Stochastic Differential Equations, Peter E Kloeden and Eckhard Platen.

---

> > ### Comment · Reviewer_RiM8 · 2023-11-22
> >
> > Thanks for your response but I disagree. The linear SDE in the narrow sense defined in Section 4.2 of [2] is
> > $$\mathrm{d}X_t = \left(a_1(t)X_t+a_2(t)\right)\mathrm{d}t + b_2(t)\mathrm{d}W_t. $$
> > However, the equation (9) is
> > $$\mathrm{d}X_\sigma = a_1(\sigma)\left(X_\sigma - X_{\theta}(X_\sigma)\right)\mathrm{d}\sigma+b_2(\sigma)\mathrm{d}W_\sigma, $$
> > which is clearly nonlinear as the function $X_{\theta}(X_\sigma)$ is a nonlinear function of $X_{\sigma}$. The nonlinearity of $X_{\theta}$ has nothing to do with whether it is pre-trained.
> >
> > [2] Numerical Solution of Stochastic Differential Equations, Peter E Kloeden and Eckhard Platen.

---

> > > ### Author Response · Authors · 2023-11-22
> > > **Reply to your valuable comments**
> > >
> > > We extend our sincere appreciation for your valuable insights. We address your query in detail as outlined below.
> > > ## **Q: Eq.(9) is nonlinear instead of linear in the narrow sense**
> > > Thank you for pointing it out! We have carefully reviewed [1], and we acknowledge that Eq.(9) is indeed nonlinear instead of linear in the narrow sense. We sincerely apologize for this mistake.
> > > In fact, when solving Eq.(9), we employ the variation-of-constants formula followed [2][3]. Therefore, our solution is accurate. In the latest revised version, we have rectified inaccuracies in our statements. Thanks for your thorough review again!
> > >
> > > [1] Numerical Solution of Stochastic Differential Equations, Peter E Kloeden and Eckhard Platen.\
> > > [2] DPM-Solver++: Fast Solver for Guided Sampling of Diffusion Probabilistic Models, Lu et al.\
> > > [3] SEEDS: Exponential SDE Solvers for Fast High-Quality Sampling from Diffusion Models, Gonzalez et al

---

> ### Comment · Reviewer_RiM8 · 2023-11-22
>
> Thanks for your detailed reply.
>
> R1: Could you provide a precise mathematical definition of the discretization error discussed in the paper and clearly show how FEI controls the discretization error? The current response seems to repeat what you wrote in the paper.
>
> R3: Thanks for referring to the Fig.6-Fig.10. However, these are just some generated samples. A more quantitative comparison of sample diversity is needed if the focus is to enhance diversity. For example, report Recall on ImageNet256 and compare with existing methods like EDM, DPM solvers.
> Right now, this is not strongly supported by the experiments. For example, in Table 13, ER-SDE-Solver has worse Recall than the deterministic DPM-solver, which is an important metric for measuring sample diversity.

---

> > ### Author Response · Authors · 2023-11-22
> > **Reply to your valuable comments**
> >
> > We extend our sincere appreciation for your valuable insights. We address your query in detail as outlined below.
> > ## **Q: A precise mathematical definition of the discretization error**
> > We apologize for any inaccuracies in our previous statement. The discretization error, as defined by us, specifically accounts for the inaccuracies in neural network estimation when discretizing the nonlinear integral term $\int_{\sigma_t}^{\sigma_s}\frac{\phi^{(1)}(\sigma)}{\phi^2(\sigma)}x_\theta(x_\sigma,\sigma)d \sigma$.\
> > As mentioned in Sec.3，we solve ER SDE for the data prediction model $x_\theta$. When the neural network can accurately estimate the data state,  $x_\theta(x_\sigma,\sigma) = x_0$. In this scenario, the expression for the solver is:
> > $$
> > x_t =  \frac{\phi(\sigma_t)}{\phi(\sigma_s)}x_s + \phi(\sigma_t) \int_{\sigma_t}^{\sigma_s}\frac{\phi^{(1)}(\sigma)}{\phi^2(\sigma)}x_\theta(x_\sigma, \sigma)d \sigma + \sqrt{\sigma_t^2 - \sigma_s^2\Big[\frac{\phi(\sigma_t)}{\phi(\sigma_s)}\Big]^2 } z_s
> > $$
> >
> > $$
> >  = \frac{\phi(\sigma_t)}{\phi(\sigma_s)}x_s + \phi(\sigma_t) \int_{\sigma_t}^{\sigma_s}\frac{\phi^{(1)}(\sigma)}{\phi^2(\sigma)}x_0 d \sigma + \sqrt{\sigma_t^2 - \sigma_s^2\Big[\frac{\phi(\sigma_t)}{\phi(\sigma_s)}\Big]^2 }z_s
> > $$
> >
> > $$
> > = \frac{\phi(\sigma_t)}{\phi(\sigma_s)}x_s + \phi(\sigma_t)\Big[- \frac{1}{\phi(\sigma)}\Big] \Big|_{\sigma_t}^{\sigma_s}  x_0 + \sqrt{\sigma_t^2 - \sigma_s^2\Big[\frac{\phi(\sigma_t)}{\phi(\sigma_s)}\Big]^2 }z_s
> > $$
> >
> > $$
> > = \frac{\phi(\sigma_t)}{\phi(\sigma_s)}x_s + \Big(1 - \frac{\phi(\sigma_t)}{\phi(\sigma_s)}\Big)x_0 + \sqrt{\sigma_t^2 - \sigma_s^2\Big[\frac{\phi(\sigma_t)}{\phi(\sigma_s)}\Big]^2 }z_s
> > $$
> >
> > $$
> > = \frac{\phi(\sigma_t)}{\phi(\sigma_s)}x_s + FEI x_0 + \sqrt{\sigma_t^2 - \sigma_s^2\Big[\frac{\phi(\sigma_t)}{\phi(\sigma_s)}\Big]^2 }z_s
> > $$
> >
> > While the expression for the first-order solver is:
> >
> > $$
> > \tilde{x} _ t =  \frac{\phi(\sigma_t)}{\phi(\sigma_s)}x_s + \phi(\sigma_t) \int_{\sigma_t}^{\sigma_s}\frac{\phi^{(1)}(\sigma)}{\phi^2(\sigma)}x_\theta(x_\sigma, \sigma)d \sigma + \sqrt{\sigma_t^2 - \sigma_s^2\Big[\frac{\phi(\sigma_t)}{\phi(\sigma_s)}\Big]^2 }z_s
> > $$
> >
> > $$
> > = \frac{\phi(\sigma_t)}{\phi(\sigma_s)}x_s + \phi(\sigma_t) \int_{\sigma_t}^{\sigma_s}\frac{\phi^{(1)}(\sigma)}{\phi^2(\sigma)}d \sigma \ \ x_\theta(x_s,s) + \sqrt{\sigma_t^2 - \sigma_s^2\Big[\frac{\phi(\sigma_t)}{\phi(\sigma_s)}\Big]^2 }z_s
> > $$
> >
> > $$
> > = \frac{\phi(\sigma_t)}{\phi(\sigma_s)}x_s + \Big(1 - \frac{\phi(\sigma_t)}{\phi(\sigma_s)}\Big)x_\theta(x_s,s) + \sqrt{\sigma_t^2 - \sigma_s^2\Big[\frac{\phi(\sigma_t)}{\phi(\sigma_s)}\Big]^2 }z_s
> > $$
> >
> > $$
> > = \frac{\phi(\sigma_t)}{\phi(\sigma_s)}x_s + FEI \ \ x_\theta(x_s,s) + \sqrt{\sigma_t^2 - \sigma_s^2\Big[\frac{\phi(\sigma_t)}{\phi(\sigma_s)}\Big]^2 }z_s
> > $$
> >
> > Therefore, the discretization error can be expressed as:
> > $$
> > discretization \ \ error = x_t - \tilde{x} _ t = FEI(x_0 - x_\theta(x_s,s))
> > $$
> > **In conclusion, a smaller FEI coefficient corresponds to a smaller discretization error, thereby better compensating for inaccuracies in neural network estimation**. We hope our explanation addresses your concerns.

---

> > > ### Comment · Reviewer_RiM8 · 2023-11-22
> > >
> > > Thanks for providing the detailed derivation. However, the third equality is incorrect because $x_0$ is a function of $\sigma$. You cannot treat it as a constant to compute the integral.
> > > $$\phi(\sigma_t)\int_{\phi_t}^{\phi_s} \frac{\phi^{(1)}(\sigma)}{\phi^2(\sigma)}x_0(\sigma)\mathrm{d}\sigma \neq \left. \phi(\sigma_t)\left(- \frac{1}{\phi(\sigma)}\right)\right|_{\sigma_t}^{\sigma_s}x_0(\sigma?). $$

---

> > > > ### Author Response · Authors · 2023-11-23
> > > > **Reply to your valuable comments**
> > > >
> > > > We extend our sincere appreciation for your valuable insights. We address each query in detail as outlined below.
> > > > ### **Q1: Correct the third equality in our response**
> > > > Thanks for pointing it out! There is indeed a small issue in our derivation in the third line. The correction is as follows:\
> > > > When the neural network can accurately estimate the data state,  $x_\theta(x_\sigma,\sigma) = x_0(x_\sigma,\sigma)$. In this scenario, the expression for the solver is:
> > > > $$
> > > > x_t = \frac{\phi(\sigma_t)}{\phi(\sigma_s)}x_s + \phi(\sigma_t) \int_{\sigma_t}^{\sigma_s}\frac{\phi^{(1)}(\sigma)}{\phi^2(\sigma)}x_0(x_\sigma,\sigma) d \sigma + \sqrt{\sigma_t^2 - \sigma_s^2\Big[\frac{\phi(\sigma_t)}{\phi(\sigma_s)}\Big]^2 }z_s
> > > > $$
> > > > When using the first-order Ito-taylor expansion:
> > > > $$
> > > > x_t = \frac{\phi(\sigma_t)}{\phi(\sigma_s)}x_s + \phi(\sigma_t) \int_{\sigma_t}^{\sigma_s}\frac{\phi^{(1)}(\sigma)}{\phi^2(\sigma)} d \sigma  x_0(x_{\sigma_s}, \sigma_s) + \sqrt{\sigma_t^2 - \sigma_s^2\Big[\frac{\phi(\sigma_t)}{\phi(\sigma_s)}\Big]^2 }z_s + R_1
> > > > $$
> > > >
> > > > $$
> > > > x_t = \frac{\phi(\sigma_t)}{\phi(\sigma_s)}x_s + FEI x_0(x_{\sigma_s}, \sigma_s) + \sqrt{\sigma_t^2 - \sigma_s^2\Big[\frac{\phi(\sigma_t)}{\phi(\sigma_s)}\Big]^2 }z_s + R_1
> > > > $$
> > > >
> > > >
> > > > Similarly, the expression for the first-order solver is:
> > > > $$
> > > > \tilde{x} _ t =  \frac{\phi(\sigma_t)}{\phi(\sigma_s)}x_s + \phi(\sigma_t) \int_{\sigma_t}^{\sigma_s}\frac{\phi^{(1)}(\sigma)}{\phi^2(\sigma)}x_\theta(x_\sigma, \sigma)d \sigma + \sqrt{\sigma_t^2 - \sigma_s^2\Big[\frac{\phi(\sigma_t)}{\phi(\sigma_s)}\Big]^2 }z_s
> > > > $$
> > > >
> > > > $$
> > > > = \frac{\phi(\sigma_t)}{\phi(\sigma_s)}x_s + \phi(\sigma_t) \int_{\sigma_t}^{\sigma_s}\frac{\phi^{(1)}(\sigma)}{\phi^2(\sigma)}d \sigma x_\theta(x_{\sigma_s}, \sigma_s) + \sqrt{\sigma_t^2 - \sigma_s^2\Big[\frac{\phi(\sigma_t)}{\phi(\sigma_s)}\Big]^2 }z_s + \tilde{R}_1
> > > > $$
> > > >
> > > > $$
> > > > = \frac{\phi(\sigma_t)}{\phi(\sigma_s)}x_s +FEI x_\theta(x_{\sigma_s}, \sigma_s) + \sqrt{\sigma_t^2 - \sigma_s^2\Big[\frac{\phi(\sigma_t)}{\phi(\sigma_s)}\Big]^2 }z_s + \tilde{R}_1
> > > > $$
> > > > Therefore, the discretization error can be expressed as:
> > > >
> > > > $$
> > > > discretization \ \ error = x_t - \tilde{x} _ t = FEI(x_0(x_{\sigma_s}, \sigma_s) - x_\theta(x_{\sigma_s}, \sigma_s)) + R_1 - \tilde{R}_1
> > > > $$
> > > > **In conclusion, a smaller FEI coefficient corresponds to a smaller discretization error, thereby better compensating for inaccuracies in neural network estimation**. We hope our explanation addresses your concerns.
> > > >
> > > > ### **Q2: The enhanced diversity of the ER SDE solver?**
> > > > Thanks for your question! We focus on elucidating the solution space of ER SDE and enhancing the rapid sampling performance of the SDE Solver, considering diversity as an additional value that serves to motivate our paper. Due to the lack of more reliable metrics to measure diversity in the community, and there haven't been any reports of metrics such as Recall in recent works within the same field[1][2]. Therefore, we have made some modifications to the motivation of our paper. Specifically, in the latest revised version, our research motivation is stated as follows: "Observations in [2][3] indicate that SDE-based stochastic samplers show promise in producing data of superior quality when increasing sampling steps, motivating us to explore the efficient SDE-based stochastic samplers further." (See the last sentence of the first paragraph on page 2).
> > > >
> > > > Here, we reiterate the advantages of stochastic samplers over deterministic samplers, specifically, **stochastic samplers can produce higher-quality images with a minimal increase in NFE** (refer to Table 2, Table 9). **In addition, stochastic samplers with classifier guidance exhibit superior performance in rapid sampling compared to deterministic samplers**(see Table 3).
> > > >
> > > > We sincerely hope that our responses align with your expectations, and we would be extremely grateful if these clarifications contribute positively to the assessment of our manuscript. Thank you for your time and consideration.
> > > >
> > > > [1] SEEDS: Exponential SDE Solvers for Fast High-Quality Sampling from Diffusion Models, Gonzalez et al.\
> > > > [2] Sa-solver: Stochastic adams solver for fast sampling of diffusion models, Xue et al.\
> > > > [3] Score-based generative modeling through stochastic differential equations, Song et al.

---

> ### Author Response · Authors · 2023-11-22
> **Reply to your valuable comments**
>
> We extend our sincere appreciation for your valuable insights. We address your query in detail as outlined below.
> ## **Q: The diversity of generated images may not be strongly supported by the experiments?**
> Thanks for your insightful question! We suppose that the **Recall metric may not adequately reflect the diversity of generated images when the Number of Function Evaluations (NFE) is small**. For instance, when employing DDIM ($\eta=1$)(Stochastic Sampler), the Recall is lower than that of DDIM (Deterministic Sampler), and there exists a significant gap between them. However, when NFE<=20, the Recall of the ER-SDE-Solver is slightly lower than that of the DPM-Solver by 0.01-0.02. **This demonstrates that our approach indeed enhances the diversity of generated images while ensuring efficient sampling performance**. This may be attributed to the lower quality of images when NFE is low, which could potentially impact the Recall metric.
>
>
> Additionally, the pre-trained model employed in Table 13 is Guided-diffusion[1]. According to the results reported in [1], when NFE is around 250, Guided-diffusion achieves its best Recall metric at only 0.65 (Table 5 in [1]). In contrast, the ER-SDE-Solver already reaches a Recall of 0.66 at NFE=30. **Due to limitations in the model's fitting capacity, even with further increases in NFE, it is challenging to achieve significant improvements in Recall**.
>
> We sincerely hope that our responses align with your expectations, and we would be extremely grateful if these clarifications contribute positively to the assessment of our manuscript. Thank you for your time and consideration.
>
> [1] Diffusion models beat gans on image synthesis, Dhariwal P, Nichol A.

---

> > ### Comment · Reviewer_RiM8 · 2023-11-22
> >
> > Thank you for your response. I agree that the Recall metric is not the most reliable way to measure diversity. It would be beneficial if the paper provided additional quantitative methods to measure sample diversity and demonstrate the superiority of the ER SDE solver.
> > However, this is not presented in the paper so far. The main concern is that the paper lacks significant evidence to support the enhanced diversity of the ER SDE solver, both theoretically and empirically. The Recall metric is the only metric provided to measure diversity, and while FID can measure diversity to some extent, it mainly focuses on sample quality. The deterministic sample DPM-solver outperforms the ER-SDE-solver over this metric, raising doubts about whether the ER-SDE solver actually provides much better sample diversity.

---

> ### Comment · Reviewer_RiM8 · 2023-11-23
>
> ## Concern 1:
> I'm not sure how you apply the Taylor expansion here. But I feel the second equation should be
> $$x_t = \frac{\phi(\sigma_t)}{\phi(\sigma_s)}+\phi(\sigma_t) \frac{\phi^{(1)}(\sigma_s)}{\phi^2(\sigma_s)}x_0(x_{\sigma_s}, \sigma_s)\int_{\sigma_t}^{\sigma_s}\mathrm{d}\sigma + ...$$
> Can you provide intermediate steps to derive that?
>
> Putting this derivation aside, suppose your derivation is correct. If I'm not mistaken, you define the $x_0$ to be the data state (some form of score function) of the actual training data distribution.
> The error you defined is the approximation error that arises from the neural network rather than the discretization error. The discretization error is from discretizing a continuous function, which has nothing to do with the accuracy of the neural network. Controlling the approximation error is not a meaningful goal for the sampling problem of the pre-trained diffusion model. The goal here is sampling from the distribution defined by the trained diffusion model rather than sampling from the training data distribution.
>
> ## Concern 2
> Thank you for explaining your motivation in detail. However, I find the paper confusing now, as I'm not sure about its key message. Here are my observations:
>
> 1. In section 4.1, the paper claims that the ODE sampler is superior to the stochastic sampler for rapid sampling based on its error analysis.
>
> 2. In contrast, the authors propose a specialized stochastic sampler, ER-SDE-solver, for rapid sampling, which seems contradictory to the previous claim. The motivation behind this proposal is not clear.
>
> 3. On page 2, the last sentence of the first paragraph suggests that stochastic samplers are more promising for larger sampling steps. However, the paper focuses on rapid sampling that uses small steps. During the rebuttal, the authors stated that their focus is on rapid sampling rather than achieving the highest quality samples. However, they also claim that the ODE sampler is better than the stochastic sampler in the rapid-sampling region. This raises my question of why not just use an ODE sampler instead of proposing a new stochastic sampler.
>
> 4. In response to this question, the authors claim that SDE-based stochastic samplers have the ability to generate diverse images by injecting additional noise into the data state at each sampling step. However, this claim is not well-supported by either empirical or theoretical results. Also, the proposed stochastic solver doesn't even outperform the deterministic DPM-solver on the only metric report in the paper for measuring diversity.
>
> I think this paper might need major revisions to address these concerns. I'll change the rating to 3.

---

> > ### Author Response · Authors · 2023-11-23
> > **Reply to your valuable comments**
> >
> > We extend our sincere appreciation for your valuable insights. We address each query in detail as outlined below.
> > ### **Response to Concern 1**:
> > Your derivation of the second equation is incorrect. The correct derivation is as follows:
> >
> > $$
> > \tilde{x} _ t =  \frac{\phi(\sigma_t)}{\phi(\sigma_s)}x_s + \phi(\sigma_t) \int_{\sigma_t}^{\sigma_s}\frac{\phi^{(1)}(\sigma)}{\phi^2(\sigma)}x_\theta(x_\sigma, \sigma)d \sigma + \sqrt{\sigma_t^2 - \sigma_s^2\Big[\frac{\phi(\sigma_t)}{\phi(\sigma_s)}\Big]^2 }z_s
> > $$
> >
> > Perform Ito-taylor expansion on $x_\theta(x_{\sigma_s}, \sigma_s) $ :
> > $$
> > x_\theta(x_\sigma, \sigma) = \sum_{n=0}^{k-1} \frac{(\sigma - \sigma_s)^n}{n!} x^{(n)}(x_{\sigma_s}, \sigma_s) + R_k
> > $$
> > Consider the first order case, and bring it into the previous equation.
> > $$
> > \tilde{x} _ t =  \frac{\phi(\sigma_t)}{\phi(\sigma_s)}x_s + \phi(\sigma_t) x_\theta(x_{\sigma_s}, \sigma_s) \int_{\sigma_t}^{\sigma_s}\frac{\phi^{(1)}(\sigma)}{\phi^2(\sigma)}d \sigma + \sqrt{\sigma_t^2 - \sigma_s^2\Big[\frac{\phi(\sigma_t)}{\phi(\sigma_s)}\Big]^2 }z_s + \tilde{R}_1
> > $$
> >
> > $$
> > = \frac{\phi(\sigma_t)}{\phi(\sigma_s)}x_s + \phi(\sigma_t) x_\theta(x_{\sigma_s}, \sigma_s) \int_{\sigma_t}^{\sigma_s}d \Big(-\frac{1}{\phi(\sigma)}\Big)  + \sqrt{\sigma_t^2 - \sigma_s^2\Big[\frac{\phi(\sigma_t)}{\phi(\sigma_s)}\Big]^2 }z_s + \tilde{R}_1
> > $$
> >
> > $$
> > = \frac{\phi(\sigma_t)}{\phi(\sigma_s)}x_s + \phi(\sigma_t)  x_\theta(x_{\sigma_s}, \sigma_s) \Big(-\frac{1}{\phi(\sigma)}\Big) \Big| _ {\sigma_t}^{\sigma_s} + \sqrt{\sigma_t^2 - \sigma_s^2\Big[\frac{\phi(\sigma_t)}{\phi(\sigma_s)}\Big]^2 }z_s + \tilde{R}_1
> > $$
> >
> > $$
> > = \frac{\phi(\sigma_t)}{\phi(\sigma_s)}x_s + \Big(1- \frac{\phi(\sigma_t)}{\phi(\sigma_s)}\Big) \ \ x_\theta(x_{\sigma_s}, \sigma_s) + \sqrt{\sigma_t^2 - \sigma_s^2\Big[\frac{\phi(\sigma_t)}{\phi(\sigma_s)}\Big]^2 }z_s + \tilde{R}_1
> > $$
> >
> > $$
> > = \frac{\phi(\sigma_t)}{\phi(\sigma_s)}x_s + FEI\ \ x_\theta(x_{\sigma_s}, \sigma_s) + \sqrt{\sigma_t^2 - \sigma_s^2\Big[\frac{\phi(\sigma_t)}{\phi(\sigma_s)}\Big]^2 }z_s + \tilde{R}_1
> > $$
> > FEI can mitigate errors arising from inaccuracies in neural network estimation, rather than addressing errors resulting from the discretization of the ER SDE, as you mentioned. We disagree with your assertion that controlling estimation errors is meaningless. As the estimation error decreases, the accuracy of the estimation improves.
> >
> > ### **Response to Concern 2:**
> > Our motivation is to obtain fast high-quality samples, which also fall within the research domain of rapid sampling [1][2]. The claim we made that stochastic samplers are more promising for larger sampling steps holds true, as the "larger sampling steps" corresponds to 30~50 NFE, which is relatively small compared to the original 1000 sampling steps. Arbitrarily reducing NFE to 10-20 is not practically feasible. To the best of our knowledge, Stable diffusion is typically configured with 40-50 NFE to achieve high-quality images. Therefore, ER SDE Solver does have an advantage over the ODE Solver in rapidly sampling high-quality images. Additionally, the ER SDE Solver with classifier guidance consistently outperforms the ODE Solver, even when NFE=10. This is why we propose a new stochastic sampler.
> >
> > [1] Martin Gonzalez, Nelson Fernandez, Thuy Vinh Dinh Tran, Elies Gherbi, Hatem Hajri, and Nader Masmoudi. SEEDS: Exponential SDE solvers for fast high-quality sampling from diffusion models. *In Thirty-seventh Conference on Neural Information Processing Systems*, 2023.\
> > [2] Shuchen Xue, Mingyang Yi,Weijian Luo, Shifeng Zhang, Jiacheng Sun, Zhenguo Li, and Zhi-Ming Ma. Sa-solver: Stochastic adams solver for fast sampling of diffusion models. *arXiv preprint arXiv:2309.05019*, 2023

---

### Official Review · Reviewer_yrnV · 2023-11-01

**Soundness:** 2 fair
**Presentation:** 2 fair
**Contribution:** 2 fair
**Rating:** 5
**Confidence:** 3

**Summary:**

This paper introduces a generalized SDE framework called extended reverse-time SDE (ER-SDE) and the solver (ER-SDE-Solver) involved with this generalized SDE formulation. And this paper provides the formulation of the sampling process using the ER-SDE, providing the exact (integral) solution and approximate (linear, or higher-order) solution in both VP/VE cases, which can be generalized to all widely used SDEs. And this paper provides insights on the reasons on why ODE solvers show superior performance in terms of fast sampling. Finally, they validate the image generation performance with ImageNet64 dataset and CIFAR-10 datasets.

**Strengths:**

* Even though existing works generalized the SDE and its equivalent ODE (i.e., yielding the same solution of the Fokker-Planck equation), this paper dealt with the ratio with "how the solver should work as an SDE solver or ODE solver" with dynamically varying rate with respect to time (=sigma, SNR). And by some designing of this new time-dependent variable, this paper showed that some of the new SDE design choices (such as ER-SDE-5) shows superior performance compared to existing methods.
* The writing is concrete, and the additional experiments in the appendix resolved some of my questions (large-scale image datasets, or some ER-SDE ablations.)

**Weaknesses:**

* The design of phi(x) is one of the keys of this paper that distinguishes this to other existing works, but this is not interpreted enough.
* The necessity of the FEI coefficient is vague. Making the FEI coefficient as small as possible is equivalent to directly removing all noise, i.e., h(t)=0. And the trivial question arises; Why not just directly use the ODE solver and the Taylor-approximation-based higher-order sampler?

**Questions:**

* I am not fully understanding the motivation part; why does the low FEI coefficient lead to high sampling performance in low-NFE regime?
* The Figure 3 shows that the FID score is always the best when we use the ODE solver. Then, what is the advantage of the stochastic solver compared to the deterministic solver? And to the best of our knowledge, the FID score is lower (=better) with the stochastic sampler in the high-NFE regime. Even though the dynamically varying phi(x) looks sound, there is not enough evidence of the design of phi(x). Specifially, it will be better if there is some reasoning with the superior performance of ER-SDE-5, compared to other designs.

* What is the phi(x) of ER-SDEs used for experiments in ER-SDE-Solvers of ImageNet64?
* Could you compare your method to other sampling methods, such as PNDM and DEIS?
* What does the 'step' in Figure 3 stand for, in both FEI coefficients and FID scores cases? It seems that the steps stand for the sampling step within the whole 200 steps of the reverse process, and the number of function evaluations (NFE) for FID scores.

* In my opinion, some of the large-scale results in the appendix better explain the benefits of using this ER-SDE-Solver than small-scale results. I recommend aligning some of these results to the main material.

**Details Of Ethics Concerns:**

None.

---

> ### Author Response · Authors · 2023-11-13
> **Reply to your valuable comments**
>
> We extend our sincere appreciation to Reviewer yrnV for the thoughtful consideration of our manuscript and the valuable insights provided. We address each query in detail as outlined below.
> ## **Q1：Why does the low FEI coefficient lead to high sampling performance in low-NFE regime?**
> Thank you for your question! In our paper, Sec.3 demonstrates that the exact solution of the ER SDE comprises three components: a linear function of the data variables, a non-linear function parameterized by neural networks and a noise term. The linear and noise terms can be precisely computed, while discretization errors are present in the non-linear term. Due to the decreasing error as the order increases (see Table1), the first-order error predominantly influences the overall error. Therefore, we exemplify the case with order k = 1 for error analysis.
> It can be observed from Eq.(16) and Eq. (17) that the error arises from the second term
> $\Big{[}1-\frac{\phi(\sigma_{t_{i}})}{\phi(\sigma_{t_{i-1}})}\Big{]} x_\theta(\tilde{x} _ {\sigma_{t_{i-1}}},\sigma_{t_{i-1}})$, as $x_\theta(\tilde{x} _ {\sigma_{t_{i-1}}}, \sigma_{t_{i-1}})$ is a neural network trained in the forward process to estimate data, whose precision is fixed. To reduce discretization error, it is necessary to minimize the coefficient $1 - \frac{\phi(x_t)}{\phi(x_s)}$, referring as the First-order Euler Integral (FEI) coefficient. **Therefore, when using the same model weights, a lower FEI coefficient leads to smaller discretization error. In turn, the images generated in the reverse process more similar to real images in the forward process, resulting in higher image quality and a smaller FID.**\
> In summary, our ER-SDE-Solvers directly utilize raw information without retraining, offering broad applicability and high flexibility. We hope our explanation addresses your concerns.
> ## **Q2：What is the advantage of the stochastic solver compared to the deterministic solver?**
> Thanks for your insightful question!
> 1. **Stochastic samplers can produce higher-quality images with a minimal increase in NFE**(refer to Table 2, Table 9).
> 2. **stochastic samplers with classifier guidance exhibit superior performance in rapid sampling compared to deterministic samplers.** We appreciate your suggestion and have incorporated this result into the main material to further substantiate the benefits of ER SDE 5.
> 3. SDE-based stochastic samplers exhibit an enhanced capability to generate variable images by injecting additional noise into the data state at each sampling step. This conclusion is further supported by Fig.6-Fig.10 in Appendix D, where stochastic samplers demonstrate a greater variability in generated images when the random seed is fixed.
> ## **Q3：What is $\phi(x)$ used for experiments in ER-SDE-Solvers on ImageNet64?**
> $\phi(x)$ used for experiments in ER-SDE-Solvers on ImageNet64 is $φ(x) = x(e^{x^{0.3}} + 10)$（ER SDE 5）. As outlined in Sec.4.2, we select ER SDE 5 as the noise scale function by default if there is no special indication. We hope our clarification addresses your concerns.
> ## **Q4：Compare ER-SDE-Solvers to PNDM and DEIS**
> Thank you for your suggestions! One of the contributions of our paper is to yield mathematical insights elucidating the superior performance of ODE solvers over SDE solvers in terms of fast sampling. Building upon it, we design a stochastic sampler with rapid sampling capabilities comparable to ODEs. To demonstrate the effectiveness of our sampler, we select only a subset of representative ODE-based samplers for comparison due to space constraints.\
> PNDM and DEIS are also ODE-based samplers with superior performance. Following your suggestion, we compare our method with them in Table 10. It can be observed that ODE-based samplers demonstrate superiority in rapid sampling, which further validates our standpoint.
> ## **Q5：What does the 'step' in Figure 3 stand for, in both FEI coefficients and FID scores cases?**
> Your opinion is valid. The steps stand for the sampling step within the whole 200 steps of the reverse process, and the number of function evaluations (NFE) for FID scores. Thank you for your thorough review!
> ## **Q6：Align some of the large-scale results to the main material**
> Appreciate your suggestion! Stochastic samplers guided by classifiers demonstrate superior performance in rapid sampling of high-resolution images compared to deterministic samplers. We have incorporated this result into the main material (Table 3) to further substantiate the benefits of ER SDE 5.
>
>
> We sincerely hope that our responses align with your expectations, and we would be extremely grateful if these clarifications contribute positively to the assessment of our manuscript.

---

> ### Author Response · Authors · 2023-11-22
> **Sincerely looking forward to the further discussions**
>
> Dear reviewer,
>
> We sincerely hope that our recent response and revisions have successfully addressed your concerns. Your feedback is invaluable to us, and if you find that our efforts have met your expectations, we would be immensely grateful if you could reconsider and possibly elevate the score for our work.
>
> If you have any additional questions or suggestions, we would be happy to have further discussions.
>
> Best regards,
>
> The Authors